**Title: Biomass burning emissions in north Australia during the early dry season:**
**an overview of the 2014 SAFIRED campaign**
**Authors:**
Marc D. Mallet[1], Maximilien J. Desservettaz[2], Branka Miljevic[1*], Andelija Milic[1],
Zoran D. Ristovski[1], Joel Alroe[1], Luke T. Cravigan[1], E. Rohan Jayaratne[1], Clare Paton-
Walsh[2], David W.T. Griffith[2], Stephen R. Wilson[2], Graham Kettlewell[2], Marcel V. van
der Schoot[3], Paul Selleck[3], Fabienne Reisen[3], Sarah J. Lawson[3], Jason Ward[3], James
Harnwell[3], Min Cheng[3], Rob W. Gillett[3], Suzie B. Molloy[3], Dean Howard[4], Peter F.
Nelson[4], Anthony L. Morrison[4], Grant C. Edwards[4], Alastair G. Williams[5], Scott D.
Chambers[5], Sylvester Werczynski[5], Leah R. Williams[6], V. Holly L. Winton[7,n], Brad
Atkinson[8], Xianyu Wang[9], Melita D. Keywood[3*]
**Affiliations:**
[1]Department of Chemistry, Physics and Mechanical Engineering, Queensland University of Technology,
Queensland, Brisbane, 4000, Australia
[2]Centre for Atmospheric Chemistry, University of Wollongong, Wollongong, New South Wales, 2522,
Australia
[3]CSIRO Oceans and Atmosphere, Aspendale, Victoria, 3195, Australia
[4]Department of Environmental Sciences, Macquarie University, Sydney, New South Wales, 2109,
Australia
[5]Australian Nuclei Science and Technology Organisation, Sydney, New South Wales, 2232, Australia
[6]Aerodyne Research, Inc., Billerica, Massachusetts, 01821, USA
[7]Physics and Astronomy, Curtin University, Perth, Western Australia, 6102, Australia
[8]Bureau of Meteorology, Darwin, Northern Territory, 0810, Australia
[9]National Research Centre for Environmental Toxicology, Brisbane, Queensland, 4108, Australia
[n]Now at the British Antarctic Survey, Cambridge, CB3 0ET, United Kingdom
**\*Corresponding Authors:**
Dr Melita Keywood
Contact Phone: +613 9239 4596
Contact Email: melita.keywood@csiro.au
Dr Branka Miljevic
Contact Phone: +61 7 3138 3827
Contact Email: b.miljevic@qut.edu.au
**Keywords:**
Biomass burning | savannah fires | greenhouse gases | aerosols | mercury

**Abstract**

The SAFIRED (Savannah Fires in the Early Dry Season) campaign took place from 29[th] of May, 2014 until the 30[th] June, 2014 at the Australian Tropical Atmospheric Research Station (ATARS) in the Northern Territory, Australia. The purpose of this campaign was to investigate emissions from fires in the early dry season in northern Australia. Measurements were made of biomass burning aerosols, volatile organic compounds, polycyclic aromatic carbons, greenhouse gases, radon, speciated atmospheric mercury, and trace metals. Aspects of the biomass burning aerosol emissions investigated included; emission factors of various species, physical and chemical aerosol properties, aerosol aging, micronutrient supply to the ocean, nucleation, and aerosol water uptake. Over the course of the month-long campaign, biomass burning signals were prevalent and emissions from several large single burning events were observed at ATARS.

Biomass burning emissions dominated the gas and aerosol concentrations in this region. Dry season fires are extremely frequent and widespread across the northern region of Australia, which suggests that the measured aerosol and gaseous emissions at ATARS are likely representative of signals across the entire region of north Australia. Air mass forward trajectories show that these biomass burning emissions are carried north west over the Timor Sea and could influence the atmosphere over Indonesia and the tropical atmosphere over the Indian Ocean. Here we present characteristics of the biomass burning observed at the sampling site and provide an overview of the more specific outcomes of the SAFIRED campaign.

## 1. Introduction

Tropical north Australia is dominated by savannah ecosystems. This region consists of dense native and exotic grasslands and scattered trees and shrubs. Conditions are hot, humid and wet in the summer months of December through March with hot, dry conditions for the rest of the year giving rise to frequent fires between June and November each year. Human settlements are relatively scarce in northern Australia, outside of the territory capital, Darwin (population of 146 000). To the north of the continent are the tropical waters of the Timor Sea, as well as the highly populated Indonesian archipelago. South of the savannah grasslands are the Tanami, Simpson and Great Sandy Deserts, spanning hundreds of thousands of square kilometers. Emissions from fires in the savannah regions of northern Australia are therefore the most significant regional source of greenhouse and other trace gases, as well as atmospheric aerosol. Globally, savannah and grassland fires are the largest source of carbon emissions from biomass burning(van der Werf et al., 2010;Shi et al., 2015) and play a significant role in the earth's radiative budget. It is therefore important to quantify, characterise and fully understand the emissions from savannah fires in northern Australia, taking into account the complexity, variability and diversity of the species emitted.

In Australia approximately 550 000 km$^2$ of tropical and arid savannahs burn each year (Meyer et al., 2012;Russell-Smith et al., 2007), representing 7% of the continent's land area. In the tropical north of Australia, the fires during the early dry season in May/June consist of naturally occurring and accidental fires, as well as prescribed burns under strategic fire management practice to reduce the frequency and intensity of more extensive fires in the late dry season in October and November (Andersen et al., 2005).

These fires in the early dry season burn with a low to moderate intensity and are
normally confined to the grass-layer. Events where fires reach the canopy level are rare.
These prescribed burns are an important process for the region and are undertaken by
local landholders with permits, as well as government supported bodies and volunteers.
There has been a recent push to reinstate traditional Aboriginal fire management
regimes in this region (Russell-Smith et al., 2013). Other fire management regimes are
implemented in similar environments around the world, such as the savannah
ecosystems of Africa (Govender et al., 2006) or the chaparral grasses in the United
States (Akagi et al., 2012). In general, fire management regimes are considered to
benefit regional biodiversity and can lead to the long-term increase in living biomass,
resulting in a reduction of greenhouse gas emissions (Russell-Smith et al., 2013).
Quantifying the emissions from dry season fires on regional scales is essential for
understanding the impact of these fires on the local and global atmosphere.

The components and concentrations of emissions from savannah fires are dependent
upon the vegetation and burning conditions. While $CO_2$ is the primary product of
biomass burning (BB), combustion processes also result in the emission of many other
trace gases such as CO, $CH_4$, NOx, $N_2O$ as well as non methane organic compounds
(NMOCs) and aerosol particles composed of elemental carbon, organic carbon and
some inorganic material (Crutzen and Andreae, 1990). The state of organics in biomass
burning aerosols can vary significantly due to the type of plant material burned, the
characteristics of the fires themselves as well as through aging processes in the
atmosphere.

The effects of these emissions on radiative forcing are complex. The global average
radiative forcing due to biomass burning aerosol-radiation interaction is estimated in
the 5[th] International Panel on Climate Change report as 0.0 W m$^{-2}$ with an uncertainty
range of -0.20 to +0.20 Wm$^{-2}$ (Bindoff et al., 2013). It is well known that greenhouse
gases have a positive radiative forcing, heating up the atmosphere. Light absorbing
carbon in the aerosol phase will also result in a positive radiative forcing (Jacobson,
2001) by absorbing shortwave radiation. Conversely, the presence of aerosol organic
and inorganic matter can result in a negative radiative forcing by scattering solar
radiation (Penner et al., 1998). In addition, biomass burning has been shown to be a
significant source of cloud condensation nuclei (CCN), despite typically being
composed of weakly hygroscopic substances  (Lawson et al., 2015), due to the high
number of particles emitted. This can result in a change in cloud droplet concentrations
and volume, thereby influencing cloud formation, albedo and lifetime. The contribution
of each species to the overall radiative forcing is also likely to change as smoke plumes
age (Liousse et al., 1995). Furthermore, not all biomass burning aerosol will interact
with radiation in the same way. For example, fresh BB emissions in the tropics has been
observed to be more absorbing than those from boreal forest fires(Wong and Li, 2002).
The role of biomass burning emissions is not limited to the Earth's radiative budget.
Certain species of emissions (e.g., mercury) can be deposited and sequestered in soil
(Gustin et al., 2008), vegetation (Rea et al., 2002) or bodies of water (LaRoche and
Breitbarth, 2005).

Large-scale studies in Africa (Keil and Haywood, 2003), North America (Yokelson et
al., 2009;Singh et al., 2006), Europe (Saarikoski et al., 2007), South America (Ferek et
al., 1998) and Asia (Lin et al., 2013;Du et al., 2011) have provided valuable insight into
the impact of fire emissions on the regional atmosphere and laboratory measurements
have proved to be useful in understanding the emission factors, composition and
atmospheric processing of these emissions (Stockwell et al., 2014). Despite this, there
is still a need for a better scientific understanding of the influence biomass burning has
on atmospheric composition and air quality (Kaiser and Keywood, 2015), particularly
around Australia. Furthermore, the tropics are disproportionately under-sampled and
the atmospheric and ocean processes in these regions are of both regional and global
consequence. The SAFIRED campaign will contribute towards better understanding
biomass burning emissions and the atmospheric composition in tropical Australia.

On a more specific level, the SAFIRED campaign was undertaken with the following
objectives:
• To obtain Australian savannah fire dry season emission factors for greenhouse

gases, polycyclicaromatic hydrocarbons, gaseous elemental mercury, non-

methane organic compounds, Aitken and accumulation mode aerosols and non-

refractory submicron organic, sulfates, ammonia, nitrates and chlorides.

• To understand the emission of mercury from north Australian fires and to

quantify the delivery of mercury to the ecosystem.

• To characterise the composition and size of aerosols in the region of north

Australia and to understand the influence and extent of biomass burning on the

total aerosol burden.

• To assess the ability of biomass burning aerosol to act as cloud condensation

nuclei and to establish a link between aerosol composition, size and CCN.

•    To assess the fractional solubility of aerosol iron and other trace metals in this

region in the context of the potential supply of micronutrients required for

marine primary production in the ocean.

## 2. Description of experiment

**2.1 Site**
The Australian Tropical Atmospheric Research Station (ATARS; 12°14'56.6"S,
131°02'40.8"E) is located on the Gunn Point peninsula in northern Australia (see Figure
1). ATARS is operated by the Australian Bureau of Meteorology and the CSIRO
(Commonwealth Scientific and Industrial Research Organisation). Standard
meteorological measurements (wind velocity, atmospheric pressure, precipitation) run
permanently at ATARS and two laboratories are in place for the installation of other
instruments. The SAFIRED campaign took place from 29th May 2014 until the 30th
June 2014, with personnel and instruments from nine institutes utilising these
laboratories to make comprehensive gaseous and aerosol measurements during this
period of the early dry season.
**2.2 Instruments and measurements**
**Table 1 A summary of the quantities measured during SAFIRED and the respective instrument or measurement technique. Detection limits and uncertainties are expressed for select**
**instruments or measurements.**

| Quantity | Instrument or Technique | Sample frequency | Reference | Detection limits | Uncertainties |
|---|---|---|---|---|---|
| **CO, CO$_2$, CH$_4$ and N$_2$O** | Fourier transform infrared spectrometry | 3 minute | (Griffith et al., 2012) | 0.04 mg CO$_2$ m$^{-2}$s$^{-1}$, 20 ngN m$^{-2}$s$^{-1}$ (N$_2$O), 30 ng CH$_4$ m$^{-2}$s$^{-1}$) | 0.02 (CO$_2$), 0.2 (CH$_4$), 0.1 (N$_2$O), 0.2 (CO) [a] |
| **O$_3$** | UV Photometric Ozone Analysis | 1 minute | | 0.50 ppb | ~1 ppb |
| **Non methane organic compounds** | Proton Transfer-Mass Spectrometry, high performance liquid chromatography of Supelco cartridge samples; gas chromatography of adsorbant tubes | 3 minute; 12 hour; 12 hour | (Galbally et al., 2007); (Cheng et al., 2016); (Lawson et al., 2015, Dunne et al. (2017)) | 2 - 563 ppt (PTR-MS ions) | <22% (PTR-MS ions) |
| **Polycyclic aromatic hydrocarbons (gas and particle phase)** | Gas chromatography and high resolution mass spectrometry of filter and foam samples | 24 hour | (Wang et al., 2017) | <1 pg m$^{-3}$ | <±20% (rep) |
| **Gaseous elemental mercury; gaseous oxidised mercury; and particulate-bound mercury** | Cold vapour atomic fluorescence spectroscopy | 5 minute; 2 hour; 2 hour | (Landis et al., 2002); (Steffen et al., 2008) | 0.1 ng m$^{-3}$ (GEM), 2 pg m$^{-3}$ (GOM), 2 pg m$^{-3}$ (PBM) | N.R. [b] |
| **Radon** | 700L dual-flow two filter detector | 1 hour | (Chambers et al., 2014) | ±0.04 Bq m$^{-3}$ | 10 - 14% |
| **Aerosol mobility size distributions (14 nm to 670 nm); neutral and charged aerosol size distributions (0.8 nm to 42 nm)** | Scanning mobility particle sizer, Neutral cluster and air ion spectrometry | 5 minute; 4 minute | (Mirme et al., 2007) | - | ±1% in size selection, ±10% in CPC counts |
| **Cloud condensation nuclei concentration (at 0.5% supersaturation)** | Supersaturated streamwise continuous-flow of aerosols in a wetted column using thermal-gradient followed by Optical Particle Counting of activated CCN | 10 second | (Gras et al., 2007) | - | ±0.1% SS, ±20% in OPC counts |
| **Elemental and organic carbon; water soluble ions; and anhydrous sugars (PM$_1$ and PM$_{10}$)** | β+ attenuation; ion chromatography; high performance anion-exchange chromatography | 12 hour | (Chow et al., 2007b); (Iinuma et al., 2009) | 0.0009 μg m$^{-3}$ (oxalate), 0.0002 μg m$^{-3}$ (levoglucosan) | N.R. |
| **Soluble and total fraction of trace metals (PM$_{10}$)** | High-resolution inductively coupled plasma mass spectrometry analysis of extracted leachates and digests. | 24 hour | (Winton et al., 2016) | < 1 pg m$^{-3}$ | ±5% in soluble Fe, ±3% in total Fe |
| **Non-refractory chemical composition (PM$_1$)** | Time-of-flight aerosol mass spectrometry | 3 minute | (Drewnick et al., 2005) | 0.003 μg m$^{-3}$ (NO$_3^-$, SO$_4^{2-}$), 0.03 μg m$^{-3}$ (NH$_4^+$, organics) | ~±20% |
| **Aerosol volatility and hygroscopicity (50 nm and 150 nm)** | Volatility and hygroscopicity tandem differential mobility analysis | 12 minute (full cycle) | (Johnson et al., 2004) | - | ±1% in size sselection, ±1% in RH, ±3% in thermodenuder temperature |

**a** Uncertainty expressed as measurement precision (Allan deviation) for one minute, expressed in $\mu$mol mol$^{-1}$

**b** To be discussed in future work


## 2.2.1 Trace Gases

**Greenhouse gases**

Continuous measurement of $CO_2$, CO, $CH_4$ and $N_2O$ were made using a high precision FTIR trace gas and isotope Spectronus analyser, developed by the Centre for Atmospheric Chemistry at the University of Wollongong. The analyser combines a Fourier Transform Infrared (FTIR) Spectrometer (Bruker IRcube), a pressure and temperature controlled multi-pass cell and an electronically cooled mercury cadmium telluride detector. A detailed description of the instrument and concentration retrieval technique are available in Griffith et al. (2012) and Griffith (1996).

**Ozone and other trace gases**

A Multi Axis Differential Optical Absorption Spectrometer (MAX-DOAS) was installed on the top of one of the laboratories during the campaign. The technique has been shown to provide the vertical profile of nitrogen dioxide, ozone, sulfur dioxide, formaldehyde, glyoxal and aerosol extinction (Sinreich et al., 2005;Honninger et al., 2004). The MAX-DOAS instrument used in this campaign was designed and built at the University of Wollongong. It consists of a vertically rotating prism capturing scattered solar radiation at different angles (1°, 2°, 4°, 8°, 16°, 30° and a reference at 90°) into a fibre optic that carries the radiation to a UV-Visible spectrometer (AvaSpec – ULS3648). Furthermore, a Thermo Scientific model 49i UV Photometric Ozone analyser was used to measure ozone concentrations. Several periods of elevated biomass burning emissions resulted in interferences with the 49i UV analyser and were removed from the analyses. These periods were marked with strong correlations with

high concentrations of acetonitrile where other UV-absorbing species, such as certain
PAH species.
**Non-methane organic compounds**
Online NMOC measurements were made using a high sensitivity Proton Transfer
Reaction-Mass Spectrometer (PTR-MS; Ionicon Analytik) using $H_3O^+$ as the primary
ion. The inlet was 10 m in length and drew air at 5 L min$^{-1}$ from 2 m above the roof
(approx 5.5 m above ground level). The PTR-MS ran with inlet and drift tube
temperature of 60 ºC, 600 V drift tube, and 2.2 mbar drift tube pressure, which equates
to an energy field of 135 Td. The PTR-MS sequentially scanned masses 15-190, with
1 second dwell time. The PTR-MS operated with the aid of auxiliary equipment which
regulates the flow of air in the sample inlet and controls whether the PTR-MS is
sampling ambient or zero air or calibration gas (Galbally et al., 2007).

Furthermore, AT VOC (adsorbent tube Volatile Organic Compounds) samples were
collected by an automatic VOC sequencer which actively draws air through two multi-
adsorbent tubes in series (Markes Carbograph 1TD / Carbopack X). The adsorbent
tubes were then analysed by a PerkinElmer TurboMatrix™ 650 ATD (Automated
Thermal Desorber) and a Hewlett Packard 6890A gas chromatography (GC) equipped
with a Flame Ionization Detector (FID) and a Mass Selective Detector (MSD) at CSIRO
Oceans and Atmosphere laboratories. Further details of the sampling and analyses are
given in Cheng et al. (2016).

During sampling, carbonyls and dicarbonyls were trapped on S10 Supelco cartridges,
containing high-purity silica adsorbent coated with 2,4-dinitrophenylhydrazine
(DPNH), where they were converted to the hydrazone derivatives. Samples were
refrigerated immediately after sampling until analysis. The derivatives were extracted
from the cartridge in 2.5 mL of acetonitrile and analysed by high performance liquid
chromatography with diode array detection. The diode array detection enables the
absorption spectra of each peak to be determined. The difference in the spectra
highlights which peaks in the chromatograms are mono- or dicarbonyl DNPH
derivatives and, along with retention times, allows the identification of the dicarbonyls
glyoxal and methylglyoxal. Further details can be found in Lawson et al. (2015).
**PAHs**
PAHs were sampled through a high-volume air sampler (Kimoto Electric Co., LTD.)
using a sampling rate typically at ~60 $m^{-3}$ $h^{-1}$. The sampling rate was calibrated using
an orifice plate prior to the sampling campaign and the sampling volume was calculated
based on the calibrated sampling rate and sampling duration. A bypass gas meter
installed on the sampler was used to monitor any anomalous fluctuation of the sampling
rate during the sampling period. Particle-associated and gaseous PAHs were collected
on glass fibre filters (Whatman™, 203×254 mm, grade GF/A in sheets) and subsequent
polyurethane foam plugs respectively. The glass fibre filters and polyurethane foam,
along with the field blank samples, were extracted separately using an Accelerated
Solvent Extractor (Thermo Scientific™ Dionex™ ASE™ 350) after being spiked with
a solution containing 7 deuterated PAHs (i.e. $^2D_{10}$-phenanthrene, $^2D_{10}$-fluoranthene,
$^2D_{12}$-chrysene, $^2D_{12}$-benzo[b]fluoranthene, $^2D_{12}$-BaP, $^2D_{12}$-indeno[1,2,3-cd]pyrene,
$^2D_{12}$-benzo[g,h,i]perylene) at different levels as internal standards for quantification
purposes. Concentrated extracts were cleaned up by neutral alumina and neutral silica.
Eluents were carefully evaporated to near dryness and refilled with 250 pg of $^{13}C_{12}$-
PCB (polychlorinated biphenyl) 141 (in 25 μL isooctane) employed as the
recovery/instrument standard for estimating the recoveries of the spiked internal
standards and monitoring the performance of the analytical instrument. Samples were
analysed using a Thermo Scientific™ TRACE™ 1310 gas chromatograph coupled to
a Thermo Scientific™ double-focusing system™ Magnetic Sector high resolution mass
spectrometer. The HRMS was operated in electron impact-multiple ion detection mode
and resolution was set to $\geq$ 10,000 (10% valley definition). An isotopic dilution method
was used to quantify 13 PAH analytes including phenanthrene, anthracene,
fluoranthene, pyrene, benzo[a]anthrancene, chrysene, benzo[b]fluoranthene,
benzo[k]fluoranthene, benzo[e]pyrene, BaP, indeno[1,2,3-cd]pyrene,
dibenzo[a,h]anthracene, benzo[g,h,i]perylene.
**Mercury**
Total gaseous mercury, gaseous elemental mercury + gaseous oxidised mercury (TGM;
GEM + GOM), was sampled from a 10 m mast and measured via gold pre-concentration
and cold vapour atomic fluorescence spectroscopy using a Tekran 2537X instrument.
Simultaneously, GEM, GOM and Particulate-bound mercury (PBM) were individually
measured using a Tekran 2537B connected to a combined Tekran 1130/1135 speciation
unit sampling at a 5.4 m height. The sampling train of the 1130/1135 collects first GOM
(KCl-coated denuder) then PBM (quartz wool pyrolyser) in series from a 10 L min$^{-1}$
sampling flow, allowing GEM only to flow onwards for detection by subsampling by
the 2537B. Due to the small atmospheric concentrations of GOM and PBM, pre-
concentration occurred over a 1-hour period with subsequent analysis taking an
additional hour. Continuous measurements of GEM at 5-minute resolution were made
possible for the 2537B unit by rotating pre-concentration/analysis roles of the two
internal gold traps. Both 2537 units sampled at 1 L min$^{-1}$ and were calibrated every 23
hours using an internal mercury permeation source. For more information on the 2537
and 1130/1135 systems see Landis et al. (2002) and Steffen et al. (2008).

GEM fluxes were measured using the methods outlined in Edwards et al. (2005). Air
samples were drawn at heights of 5.2 and 8.0 m through 46.4 m of nylon tubing using
a PTFE diaphragm pump operating at 10 L min$^{-1}$. Subsampling from this flow through
a 0.2 μm PTFE filter at 1 L min$^{-1}$ by a Tekran 2537A, and switching between sample
intakes, allowed resolution of a GEM gradient every 30 minutes. The transfer velocity
was measured using a Campbell Scientific CSAT3 sonic anemometer and LI-COR
7200 closed path infrared gas analyser for $CO_2$, both located on the same tower as the
gradient intakes at 6.6 m and sampling at 20 Hz.
Radon
In order to measure Radon concentrations, a 700 L dual-flow-loop two-filter radon
detector, designed and built by the Australian Nuclear Science and Technology
Organisation (Whittlestone and Zahorowski, 1998;Chambers et al., 2014), was installed
at the ATARS in 2011 and has been fully operational since July 2012. The detector
provided continuous hourly radon concentrations for the duration of the SAFIRED
campaign, sampling air at 40 L min$^{-1}$ from 12 m above ground level through 25 mm
high-density polyethylene agricultural pipe. A coarse aerosol filter and dehumidifier
were installed "upstream" of the detector, as well as a 400 L delay volume to ensure
that thoron ($^{220}$Rn, half-life 55 s) concentrations in the inlet air stream were reduced to
less than 0.5 % of their ambient values. The detector's response time is around 45
minutes, and the lower limit of detection is 40 - 50 mBq m$^{-3}$. Calibrations are performed
on a monthly basis by injecting radon from a PYLON 101.15±4% kBq Ra-226 source
(12.745 Bq min$^{-1}$ $^{222}$Rn), traceable to NIST standards, and instrumental background is
checked every 3 months. In post processing, half-hourly raw counts were integrated to
hourly values before calibration to activity concentrations (Bq m$^{-3}$).

## 2.2.2 Aerosols

**Aerosol Drying System**

An Automated Regenerating Aerosol Diffusion Dryer (ARADD) is permanently installed on the roof of the laboratory containing the aerosol instrumentation for this campaign. This was used in front of the aerosol manifold to continuously dry the aerosol sample. The ARADD design, similar to that described by Tuch et al. (2009), continuously conditions the aerosol sample to a relative humidity of below 40% with maximum aerosol transmission efficiency. The ARADD utilizes two diffusion drying columns in parallel, each containing 7 stainless steel mesh tubes of 10 mm internal diameter and approximately 800 mm length, surrounded by a cavity packed with silica gel. The aerosol sampled is directed into one column at a time, while the other column is regenerated by an ultra-dry compressed air system. All flows are controlled by software that directs sample flow and compressed air flow to the appropriate column with a series of valves. The ARADD has total suspended particulate style intake at the inlet of the aerosol sample path. This is a non-size-selective stainless-steel inlet with a semi-cicular hat over an inverted conical funnel of variable pitch ending with a 3/4" stainless-steel tube. In practise, the aerosols collected have an equivalent aerodynamic diameter of 100 μm or less depending on sampling conditions. The inlet led to a sample manifold at the exit of the system to provide sampling take-offs for the various aerosol instruments connected to the ARADD. Flow through the ARADD is provided by the instruments and pumps connected downstream. The ambient and inlet relative humidity for the entire sampling period were logged and are displayed in Supplementary Figure S1.

**Aerosol Size**

Aerosol size distributions were measured with a Scanning Mobility Particle Sizer (SMPS). A TSI 3071 long-column electrostatic classifier with a TSI 3772 Condensation Particle Counter (CPC) measured the size distribution over a range of 14 nm to 670 nm at a scan interval of 5 minutes.

In addition to the aerosol size distributions measured by the SMPS, neutral and charged aerosol particle distributions from 0.8 nm to 42 nm were measured using a Neutral cluster and Air Ion Spectrometer (NAIS)(Manninen et al., 2009;Mirme et al., 2007). In this study, the NAIS was set to operate in a cycle of 4 min including ion and neutral particle sampling periods of 2 and 1 minute, respectively, with the remaining minute being an offset period which is required to neutralize and relax the electrodes. The total sampling air flow was 60 L min$^{-1}$, the high flow rate being used to minimize ion diffusion losses and maximize the measured ion concentration sensitivity. Ion losses are accounted for during post-processing of the data by the software (Mirme et al., 2007).

**Aerosol Composition and Water Uptake**

$PM_1$ and $PM_{10}$ 12-hour filter samples (night and day) were collected on a TAPI 602 Beta plus particle measurement system (BAM). Portions of the $PM_1$ filters have been analysed for elemental and organic carbon mass loadings using a DRI Model 2001A Thermal-Optical Carbon Analyzer following the IMPROVE-A temperature protocol (Chow et al., 2007b). Additional portions of the $PM_1$ filters were extracted in 5 ml of 18.2 m$\Omega$ de-ionized water and preserved using 1% chloroform. These extracts have been analysed for major water-soluble ions by suppressed ion chromatography and for

anhydrous sugars including levoglucosan by high-performance anion-exchange
chromatography with pulsed amperometric detection (Iinuma et al., 2009).

Daily aerosol filters were collected using two Ecotech 3000 high-volume volumetric
flow controlled aerosol samplers with $PM_{10}$ size selective inlets. One high-volume
sampler was used to collect aerosols on acid cleaned Whatman 41 filters to determine
the soluble and total fraction of trace metals. Soluble trace metals were extracted from
a filter aliquot using ultra-pure water (>18.2 m$\Omega$) leaching experiments. Total trace
metal concentrations were determined by digesting a second filter aliquot with
concentrated nitric and hydrofluoric acids. Leachates and digested solutions were
analysed by high resolution inductively couple plasma mass spectrometry. The second
sampler was used to collect a set of aerosol samples on quartz filters for elemental and
organic carbon analysis following (Chow et al., 2007a), and major anion and cation
analysis.

The volatility and hygroscopicity of 50 nm and 150 nm particles were measured with a
custom built Volatility and Hygroscopicity Tandem Differential Mobility Analyser
(VH-TDMA). Inlet dried particles were size selected (alternating between 50 and 150
nm) using a TSI 3080 electrostatic classifier. Scans alternated between two different
sample pathways. In the first, after size selection, particles were passed through a
thermodenuder set to 120°C. The sample line was then split so that half went to an
SMPS comprised of a TSI 3080 classifier and a TSI 3010 CPC (V-TDMA). The rest of
the sample was passed through a humidifying system that exposed the particles to a
relative humidity of 90% before being brought into another SMPS with a 3080 classifier
and 3010 CPC (H-TDMA). Alternatively, the thermodenuder was bypassed in every
second scan so that the V-TDMA was used to verify the size selection and the H-TDMA
was able to observe the hygroscopic growth of ambient particles. Each scan ran for 3
minutes, giving a full set of data every 12 minutes.

The chemical composition and properties of non-refractory sub-micron particles were
investigated with a compact Time-of-Flight Aerosol Mass Spectrometer (cToF-AMS,
Aerodyne Research, Inc.) and a Time of Flight Aerosol Chemical Speciation Monitor
(ToF-ACSM, Aerodyne Research, Inc.). Both of these instruments operate with the
same principle and have many identical components. An aerodynamic lens in the inlet
of each instrument focuses the particles into a beam and differential pumping removes
most of the gas phase. Particles are flash vaporized at 600°C and ionized by electron
impact before passing through a time-of-flight mass spectrometer to a multi-channel
plate detector in the cToF-AMS and a dynode detector in the ToF-ACSM. The cToF-
AMS has the added benefit of having a particle Time-of-Flight (pToF) mode, which
allows the size resolved chemical composition to be measured. Both instruments
sampled through a $PM_{2.5}$ inlet and nafion dryer. In addition, the inlet of the cToF-AMS
was incorporated into the VH-TDMA system, so that when the VH-TDMA was
measuring ambient particles, the cToF-AMS would draw particles through the
thermodenuder set at 120°C and vice-versa. This gives additional information about the
chemical composition of the volatile component of submicron particles.

The number of particles activated to cloud droplets were measured using a Continuous-
Flow Steam Wise Thermal Gradient Cloud Condensation Nuclei Counter (CCNC) from
Droplet Measurement Technologies Inc. (DMT, model No. 100). Particles were
exposed to a 0.5% supersaturation and activated particles greater than 1μm were
counted with an Optical Particle Counter using a 50 mW, 658 nm laser diode.
**Back trajectories**
Hourly 10-day air mass back trajectories terminating at ATARS were produced using
the NOAA HYSPLIT model (Draxler and Rolph, 2003), and catalogued in a data base
for use with the SAFIRED campaign data set. Global Data Assimilation System input
files with 0.5° resolution were obtained from NOAA ARL FTP site
(http://ready.arl.noaa.gov/gdas1.php) to drive the HYSPLIT model.
**Satellite detection of fires**
Data on the location of fires was collected from the Australian national bushfire
monitoring system, Sentinel Hotspots. Hotspot locations are derived from the Moderate
Resolution Imaging Spectroradiometer (MODIS) sensors on the Terra and Aqua
satellites and the Visible Infrared Imaging Radiometer Suite (VIIRS) sensor on the
Suomi NPP satellite. The Terra, Aqua and Suomi NPP satellites fly over the region
around ATARS at approximately 10:30 am, 3 pm and 2:30 pm, respectively. Detection
of fires is therefore limited to those that are flaming during these times.

## 3. Overview of Campaign

**Fires and air masses**

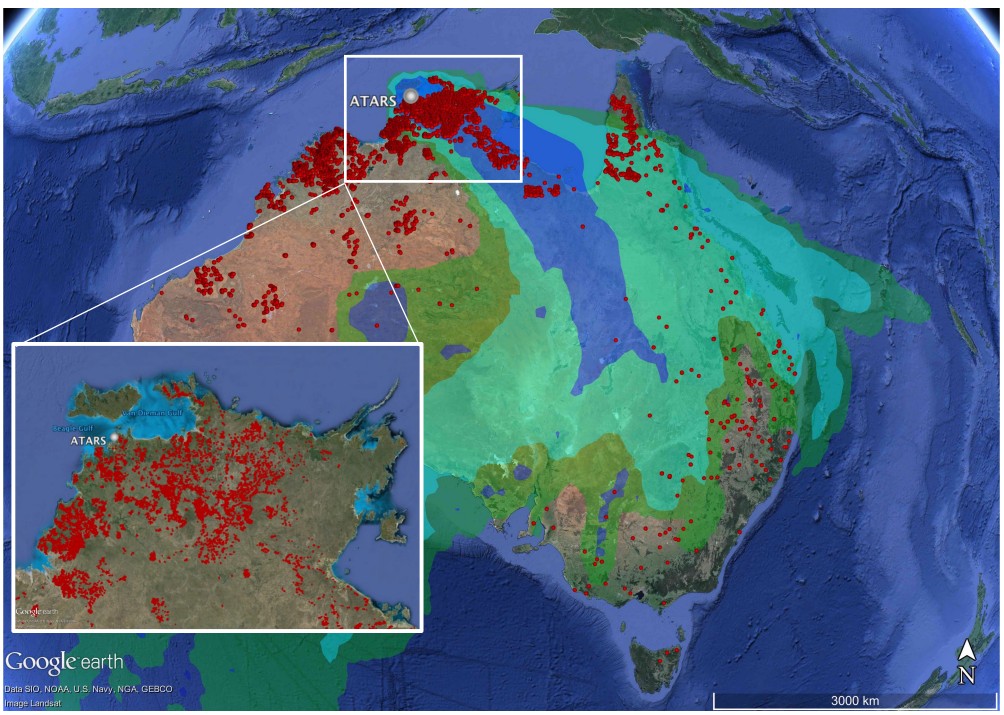

**Figure 1** All satellite-detected fires with >50% detection confidence in June 2014 in Australia. Trajectory densities are shown as shaded regions (blue - >10% of all data; cyan - >1% of all data; green - >0.1% of all data)

Thousands of fires were observed in during the period of the SAFIRED campaign in Australia by the MODIS and VIIRS sensors on the Terra and Aqua NASA satellites. The vast majority of these occurred in the savannah regions of northern Australia. Over 28000 fires were detected within 400 km of ATARS during the sampling period. . Airmass back trajectories from the sampling site show that air masses over the study period predominately originated from the southeast (see Figure 1), generally over the regions where fires were frequently detected. Considering the daily satellite observations of close and distant fires, as well as meteorological, gaseous and aerosol measurements over the duration of SAFIRED, five periods were distinguished; four biomass burning related periods (BBP1, BBP2, BBP3 and BBP4) and a "coastal" period (CP). The dates for these periods are displayed in Table 2.


**Table 2 The start and end dates for the four identified Biomass Burning Periods (BBP1, BBP2, BBP3 and BBP4) and the Coastal Period (CP).**

| Period | Start date (mm/dd/yy hh:mm) | End date (mm/dd/yy hh:mm) |
|--------|------------------------------|----------------------------|
| BBP1 | 05/30/14 00:00 | 05/31/14 23:59 |
| BBP2 | 06/06/14 00:00 | 06/12/14 23:59 |
| BBP3 | 06/14/14 00:00 | 06/17/14 23:59 |
| CP | 06/19/14 12:00 | 06/22/14 23:59 |
| BBP4 | 06/23/14 00:00 | 06/28/14 23:59 |


The number of detected fires on each day within 10 km, 20 km, 50 km, 100 km and 200
km of the sampling location was determined (see Figure 2). Several fires within 10 km
were detected on the 30th of May (BBP1), the 9th and 10th of June (BBP2) and the
25th and 26th of June (BBP4). BBP1, BBP2 and BBP4 were also associated with the
highest concentrations of most of the measured gaseous (Figure 3) and aerosol species
(Figure 4). The periods between the 12th and 23rd of June (BBP3 and CP) had very
few detected fires within 50 km of the station, corresponding to smaller gaseous and
aerosol concentrations.

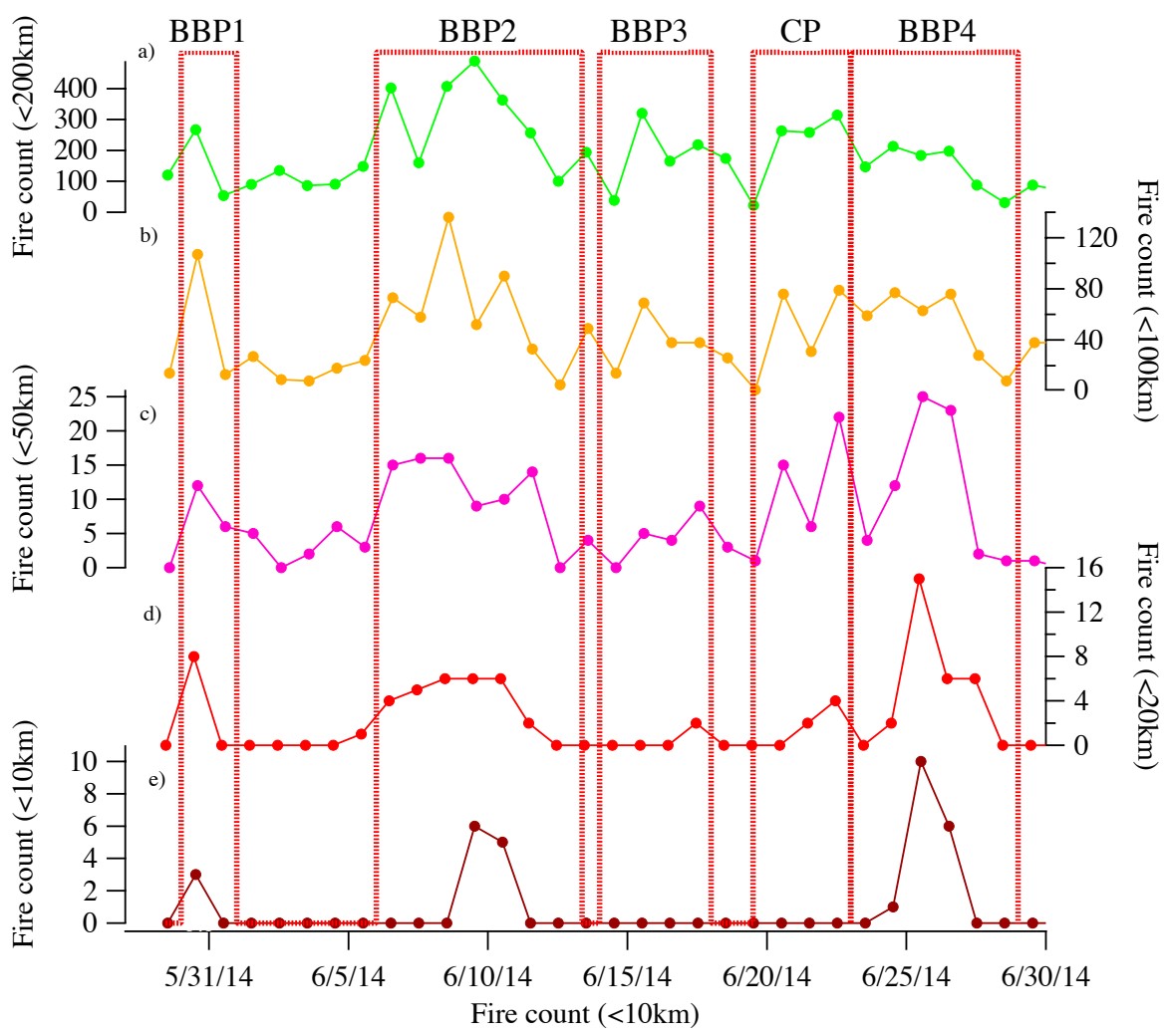


Figure 2 The number of hotspots observed each day within (a) 200 km, (b) 100 km, (c) 50 km, (d) 20 km and (e) 10 km of the ATARS, as detected by the MODIS and VIIRS sensors on the Terra and Aqua satellites.

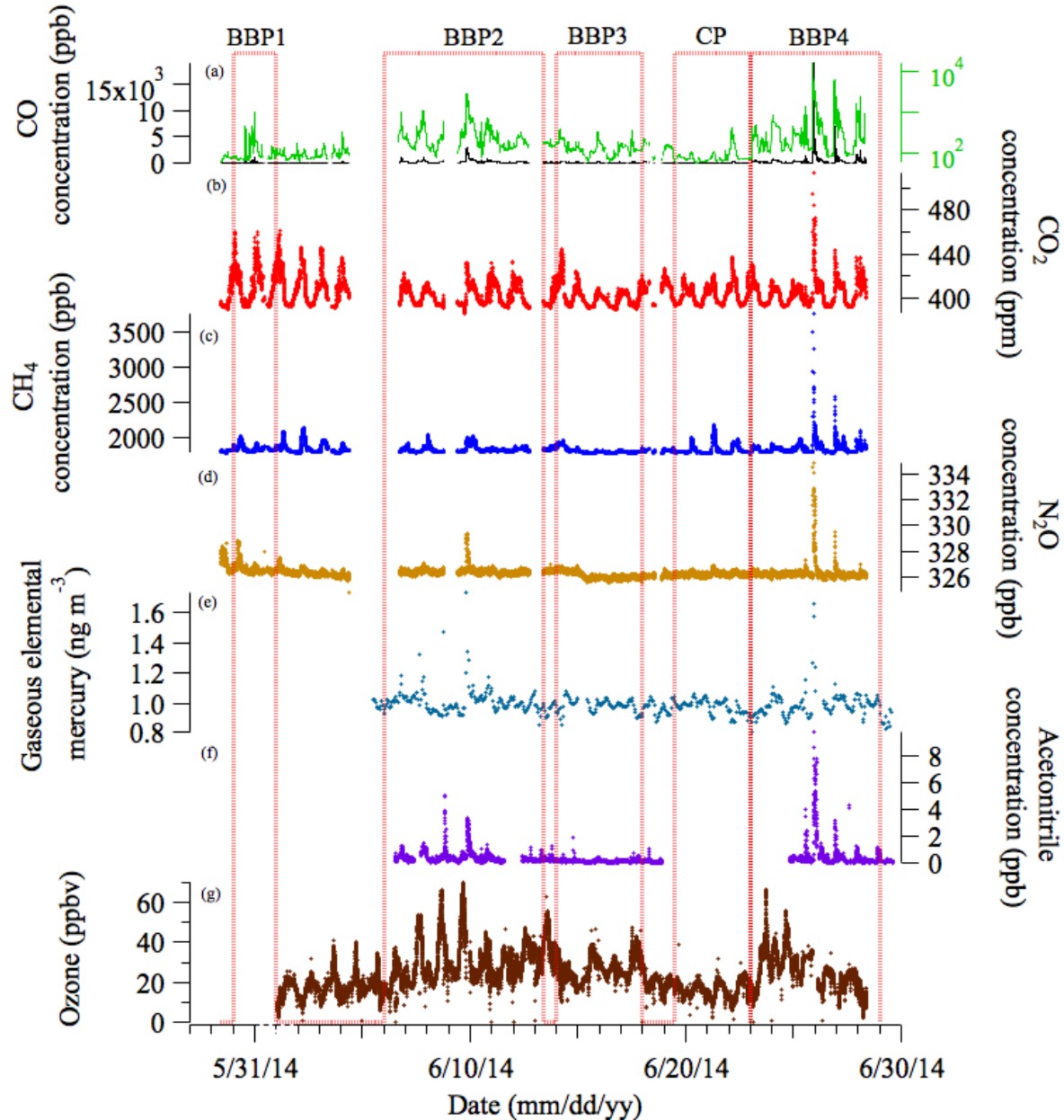


Figure 3 The time series of the major measured gaseous species during the SAFIRED campaign: (a) carbon
monoxide, (b) carbon dioxide, (c) methane, (d) nitrous oxide, (e) gaseous elemental mercury, (f) acetonitrile
and (g) ozone. The biomass burning and coastal periods are indicated by the red dotted lines. All parts-per
notation refer to mole fractions unless otherwise indicated. The date and time is local time.

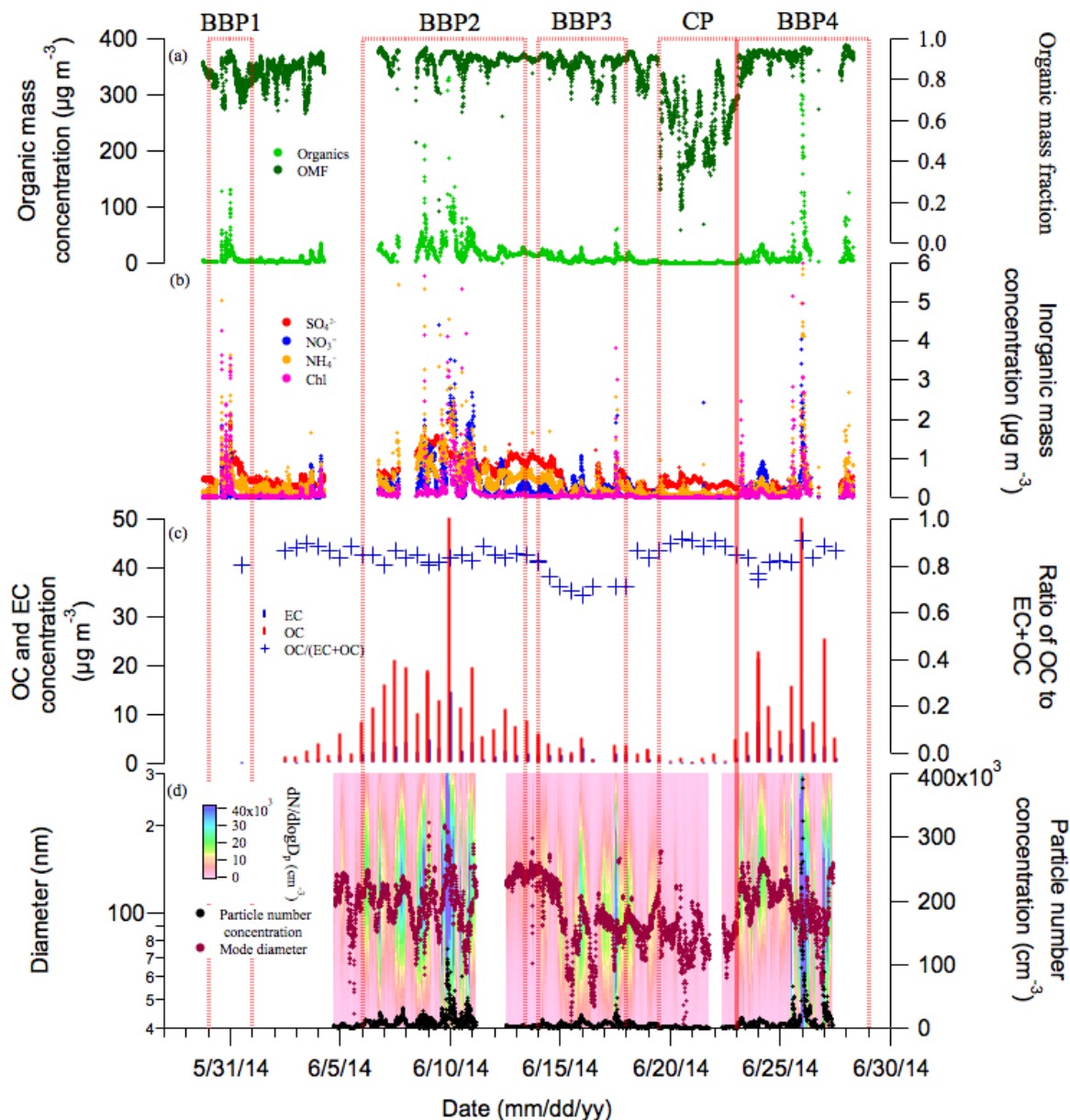

448

Figure 4 The times series of the major aerosol properties during the SAFIRED campaign: (a) the non-refractory PM$_1$ organic mass concentration (left) and organic mass fraction (right), b) the inorganic non-refractory PM$_1$ mass concentrations, (c) the 12-hour filter OC and EC PM$_1$ mass concentrations (left) and the ratio of OC to OC+EC (right), (d) the particle size distributions and particle size mode (left) and the total particle number concentration (right) and (e) the wind direction at ATARS. The date and time is local time.

Most of the gaseous and aerosol time series show a pronounced diurnal trend, with higher concentrations typically observed during the night (see Figure 5 and Supplementary Figure and S2). This is likely due to a combination of variations in fire locations, time of burns, and changes in the boundary layer height or wind velocity. The diurnals trends of radon concentrations, temperature, wind speed, wind direction and greenhouse gases for each of the BBPs and the CP are displayed in Figure 5. The

radon concentrations provide further information regarding the regional air mass
origins and the degree of contact with the land surface and give insight into the
boundary layer. Sharp decreases in the radon concentrations were observed after 09:00
local time and did not increase until after sunset at approximately 18:00 for all periods
(Figure 5a), suggesting a pronounced diurnal variation in the boundary layer height.
Furthermore, radon concentrations were consistently lower during the CP than the BB
periods, suggesting less terrestrial influence than the rest of the sampling period. The
HYSPLIT air mass back trajectory for the CP originated along the east coast of
Australia and passing over little land before arriving at the station. Figure 5d supports
this, showing predominately easterly and northeasterly winds during the night and day,
respectively. The diurnal variations during the BB periods were more pronounced. The
winds during these periods were predominately southeasterly during the night and
morning, turning easterly during the afternoon before reverting at approximately 20:00
local time. The HYSPLIT air mass back trajectories for the BB periods indicated
terrestrial origins, with air masses passing predominately over the savannah region of
northern Australia where the fires occurred.

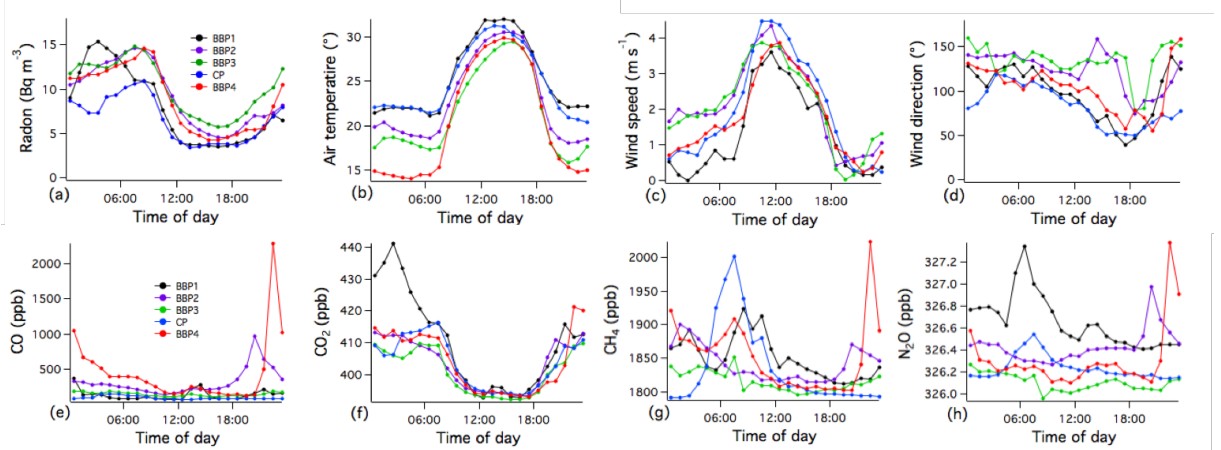


**Figure 5 Mean hourly diurnal (a) radon, (b) wind speed, (c) wind direction (d) dew point temperature (e) CO,**
**(c) CO₂, (d) CH₄, and (e) N₂O at ATARS, separated into different biomass burning periods (BBP) and a**
**coastal period (CP)**
With numerous fires occurring across the region and the limitations of once-per-day
satellite fly-overs and stationary measurements, it can be difficult to identify the exact
source of these elevated signals. Nonetheless, it is possible to link detected plumes with
fires given back trajectory analysis. The elevated signals during BBP1 were likely a
result of several fires that were burning and observed on the 30th of May at 14:00 local
time approximately 2 and 10 km from ATARS during the day. While the elevated
signals were observed later in the evening, it is likely that they were due to a
continuation or evolution of those fires. Some of the most intense signals of the
campaign were observed during BBP2, with numerous close (within 50 km) and distant
(within 200 km) detected. Due to the limitations of the once-per-day satellite fly-by, it
was only possible to link one of the observed plumes to a source during this period. A
large event observed on the evening of the 9th of June was likely due to a cluster of
fires detected approximately 5 km southeast of ATARS. Only one fire within 20 km of
ATARS was observed via satellite during BBP3 on the 17th of June but this was not
associated with any significant increase in gaseous or aerosol concentrations. Several
fires were also observed between 20 km and 50 km from the station. One close fire was
also observed during CP, however wind directions during this period were typically
north-easterly and concentrations were therefore much lower. 5-day HYSPLIT
trajectories also show that air mass during the CP originated along the east coast of
Australia before travelling towards the sampling station with very little terrestrial
influence.

For a portion of BBP4, fires were burning within several kilometers of ATARS and
several plumes were easily observed from the station.  The signals from these plumes
are shown in Figure 6. The observed enhancements between 12:30 and 15:00 pm on
the 25th June during BBP4 were a result of grass fires burning approximately 2 km
south-east from the station. During this event, the wind direction was highly variable,
changing between 140° and 80° True Bearing (TB) multiple times. As a result, the
sampling changed from measuring the air mass with and without the plume from this
fire, which led to sharp increases and decreases in biomass burning-related signals.
Visually, the fire area and extent of the plume was larger at 4:00pm than earlier,
however the wind direction changed to north-easterly which directed the plume away
from the station.   From 16:00 until 22:00, the wind direction was stable at
approximately 50° TB. At 22:00, the wind direction rapidly changed to directly south
and the largest enhancements for the whole campaign were observed until
approximately 2:00 am on the 26th of June. It is very likely that these signals were a
result of a continuation and evolution of these fires as the night progressed. Portions of
a ~0.25 $km^2$ grassland field within 500 m directly south of ATARS were observed to
be burned upon arrival at the station on the morning of the 26th of June and we speculate
that the burning of this field contributed to the large enhancements in measured biomass
burning emissions. The emissions during this portion of BBP4 are likely to be the most
representative of fresh biomass burning smoke during the SAFIRED campaign.
Significant ozone enhancements over 80 ppb were observed during this event, although
this was likely result of a cross-contamination due to concurrently high concentrations
of UV-absorbing organic compounds in the gaseous phase. This enhancement would
only be possible with significant photochemical processing which is very unlikely
considering the time of the event, the visual evidence of close fires, and the large
concentrations observed.

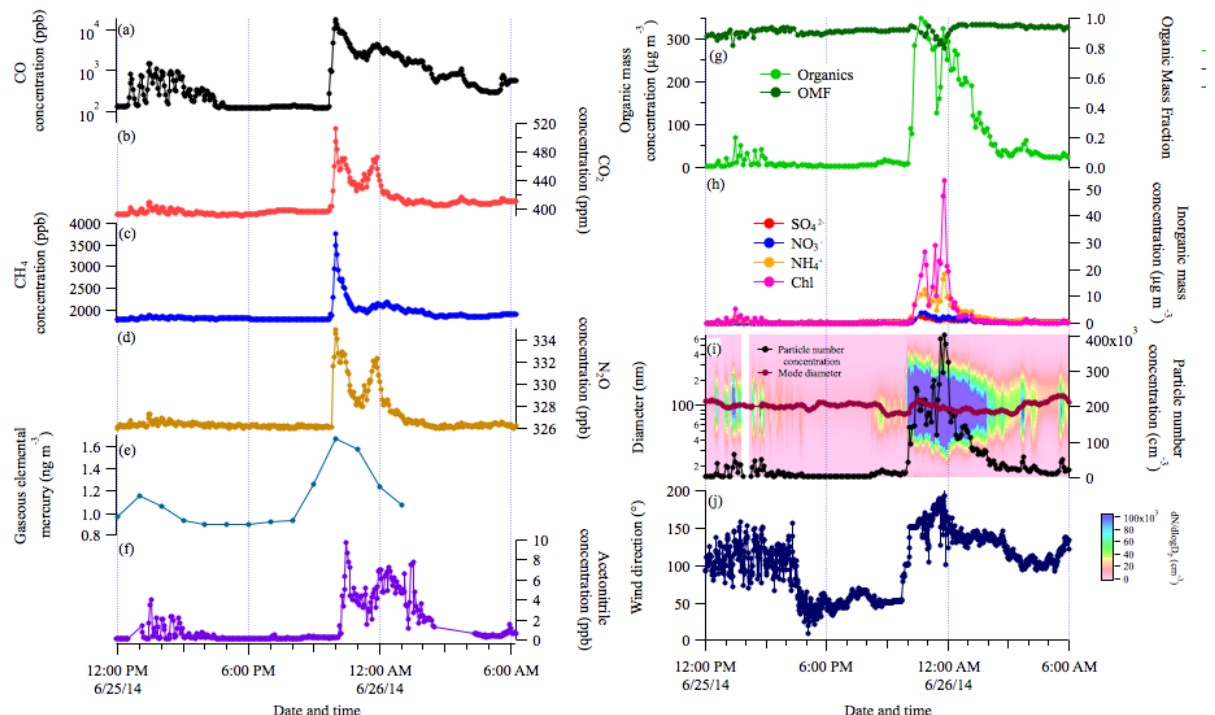


**Figure 6 The major gas and aerosol concentrations measured during two biomass burning events within 1 km of ATARS during BBP4. (a) through (g) and (h) through (k) are as per Figures 3 and 4, respectively. All parts-per notation or mole fractions unless otherwise indicated. The date and time are local time.**

Based on the elevated concentrations of biomass burning related gaseous and aerosol species, detection of close fires and the air mass back trajectory analysis during portions of BBP1, BBP2 and BBP4, these periods are likely associated with fresh biomass burning smoke from nearby fires. With smaller concentrations and more distant observed fires, the signals observed during BBP3 are possibly more characteristic of aged biomass burning smoke. The influence of biomass burning during CP was much smaller than the rest of the campaign. Investigating the relationship between toluene and acetonitrile, two NMOCs emitted from biomass burning, can provide further information on the aging of BB emissions. Toluene is much shorter lived than acetonitrile as it readily reacts in the presence of the OH radical. Assuming a consistent emission ratio of these two NMOCs from fires in this region, the ratio of toluene/acetonitrile thereby provides a proxy for photochemical age. Unfortunately, the PTR-MS which measures these species was not operational during BBP1 and CP. The

diurnal trends for the toluene and acetonitrile concentrations and the
toluene/acetonitrile ratio is shown in Figure 7 for BBP2, BBP3 and BBP4. The
toluene/acetonitrile ratio was highest during the night, indicating more photochemically
aged smoke throughout the day. Interestingly, while the toluene and acetonitrile
concentrations were consistently higher during BBP2 and BBP4 than BBP3, the
toluene/acetonitrile ratio was of the same magnitude and followed the same trend. It is
therefore plausible that, while there were not large enhancements in concentrations
during BBP3 and there were few fires detected close-by during the daytime satellite
flyovers, there were small-scale burns during the night that were close enough for the
emissions to reach sampling site. This observation highlights the limitation of using
satellite hotspot detection in fully understanding the aging processes of biomass
burning emissions.

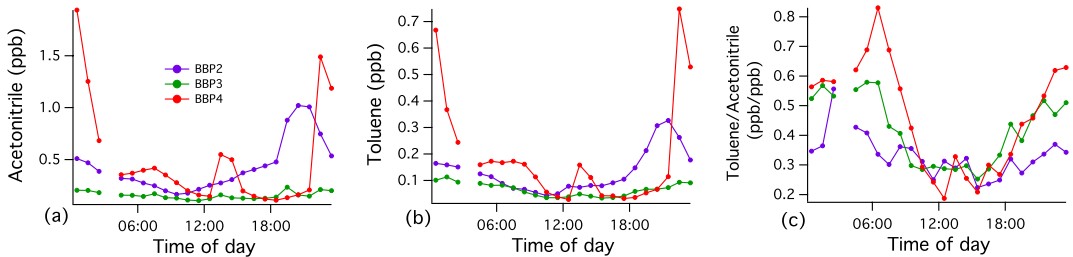


**Figure 7 Mean hourly diurnal (a) acetonitrile concentration, (b) toluene concentration, (c) toluene/acetonitrile**
**ratio, separated into different biomass burning periods (BBP).**

Particle size distributions were unimodal for the majority of the sampling period with
a mode of approximately 100 nm on average (see Figure 8). The SMPS was not
operational during BBP1. Although the shape of the BBP4 size distribution was similar
to the campaign average, concentrations were much higher and a result of close fires.
BBP2 had a slightly larger size distribution centered on 110 nm. The size distribution
during BBP3 was slightly smaller than the campaign average and BBP2 and BBP4,
with a mode centered on ~95 nm. Furthermore, the diurnal trends of the BBA mode
diameter during BBP2, BBP3 and BBP4 and CP all showed a clear maximum during
the night (see Supplementary Figure S2d). The diurnal trends of the toluene/acetonitrile
ratios (Figure 7c) as well as the ratio of oxygenated organic aerosol to total organics
(see Supplementary Figure S2c) suggest that the larger night time particle sizes are
more associated with fresh biomass burning. The contrast between these size
distributions could be a result of atmospheric aging and dilution in which organic mass
condenses onto or evaporates from the particle. Variations in fuel load or burning
conditions could also contribute to this difference. The size and concentration of
particles during the Coastal Period (CP) were much smaller than the rest of the
campaign. There were two periods during CP where a bimodal size distribution was
observed; one from approximately 3 pm until midnight on the 19[th] of June and the other
between 2 pm and 6 pm on the 20[th] of June. The size distributions for both of these
periods had a mode at approximately 20 nm and another at approximately 85 nm.
Submicron sulfates made up to 32% of the total submicron non-refractory mass
concentrations, as reported by the cToF-AMS from the period of midday on the 19[th] of
June until midnight on the 22[nd] of June, whereas the average sulfate contribution for
the rest of the campaign was approximately 8%. The low radon values, small particle
concentrations, bimodal size distributions and significant contributions of sulfate
during this period also suggest very little biomass burning signal and a more marine-
like aerosol. No particle nucleation events were observed over the entire sampling
period (See Supplementary Figure S3). This is likely due to the elevated particle
concentrations acting as a condensation sink.

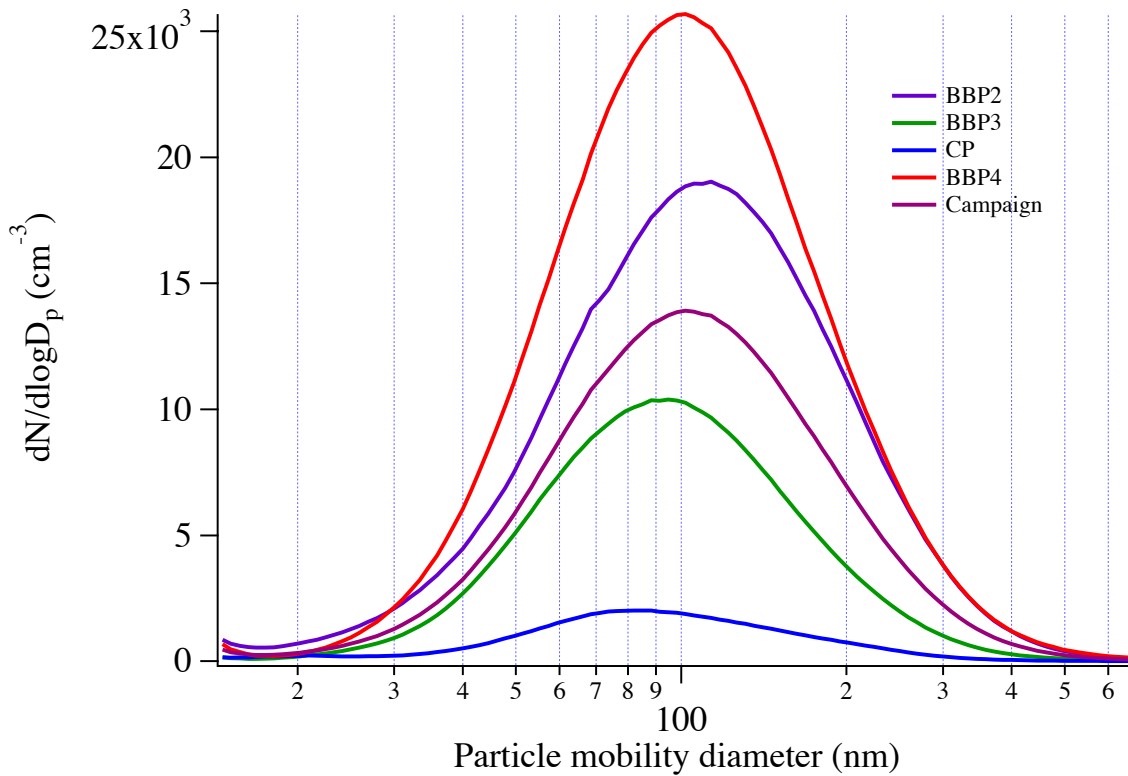


**Figure 8 The average number size distribution during BBP2, BBP3, BBP4, CP and the campaign average.**

Over the campaign, organics dominated the non-refectory sub-micron aerosol mass
contributing, on average, 90% of the total mass. Sulphate, nitrates, ammonium and
chloride species contributed the rest of this mass, with the largest contributions from
sulphate and ammonium. Sulphate contributions were very significant during the
coastal period, contributing up to 32% of the total mass. Although chlorides contributed
the least to the total mass on average, during clear biomass burning events where sharp
increases in CO and organics were observed, chlorides made up the largest component
of inorganic aerosol. Organic carbon made up approximately 80% to 90% of the total
carbon (organic carbon + elemental carbon) $PM_1$ mass during the campaign, with the
exception of BBP3, when this dropped to 70%. Whether these observations were a
result of burn conditions or aging processes (i.e. evaporation of organic compounds
from the aerosol phase) is unclear.

## 4. Outcomes of SAFIRED

The overall aim of this study was to investigate the characteristics of BB emissions in the tropical savannah region of northern Australia during the early dry season. For many gaseous and aerosol species, elevated signals were observed for much of the month-long sampling period due to the high frequency of fires. Further analysis of these species can provide more insight into the impact of these fires on the regional atmosphere. Table 2 displays a summary of companion studies undertaken within the SAFIRED campaign.

Table 3 A list of currently published companion studies undertaken during SAFIRED.

| Reference | Title |
|---|---|
| **Winton et al., (2016)** | Dry season aerosol iron solubility in tropical northern Australia |
| **Wang et al., (2017)** | Emissions of selected semivolatile organic chemicals from forest and savannah fires |
| **Milic et al., (2017)** | Biomass burning and biogenic aerosols in northern Australia during the SAFIRED campaign |
| **Mallet et al., (2017)** | Composition, size and cloud condensation nuclei activity of biomass burning aerosol from northern Australian savannah fires |
| **Desservattaz et al., (2017)** | Emission factors of trace gases and particles from tropical savanna fires in Australia |

| **Howard et al., (2017)** | Atmospheric mercury in the southern hemisphere tropics: seasonal and diurnal variations and influence of inter-hemispheric transport |
|---|---|


### 4.1. Emission factors and gaseous species loadings

Desservettaz et al. (2017) identified individual plumes with high signals during
SAFIRED in order to determine emissions factors $CO_2$, CO, $CH_4$. $N_2O$, as well as GEM,
Aitken and Accumulation mode aerosols and submicron non-refractory particle species
(organics, sulfates, nitrates, ammonium and chlorides). Seasonal emission factors for
the major greenhouse gases are important for national greenhouse gas inventories and
in understanding the impact of savannah fires. Furthermore, these results will be the
first set of emission factors for aerosol particles from savannah fires in Australia, with
early results suggesting higher factors than those observed from African and South
American savannah fires. Emission factors were mostly found to be dependent on the
combustion conditions (using the modified combustion efficiency as a proxy) of the
fires.

Wang et al. (2017) investigated 13 major PAH compounds in both the gaseous and
aerosol phase during the SAFIRED campaign and estimated their emission factors from
savannah fires, as well as for subtropical eucalypt forest fires. Concentrations of these
PAHs varied from from ~ 1 to over 15 ng $m^{-3}$ within different BB periods and the
emission factor for savannah fires for $\sum_{13}$ PAHs were estimated to be $1600 \pm 110$ µg
$kg^{-1}$ In the gas phase, 3- and 4-ring compounds typically contributed ~ 90% to the sum
concentrations whereas the particle-associated PAHs were dominated by 5- and 6-ring
compounds (> 80%). Measured PAH concentrations were significantly higher during
BBP2 and BBP4. During these periods, concentrations of BaP exceeded the monitoring
investigation level for atmospheric BaP (0.30 ng m$^{-3}$) in Australia (National-
Environment-Protection-Council-Service-Corporation, 2011) by up to 200%.

Biomass burning produces significant amounts of semi-volatile NMOC which can be
difficult to quantify and identify with current measurement techniques. However recent
studies have shown that including semi volatile NMOC chemistry in models improves
the agreement between the modeled and observed organic aerosol (Alvarado et al.,
2015; Konovalov et al., 2015) and ozone (Alvarado et al., 2015). High quality NMOC
emission factors are crucial for models to assess the impact of biomass burning plumes
on air quality and climate. Future analyses will be undertaken on the SAFIRED data to
quantify emission factors for various NMOCs.

SAFIRED represents the first measurements of atmospheric mercury undertaken in the
tropical region of the Australian continent. The mean observed GEM concentration
over the study period was $0.99 \pm 0.09$ ng m$^{-3}$, similar to the average over that month
(0.96 ng m$^{-3}$) for 5 other Southern Hemisphere sites and slightly lower than the average
(1.15 ng m$^{-3}$) for 5 tropical sites (Sprovieri et al., 2016). Mean GOM and PBM
concentrations were $11 \pm 5$ pg m$^{-3}$ and $6 \pm 3$ pg m$^{-3}$ respectively, representing 0.6 –
3.4% of total observed atmospheric mercury. During periods of pronounced trace gas
and aerosol concentrations during the campaign, spikes in GEM concentrations were
also observed, though there were no significant increases in GOM or PBM. Emission
ratios calculated during the campaign were two orders of magnitude higher than those
reported by Andreae and Merlet (2001). Future outcomes from the SAFIRED campaign
will focus on the use of micrometeorological techniques and the passive tracer radon to
quantify delivery of atmospheric mercury to tropical savannah ecosystems. ATARS
also now serves as an additional site measuring continuous GEM as part of the Global
Mercury Observation System (GMOS), one of only two tropical observing sites in the
Eastern Hemisphere and the third such site located in Australia. A discussion of the
seasonal and diurnal variations of atmospheric mercury at the ATARS site can be found
in Howard et al. (2017).
**4.2. Biomass burning aerosol chemistry**
Milic et al., (2017) provided further analysis into the aerosol chemical composition to
elucidate the aging of early dry season biomass burning emissions. Fractional analysis
(e.g., f44 and f60, the fraction of m/z 44 and m/z 60 to all organic masses, indicated
oxygenation and BB sources, respectively) and factor analysis using positive matrix
factorisation (PMF) of cToF-AMS data was investigated over the entire sampling
period. Outside of the periods of significant influence from BB events, three PMF-
resolved organic aerosol factors were identified. A BB organic aerosol factor was found
to comprise 24% of the submicron non-refractory organic mass, with an oxygenated
organic aerosol factor and a biogenic isoprene-related secondary organic aerosol factor
comprising 47% and 29%, respectively. These results indicate the significant influence
of fresh and aged BB on aerosol composition in the early dry season. The emission of
precursors from fires is likely responsible for some of the SOA formation.

The water uptake of aerosols during SAFIRED was further investigated in Mallet et al.,
(2017) to identify the influence of early dry season BB in this region on cloud
formation. The concentrations of cloud condensation nuclei at a constant
supersaturation of 0.5% were typically of the order of 2000 $cm^{-3}$ and reached well over
10000 $cm^{-3}$ during intense BB events. Variations in the ratio of aerosol particles
activating cloud droplets showed a distinct diurnal trend, with an activation ratio of
40% ± 20% during the night and 60% ± 20% during the day. The particle size
distribution and the hygroscopicity of the particles were found to significantly influence
this activation ratio. Particles were generally extremely hydrophobic, particularly
during the night and during the BB periods shown in this paper. Modelling CCN
concentrations using the size distributions of aerosols and typical continental and
terrestrial values of hygroscopicities yielded significant over predictions of up more
than 200%, highlighting the need to include more regional parameterisations of aerosol
composition and hygroscopicity.

Furthermore the fractional solubility of aerosol iron and other trace metals during
SAFIRED were investigated in Winton et al., (2016). The fractional iron solubility is
an important variable determining iron availability for biological uptake in the ocean.
On a global scale, the large variability in the observed fractional iron solubility results,
in part, from a mixture of different aerosol sources. Estimates of fractional iron
solubility from fire combustion (1 - 60 %) are thought to be greater than those
originating from mineral dust (1 - 2%) (Chuang et al., 2005;Guieu et al., 2005;Sedwick
et al., 2007), and may vary in relationship to biomass and fire characteristics as well as
that of the underlying terrain (Paris et al., 2010;Ito, 2011). Iron associated with BB may
provide information with respect to BB inputs of iron to the ocean (Giglio et al.,
2013;e.g. Meyer et al., 2008). The ATARS provides an ideal location to further
investigate BB derived fractional iron solubility at the source. The results from this
study can be found in Winton et al. (2016) and show that soluble iron concentrations
from BB sources are significantly higher than those observed in Southern Ocean
baseline air masses from the Cape Grim Baseline Air Pollution Station, Tasmania,
Australia (Winton et al., 2015). Aerosol iron at SAFIRED was a mixture of fresh BB,
mineral dust, sea spray and industrial pollution sources. The fractional iron solubility
(2 - 12%) was relatively high throughout the campaign and the variability was related
to the mixing and enhancement of mineral dust iron solubility with BB species.

## 719   **5. Conclusions and looking forward**

Biomass burning was found to significantly influence the surface atmospheric
composition during the 2014 early dry season in north Australia. Over 28000 fires were
detected via satellite retrieval during the sampling period. Several periods were
identified when fires within 20 km of the research station resulted in significant
enhancements of greenhouse gases, non-methane gaseous organic compounds, gaseous
elemental mercury and polycyclic aromatic hydrocarbons and aerosol loadings. Much
of the $PM_1$ mass was comprised of organic material. The aerosol particle number size
distributions were typically unimodal and centered around 100 nm which is smaller
than BBA observed in other regions. The analysis of the time series of these measured
quantities has so far allowed the quantification of savannah fire emission factors for
these aerosol and gaseous species and has provided and understanding of the aerosol
aging, water uptake and solubility in this region.

While the specific outcomes of the SAFIRED campaign are reviewed in the previous
section, the general importance of this study can be discussed in a greater context. This
is the first large-scale collaborative project undertaken in this region and draws on the
resources and expertise of most of Australia's research institutes focused on atmosphere
chemistry and composition. Large scale, multidisciplinary measurement campaigns in
the tropics, such as SAFIRED, are needed to make distinctions between different types
of fires in different regions to reduce uncertainties in global climate models (Keywood
et al., 2013). This need has been recognized with the formation of global collaborative
initiatives promoting interdisciplinary collaboration in biomass burning research
(Kaiser and Keywood, 2015). As the world moves towards a warmer climate, it is
plausible that the frequency and intensity of biomass burning will increase, and these
emissions will become an increasingly important source of trace gases and aerosols to
the atmosphere.

SAFIRED lays the foundation for future measurements at ATARS that could make
measurements throughout the whole dry season and on a more long-term scale. Future
work in this region should focus on 1) the detailed characterisation of individual fires
and their emissions, 2) biomass burning emissions throughout the late dry season and
3) the vertical and horizontal transport of biomass burning emissions in this region.
With well-established emission factors, a concentrated effort should be made to link
modelled aerosol gaseous and aerosol loadings with *in situ* and remote sensing
measurements. This should be done not just at the surface, but throughout the boundary
layer as well as over the waters north of Australia. Furthermore, a further investigation
of the radiative influence of the gaseous and aerosol species should be done for this
region.
**Data availability**
All data are available upon request from the corresponding authors (Branka Miljevic,
b.miljevic@qut.edu.au; Melita D. Keywood; melita.keywood@csiro.au).

## Author Contributions

M.D. Mallet[a,b,c,d,e], M.J. Desservettaz[b,c,d,e], B. Miljevic[b,c,e*], A. Milic[b,d,e], Z.D. Ristovski[b,e], J. Alroe[b,c,e], L.T. Cravigan[b,c,e], E.R. Jayaratne[d,e], C. Paton-Walsh[b,c,e], D.W.T. Griffith[b,d,e], S.R. Wilson[b,d,e], G. Kettlewell[b,e], M.V. van der Schoot[b,e], P. Selleck[b,c,d,e], F. Reisen[b,c,e], S.J. Lawson[b,c,d,e], J. Ward[b,c,d,e], J. Harnwell[b,c,e], M. Cheng[b,c,d,e], R.W. Gillett[b,c,d,e], S.B. Molloy[d,e], D. Howard[b,c,d,e], P.F. Nelson[b,e], A.L Morrison[b,e], G.C. Edwards[b,c,e], A.G. Williams[b,c,e], S.D. Chambers[b,c,d,e], S. Werczynski[b,c,e], L.R. Williams[c,d,e], V.H.L. Winton[b,c,d,e], and B. Atkinson[b,c], X. Wang[b,d,e], M.D. Keywood[b,c,d,e,f*]

a: Wrote and organised the manuscript

b: Contributed to the organisation of the campaign

c: Installed and/or operated instrumentation during the sampling period

d: Analysed data

e: Contributed to the manuscript and/or data interpretation

f: Designed and led the campaign.

*: Corresponding author

## Competing interests

The authors declare that they have no conflict of interest.

## Acknowledgements

The majority of the campaign was internally funded. The input of QUT was supported by the Australian Research Council Discovery (Grant DP120100126). The work on aerosol iron solubility was supported by Curtin University (RES-SE-DAP_AW-47679-

1), the University of Tasmania (B0019024) and the Australian Research Council (Grant
FT130100037).

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
