# Peer review of "Title: Biomass burning emissions in north Australia during the early dry season"

_Atmospheric Chemistry and Physics, 2016_

## Referee Comment (RC1) · Anonymous Referee #1 · 3 Dec 2016

Review of ACP-2016-866

This paper is the overview paper for a special issue on the SAFIRED experiment, and is intended to introduce the project, provide details on the instrumentation, meteorology, and give a brief synopsis of the results that will be discussed in the individual papers. This paper does a reasonable job of this, although with a number of places that require some clarification (see below). One major drawback with this paper is that it is too long and spends too much time and detail on the results. Presumably, those results are covered in detail in the individual papers. Another problem is that the text in almost all the figures is too small to read, and the time needs to be defined, is it local time?

Abstract; This section is too long and needs to be tightened up considerably. There are also the following problems:
Line 45. How does one measure the "mercury cycle"? It is possible to measure the chemical species that make up the mercury cycle.
Line 47. The word "emitted' is redundant.
Lines 52 and 53. What distinguishes 'intense' and 'close' smoke plumes?
Lines 59 and 60. These few sentences are examples of extraneous material not appropriate to an abstract.

Introduction:
Lines 80 and 81. Savannah and grassland fires are not the largest source of carbon to the atmosphere, as is clear when comparing the numbers from the quoted references with the global anthropogenic source of $CO_2$ for example. Do the authors mean the largest source of black carbon?
Line 109. NOx is not an incomplete combustion product, in fact NO is most definitely a flaming stage compound. The authors would know this if they referred to the numerous references that have come after Crutzen and Andreae [1990], Akagi et al [2011] (referenced later on)s is a nice recent review of BB emissions.

Description of experiment;
Line 194. If this is meant to be only place the details of PTRMS calibration is discussed, then we need a reference or further explanation for calculating approximate response factors.
Line 241. The term PBM need to be defined here.
Lines 256-258. Doesn't the $CO_2$ from fires mess with the 'transfer velocity' measurements?
Lines 265-267. The half-life of Radon is much longer than that of NOx (which is about 3-4 hours) about the same as $SO_2$, and shorter than aerosols (which is about 2 weeks), not sure how to think about water vapor.
Line 307. What is a 'Total Suspended Particulate style inlet'? I've never heard of this, so it needs to be explained further or a reference given that explains it.
Line 335. Are you saying that the extracts have not been analyzed yet?

Overview of the Campaign;
Lines 446-450. Isn't it both boundary layer mixing and time spent over land that determine Rn concentrations? This section needs a better explanation of how these two effects were differentiated.

Line 457 and Table 1. The term "background concentration" is difficult to define and is not consistently applied in this paper. Those medians should not be considered "background" values. Background to me means the value that would be observed in the absence of a continental source (urban, fire etc.). This is particularly true for CO, 130ppbv is much higher than background, which is probably around 90. I point out that later in the paper, (line 554) much lower numbers were quoted for O3 and CO, 10 ppbv and 66ppbv, which are obviously too low. Table 1. The measurements do not justify the number of significant figures reported for most of these quantities.

Figure 5 Caption. Plot (d) is mistakenly attributed to 'nitrogen dioxide' and should be nitrous oxide.

Figure 6. The legends and scale insets are too small and, in one case, not next to the panel to which they refer.

Lines 496-498. It is well known in the community that the AMS technique does not work for much of the chloride that one finds in the atmosphere, particularly near coasts, because it is in the form of refractory salts. What is worse is that because of background subtraction issues, other chloride is actually under-measured. Fires likely emit chloride as ammonium chloride, which is volatile and will be measured by AMS. These are probably the main reasons for your observations.

Lines 504-506. Another cause of this effect is the above limitation of the AMS instrument.

Lines553-558. The ratio $\Delta O_3/\Delta CO$ is not a good indicator of photochemical age or processing. One only has to look at the high $O_3$ plume during BBP4 (6/26 as far as I can tell), which reaches 100ppbv. This obviously had substantial photochemical processing, how else does that much $O_3$ get made? Yet the ratio is low, probably because the CO had not been mixed out as much as in the other plumes. You need to find some other indicator.

Line 623. The authors seem to be ignoring the high $O_3$ plume during BBP4 which indicates faster $O_3$ production in this plume. This would seem to be one of the more interesting observations of this study.

---

## Referee Comment (RC2) · Anonymous Referee #3 · 8 Dec 2016

Review for the paper: Biomass burning in north Australia: overview of the 2014 SAFIRED campaign.

This paper gives an overview of the SAFIRED2014 campaign in Northern Australia aimed at investigating biomass burning in an area that has very frequent burning but is clearly understudied. This paper suffers from the typical issues of overview papers, where there is a long introduction of instruments and methods, but no actual results. In this paper especially the last section "Outcomes of SAFIRED" is very long, includes short literature reviews, but teases at potential results and points to other related papers without giving any results. Overview papers clearly serve a purpose and should include four major points: 1) description of the science goals and how the campaign

comment

was designed to answer them, 2) a systematic description of the used instrumentation, 3) a big picture overview of the results and 4) a conclusion of how the campaign results are usable for answering the science question. This overview paper here describes most of the above points, but could benefit from some improvements and in particular would benefit from summarizing the results more systematically.

Specific Comments:

I think it would be helpful to actually list the specific science questions at the end of the introduction or in a new section before the instrument descriptions.

Instruments and Measurements

- The chapter 2.2 Instruments and Measurements should be made more consistent between the individual instrument descriptions and also misses some critical information. Most of the instrument detection methods are described well, but the most important information for all the measurements are missing. For each instrument description the following needs to be added: sensitivity (precision and accuracy), limit of detection, time resolution and used inlet. A table should be added that lists all of these instrument parameters and also a reference to the technique.

- The Radon instrument description also includes a summary of how Radon measurements are used in atmospheric research. This is not appropriate here and should be moved to the results section around page 21.

- The chapter Aerosols should be numbered consistently with 2.2.2

Fires and Air Masses

- What I was mostly missing in this chapter was putting SAFIRED into the bigger picture of fire emissions in Australia, e.g.: how representative is SAFIRED, was this a typical year and what could SAFIRED potentially tell us about emission estimates in northern Australia. How many fires did you observe during SAFIRED? How many of those measured plumes were fresh (for emission ratios) and how many were aged?

[Figure]

- Figure 4: The data here are split into weak moderate and strong mixing, but nothing is really done with this separation later. Also the differences are not very strong. In the next Figure and the rest of the manuscript the data get separated into different BB and costal periods. This seems a better separation. I suggest removing the mixing categories. I am also wondering how the wind direction plot looks for the Coastal Period. This would be more helpful for a separation.

- Figure 4c y-axis should go from 0-360.

- page 24 line 473: What are the criteria used to separate the data into these periods? The separation seems very arbitrary to me, especially what is the difference between BBP2 and BBP3. Also the coastal period has large CO mixing ratios and very similar O3/CO ratios as BBP3. Please explain in more detail what is difference between the periods and how you define BBP. Are these by CO or acetonitrile enhancements, back trajectories, or fire counts?

- diurnal trend e.g.: page24 line 470-471: The authors argue here and in other places that the diurnal variations are caused by the mixing height. This is probably right, but no actual evidence is presented. The wind direction changes as can be seen clearly in Figure 4. Looking only at the time series in Figures 5 and 6 one cannot judge, if the diurnal changes align with wind direction change or more with the Radon profile. A diurnal profile of some trace gases and aerosol species should be added. I would also like to see that separated for the different BBP and CP.

- Figures 5 and 8. It would be good to also show the CO data on a linear scale.

Close Proximity Fires versus Aged Fires

- On several places on pages 27-30 the age of fire plumes are discussed in rather vague terms sometimes using organic aerosol or size distributions as chemical indicator in addition to the fire locations. To show photochemical aging the most commonly used way is to look at ratios of a short lived tracer to an inert tracer on the time scale

of the transport. Ratios of some of the VOC measurements versus CO or acetonitrile would be best used to show aging, most commonly used are aromatic species, benzene for longer time scales, toluene or larger aromatics for shorter time scales. Enhancement ratios of fresh fires seem be available from the "close proximity fires" or nighttime fire plumes, although I have my doubts about how close those fires were, as I will describe below. Fires in the region are relatively similar and the emission ratios should therefore also be similar enough to distinguish between fresh plumes and plumes transported over 200-300km to the site using VOC/CO ratios. I would suggest replacing all the vague discussions about plume age with adequate VOC/CO enhancement ratios.

- O3/CO ratios: The O3/CO is used in Figures 5 and 8 and is described at giving an indication of photochemical age. Unfortunately O3/CO are much more complicated than that and depend on many different factors such as VOC/NOx ratios such that the ratio really cannot be used as "photochemical age". I think for this paper here, it is best to remove the O3/CO ratios instead of adding a proper explanation.

- The ozone enhancement shown in Figure 8 for the close proximity fire is substantial and ozone values of almost 100ppb are detected in the plume. This means that there has been significant photochemical processing of potentially several hours during plume transport. If the plume would be really fresh, ozone would actually be titrated. Again VOC/CO could be very helpful here and should be looked into. Also a comparison to a nighttime plume measurement would be very useful. Again, I doubt that this plume is very fresh.

Outcomes of SAFIRED

The paper is rather long in its current form and in particular this chapter is more of a literature review, of what could potentially be done with the specific measurement. I actually think this is not appropriate for an overview paper and would be more appropriately discussed in the detailed follow-up papers. I suggest deleting this whole section

and just briefly mentioning the potential major outcomes in the "Looking Forward" section.

The picture quality of all Figures needs to be improved.

---

## Referee Comment (RC3) · Anonymous Referee #2 · 23 Dec 2016

Mallet et al. provide an overview of the multi-institutional measurement campaign conducted in Northern Australia during the dry season to measure the emissions and transformations of trace gases and particles emitted by savannah and grassland fires. The motivation for the measurement campaign is novel, the manuscript is well written and the results are appropriately described. The measurements from this campaign are likely to improve our understanding of biomass burning emissions at the local and global scale. The only major concern I have is that the manuscript, being an overview article, could be improved in terms of the presentation of the campaign specific information and data (see comment #s 3, 4, 5, 11, 12). This would make the manuscript much more citable and a serve as a gateway for anyone interested in SAFIRED-related literature. I recommend publication of the manuscript after the following minor comments have been addressed and/or clarified.

1. The resolution of the figures is too poor and needs to be fixed. I would ask the authors to consider using vector images.

2. The font size on the figures in some cases is too small and very hard to discern on a printed copy of the manuscript.

3. While some of the relevant literature has been cited, it would be worthwhile to discuss (likely in the Introduction) similar measurement campaigns performed in other parts of the world that have examined emissions from biomass burning and how those earlier lab and field efforts (e.g., BBOP, SCREAM, FLAME1-5, etc) have helped inform critical gaps, research questions, instrumentation, analysis techniques etc. for the SAFIRED campaign.

4. Being an overview article, I think the manuscript could benefit from a schematic and/or cartoon in the introduction that sketches the region of interest (Northern Australia) and caricatures the emissions, processes and impacts being studied in detail in this campaign. Furthermore, a bulleted list in the beginning of the manuscript that lists the research/science questions for SAFIRED would provide context for the various measurements and analysis performed.

5. In the methods section, the manuscript could benefit from a Table that lists the instrument, quantity measured, accuracy/precision, frequency. For example, see Figure 1b, 1c, 2b, 2c, etc in Ryerson et al., (JGR, 2013).

6. In some cases (e.g., non methane organic compounds), too much detail is provided in the methods section describing the measurement.

7. I did not bother to investigate this further but I wasn't quite sure what the technical definition of the word 'fetch' is. It might be helpful to clarify this for the reader.

8. Too many significant figures (up to 6!) for some of the measurements in Table 1.
9. Figure 7: Are those raw SMPS data or lognormal fits to the SMPS data? The distributions look uncannily smooth.

10. Line 590: The word 'aging' is commonly used to refer to chemical reactions but in the example is used here to refer to thermodynamics.

11. The 'Outcomes' section could benefit from the following: (i) discussion of the results in the context of earlier work and how the findings here are similar or different, (ii) how the SAFIRED measurements were insightful (iii) what questions still remain unanswered, and (iv) directions for future work.

12. Similar to comment 5, a Table listing the companion publications and its central finding would be helpful for the interested reader to track the measurement-specific paper.

13. Line 658-677: Will the NMOC emissions and speciation be discussed in a forthcoming publication? I did not see a SAFIRED-related reference for this section.

14. Line 762-763: How does primary organic aerosol interact with NMOCs to form SOA? I am not sure this sentence is phrased correctly. Do you mean primary organic aerosol serves as a seed for the SOA produced from NMOC oxidation?

15. Clarification question: Were aircrafts used to study the biomass burning plumes?

16. While I understand that the majority of the companion papers that deal with the specifics of each measurement are in the process of being prepared or are currently under review, are there any novel campaign-wide conclusions that the authors would like to discuss in the concluding section of the manuscript?

---

## Author Response (AR1)

acp-2016-866 Author Comment

**Anonymous Referee #1**

This paper is the overview paper for a special issue on the SAFIRED experiment, and is intended to introduce the project, provide details on the instrumentation, meteorology, and give a brief synopsis of the results that will be discussed in the individual papers. This paper does a reasonable job of this, although with a number of places that require some clarification (see below). One major drawback with this paper is that it is too long and spends too much time and detail on the results. Presumably, those results are covered in detail in the individual papers.

Another problem is that the text in almost all the figures is too small to read, and the time needs to be defined, is it local time?

*The authors thank the referee for their comments and suggestions. The manuscript has undergone numerous changes, including removing or shortening sections that were previously too long. Furthermore, figures now have a higher resolution and with larger text. The date and time has also been defined as local time within the figures and discussion.*

Abstract; This section is too long and needs to be tightened up considerably. There are also the following problems:

Line 45. How does one measure the "mercury cycle"? It is possible to measure the chemical species that make up the mercury cycle.

Line 47. The word "emitted' is redundant.

Lines 52 and 53. What distinguishes 'intense' and 'close' smoke plumes?

Lines 59 and 60. These few sentences are examples of extraneous material not appropriate to an abstract.

*(now L47) The term "mercury cycle" has been replaced with "speciated atmospheric mercury".*

*(now L49) The word "emitted" has been removed.*

*The sentence discussing "intense or close" smoke plumes has been removed.*

*(now L61) The last section of the abstract has been replaced with a concise sentence summarising the content of the manuscript.*

Introduction:

Lines 80 and 81. Savannah and grassland fires are not the largest source of carbon to the atmosphere, as is clear when comparing the numbers from the quoted references with the global anthropogenic source of $CO_2$ for example. Do the authors mean the largest source of black carbon?

*(now L91) This sentence has been amended to mention that savannah and grassland fires are the largest source of carbon emissions **from biomass burning**.*

Line 109. NOx is not an incomplete combustion product, in fact NO is most definitely a flaming stage compound. The authors would know this if they referred to the numerous references that have come after Crutzen and Andreae [1990], Akagi et al [2011] (referenced later on)s is a nice recent review of BB emissions.

*(now L119) "Incomplete combustion" has been changed to "combustion processes".*

Description of experiment;

Line 194. If this is meant to be only place the details of PTRMS calibration is discussed, then we need a reference or further explanation for calculating approximate response factors.

*The discussion of instrument calibrations has been removed from the manuscript.*

Line 241. The term PBM need to be defined here.

*(now L294) Particulate-bound mercury (PBM) has now been defined at the first point of mention.*

Lines 256-258. Doesn't the $CO_2$ from fires mess with the 'transfer velocity' measurements?

*(now L312) The transfer velocity and $CO_2$ measurements were taken with two different instruments, as indicated.*

Lines 265-267. The half-life of Radon is much longer than that of NOx (which is about 3-4 hours) about the same as $SO_2$, and shorter than aerosols (which is about 2 weeks), not sure how to think about water vapor.

*(now R316) The section discussing Radon has been significantly shortened and this sentence is no longer present.*

Line 307. What is a 'Total Suspended Particulate style inlet'? I've never heard of this, so it needs to be explained further or a reference given that explains it.

*(now L374) The TPS inlet has now been given a brief description*

Line 335. Are you saying that the extracts have not been analyzed yet?

*(now L409) "These extracts will be analysed..." has been changed to "These extracts have been analysed".*

Overview of the Campaign;

Lines 446-450. Isn't it both boundary layer mixing and time spent over land that determine Rn concentrations? This section needs a better explanation of how these two effects were differentiated.

*(now L547) This section has been altered in the context of the biomass burning periods (BBPs). Both the diurnal variations and the variations in the magnitude of radon concentrations across different BBPs at the same daily hour provide insight into the boundary layer and the terrestrial residence time.*

Line 457 and Table 1. The term "background concentration" is difficult to define and is not consistently applied in this paper. Those medians should not be considered "background" values. Background to me means the value that would be observed in the absence of a continental source (urban, fire etc.). This is particularly true for CO, 130ppbv is much higher than background, which is probably around 90. I point out that later in the paper, (line 554) much lower numbers were quoted for O3 and CO, 10 ppbv and 66ppbv, which are obviously too low. Table 1. The measurements do not justify the number of significant figures reported for most of these quantities.

*The table with the summary measurements has been removed along with the discussion of "background concentrations".*

Figure 5 Caption. Plot (d) is mistakenly attributed to 'nitrogen dioxide' and should be nitrous oxide.

*(now Figure 3) This has been fixed.*

Figure 6. The legends and scale insets are too small and, in one case, not next to the panel to which they refer.

*(now Figure 4) This has been fixed.*

Lines 496-498. It is well known in the community that the AMS technique does not work for much of the chloride that one finds in the atmosphere, particularly near coasts, because it is in the form of refractory salts. What is worse is that because of background subtraction issues, other chloride is actually under-measured. Fires likely emit chloride as ammonium chloride, which is volatile and will be measured by AMS. These are probably the main reasons for your observations.
Lines 504-506. Another cause of this effect is the above limitation of the AMS instrument.

*(now L813) The speculation of the origin of chloride being due to the coastal location has been removed.*

Lines553-558. The ratio $\Delta$O3/$\Delta$CO is not a good indicator of photochemical age or processing. One only has to look at the high O3 plume during BBP4 (6/26 as far as I can tell), which reaches 100ppbv. This obviously had substantial photochemical processing, how else does that much O3 get made? Yet the ratio is low, probably because the CO had not been mixed out as much as in the other plumes. You need to find some other indicator.

*(now Figure 7) $\Delta$O3/$\Delta$CO data and the discussion of it has been removed from the manuscript. Acetonitrile and toluene data has been included and their ratios have been discussed in the context of photochemical processing.*

Line 623. The authors seem to be ignoring the high O3 plume during BBP4 which indicates faster O3 production in this plume. This would seem to be one of the more interesting observations of this study.

*(now Figure 4) The spike in the O3 during BBP4 has been attributed to cross-contamination within the ozone analyser. Substantial photochemical processing of that plume was highly unlikely given large concentrations observed more than 4 hours after sunset, observed fire proximities, and indicators within the AMS (e.g f44).*

**Anonymous Referee #2**

Mallet et al. provide an overview of the multi-institutional measurement campaign conducted in Northern Australia during the dry season to measure the emissions and transformations of trace gases and particles emitted by savannah and grassland fires. The motivation for the measurement campaign is novel, the manuscript is well written and the results are appropriately described. The measurements from this campaign are likely to improve our understanding of biomass burning emissions at the local and global scale. The only major concern I have is that the manuscript, being an overview article, could be improved in terms of the presentation of the campaign specific information and data (see comment #s 3, 4, 5, 11, 12). This would make the manuscript much more citable and a serve as a gateway for anyone interested in SAFIRED-related literature. I recommend publication of the manuscript after the following minor comments have been addressed and/or clarified.

*The authors thank the referee for their comments and suggestions. We have taken these on board and have improved the quality of the manuscript accordingly.*

The resolution of the figures is too poor and needs to be fixed. I would ask the authors to consider using vector images.

The font size on the figures in some cases is too small and very hard to discern on a printed copy of the manuscript.

*The resolution of the figures has been improved (either larger files or .eps format) and the font size has been increased.*

While some of the relevant literature has been cited, it would be worthwhile to discuss (likely in the Introduction) similar measurement campaigns performed in other parts of the world that have examined emissions from biomass burning and how those earlier lab and field efforts (e.g., BBOP, SCREAM, FLAME1-5, etc) have helped inform critical gaps, research questions, instrumentation, analysis techniques etc. for the SAFIRED campaign.

*(L151) A short discussion of previous field and laboratory measurements has been added to the end of the Introduction.*

Being an overview article, I think the manuscript could benefit from a schematic and/or cartoon in the introduction that sketches the region of interest (Northern Aus- tralia) and caricatures the emissions, processes and impacts being studied in detail in this campaign. Furthermore, a bulleted list in the beginning of the manuscript that lists the research/science questions for SAFIRED would provide context for the various measurements and analysis performed.

*(L 164) The authors think that a schematic/cartoon is not appropriate for this publication. A bulleted list at the end of the Introduction has been added, however.*

In the methods section, the manuscript could benefit from a Table that lists the instrument, quantity measured, accuracy/precision, frequency. For example, see Figure 1b, 1c, 2b, 2c, etc in Ryerson et al., (JGR, 2013).

*(L196 and Table 1) A table summarising the quantities measured and the instruments used has been included at the beginning of the Instruments and measurements section.*

In some cases (e.g., non methane organic compounds), too much detail is provided in the methods section describing the measurement.

*The discussion of calibration techniques used in the NMOC measurements has been removed for concision and consistency with the other measurements.*

I did not bother to investigate this further but I wasn't quite sure what the technical definition of the word 'fetch' is. It might be helpful to clarify this for the reader.

*This term is no longer present within the manuscript.*

Too many significant figures (up to 6!) for some of the measurements in Table 1.

*This table was superfluous and has been removed from the results section.*

Figure 7: Are those raw SMPS data or lognormal fits to the SMPS data? The distributions look uncannily smooth.

*(now Figure 8) The data presented in this figure are averaged raw SMPS data. They are smooth due to unimodal shape and little variation in the size distributions during each BBP.*

Line 590: The word 'aging' is commonly used to refer to chemical reactions but in the example is used here to refer to thermodynamics.

*(now L820) The evaporation of organic compounds typically occurs after some sort of chemical reaction, but there is no reason that the evaporation of organic compounds cannot be included within "aging processes".*

The 'Outcomes' section could benefit from the following: (i) discussion of the results in the context of earlier work and how the findings here are similar or different, (ii) how the SAFIRED measurements were insightful (iii) what questions still remain unanswered, and (iv) directions for future work.

Similar to comment 5, a Table listing the companion publications and its central finding would be helpful for the interested reader to track the measurement-specific paper.

*The "Outcomes" section has been significantly altered. A summary table listing the companion publications has been included at the beginning of this section. A lot of "background"-like text has been removed so that more of a focus has been placed on the all of the outcomes and future work.*

Line 658-677: Will the NMOC emissions and speciation be discussed in a forthcoming publication? I did not see a SAFIRED-related reference for this section.

*(L1002) A manuscript containing the NMOC data and discussion has not been finalised at the time of this response and submission of this version of the manuscript. However, this data is currently being examined and will be published in the future. This has been indicated.*

Line 762-763: How does primary organic aerosol interact with NMOCs to form SOA? I am not sure this sentence is phrased correctly. Do you mean primary organic aerosol serves as a seed for the SOA produced from NMOC oxidation?

*This sentence and section has been removed from the manuscript. The referee was correct in their interpretation.*

Clarification question: Were aircrafts used to study the biomass burning plumes?

While I understand that the majority of the companion papers that deal with the specifics of each measurement are in the process of being prepared or are currently under review, are there any novel campaign-wide conclusions that the authors would like to discuss in the concluding section of the manuscript?

*(L1217) The "Looking forward" section has been changed to "Conclusions and looking forward". A paragraph summarising the campaign-wide conclusions have been included at the beginning of this section. Aircrafts were not used in this study. Several recommendations have been amended to the end of this section, including taking in situ measurements at the surface and throughout the boundary layer.*

**Anonymous Referee #3**

This paper gives an overview of the SAFIRED2014 campaign in Northern Australia aimed at investigating biomass burning in an area that has very frequent burning but is clearly understudied. This paper suffers from the typical issues of overview papers, where there is a long introduction of instruments and methods, but no actual results. In this paper especially the last section "Outcomes of SAFIRED" is very long, includes short literature reviews, but teases at potential results and points to other related pa- pers without giving any results. Overview papers clearly serve a purpose and should include four major points: 1) description of the science goals and how the campaign was designed to answer them, 2) a systematic description of the used instrumenta- tion, 3) a big picture overview of the results and 4) a conclusion of how the campaign results are usable for answering the science question. This overview paper here de- scribes most of the above points, but could benefit from some improvements and in particular would benefit from summarizing the results more systematically.

*The authors thank the referee for their comments and suggestions. The manuscript has been significantly altered in order to avoid the problems that the referee finds common in overview papers. More emphasis has been placed on the results and introductory text has been removed or shortened, especially within the "Outcomes of SAFIRED" section.*

Specific Comments:

I think it would be helpful to actually list the specific science questions at the end of the introduction or in a new section before the instrument descriptions.

*(L164) The specific science questions have been summarised at the end of the Introduction section.*

Instruments and Measurements

The chapter 2.2 Instruments and Measurements should be made more consistent be- tween the individual instrument descriptions and also misses some critical information. Most of the instrument detection methods are described well, but the most important information for all the measurements are missing. For each instrument description the following needs to be added: sensitivity (precision and accuracy), limit of detection, time resolution and used inlet. A table should be added that lists all of these instrument parameters and also a reference to the technique. The Radon instrument description also includes a summary of how Radon measurements are used in atmospheric research. This is not appropriate here and should be moved to the results section around page 21.

*(L196 and Table 1) A table has been added to this section, giving a summary of the Instruments and Measurements (quantity, instrument, time resolution, reference;*

*other details are discussed in relevant companion or referenced studies). The summary of the use of Radon measurements in atmospheric research has been mostly removed or moved to later sections.*

The chapter Aerosols should be numbered consistently with 2.2.2 Fires and Air Masses

*(L360) This has been fixed.*

What I was mostly missing in this chapter was putting SAFIRED into the bigger picture of fire emissions in Australia, e.g.: how representative is SAFIRED, was this a typical year and what could SAFIRED potentially tell us about emission estimates in northern Australia. How many fires did you observe during SAFIRED? How many of those measured plumes were fresh (for emission ratios) and how many were aged?

*With a more concise manuscript, the focus is now on the bigger picture of fire emissions in Australia. Without long term in situ measurements it is difficult to conclude whether the 2014 early dry season was atypical or not. This has been discussed in the "Conclusions and Looking Forward" section.*

*Given the high frequency of fires across the regions and the mixing and differing trajectories of smoke plumes, it is difficult to attribute the constantly elevated signals to individual (fresh or aged) plumes. Nonetheless, a lot of the discussed of the spikes in the gaseous and aerosol species in the "Result" section is devoted to trying to link the measured emissions to fires. Furthermore, the two companion papers, Desservattaz et al., 2017 and Milic et al., 2016, provide in-depth investigations of the identification of individual plumes, emission factors and the atmospheric aging processes of aerosol during SAFIRED.*

Figure 4: The data here are split into weak moderate and strong mixing, but nothing is really done with this separation later. Also the differences are not very strong. In the next Figure and the rest of the manuscript the data get separated into different BB and costal periods. This seems a better separation. I suggest removing the mixing categories. I am also wondering how the wind direction plot looks for the Coastal Period. This would be more helpful for a separation.

*(now Figures 5 and 7) The data has been split into the BBPs and CP rather than "mixing" category. Diurnal trends of radon, wind speed, wind direction, temperature and select VOCs for each period have been displayed. The same has been done for the greenhouse gases and aerosol species and size and is displayed in the supplementary material.*

Figure 4c y-axis should go from 0-360.

*(now Figures 5 and 6) Because the wind direction never was between 200° and 360°, the axis has been kept from 0° to 200° so that variations are more easily distinguished.*

page 24 line 473: What are the criteria used to separate the data into these periods? The separation seems very arbitrary to me, especially what is the difference between BBP2 and BBP3. Also the coastal period has large CO mixing ratios and very similar O3/CO ratios as BBP3. Please explain in more detail what is difference between the periods and how you define BBP. Are these by CO or acetonitrile enhancements, back trajectories, or fire counts? diurnal trend e.g.:

*(now L489) There was no strict criteria in separating the data into the 5 different periods. A combination of the daily satellite observed fires and the meteorological, gaseous and aerosol measurements were used to distinguish periods with BB and marine influence. Furthermore, the BB periods were selected as full days and the CP was selected as 1.5 days exactly to provide further insight into the diurnal variations*

page24 line 470-471: The authors argue here and in other places that the diurnal variations are caused by the mixing height. This is probably right, but no actual evidence is presented. The wind direction changes as can be seen clearly in Figure 4. Looking only at the time series in Figures 5 and 6 one cannot judge, if the diurnal changes align with wind direction change or more with the Radon profile. A diurnal profile of some trace gases and aerosol species should be added. I would also like to see that separated for the different BBP and CP.

*(now L546) It now reads that mixing height, wind velocity, fire locations and the time of fires are the cause of the diurnal variations. Diurnal variations have been separated into the BBPs and CP.*

Figures 5 and 8. It would be good to also show the CO data on a linear scale.

*(now Figures 3 and 7) The CO data has now been presented on a linear scale.*

Close Proximity Fires versus Aged Fires

On several places on pages 27-30 the age of fire plumes are discussed in rather vague terms sometimes using organic aerosol or size distributions as chemical indica- tor in addition to the fire locations. To show photochemical aging the most commonly used way is to look at ratios of a short lived tracer to an inert tracer on the time scale of the transport. Ratios of some of the VOC measurements versus CO or acetoni- trile would be best used to show aging, most commonly used are aromatic species, benzene for longer time scales, toluene or larger aromatics for shorter time scales. Enhancement ratios of fresh fires seem be available from the "close proximity fires" or nighttime fire plumes, although I have my doubts about how close those fires were, as I will describe below. Fires in the region are relatively similar and the emission ra- tios should therefore also be similar enough to distinguish between fresh plumes and plumes transported over 200-300km to the site using VOC/CO ratios. I would suggest replacing all the vague discussions about plume age with adequate VOC/CO enhance- ment ratios.

O3/CO ratios: The O3/CO is used in Figures 5 and 8 and is described at giving an indication of photochemical age. Unfortunately O3/CO are much more complicated than that and depend on many different factors such as VOC/NOx ratios such that the ratio really cannot be used as "photochemical age". I think for this paper here, it is best to remove the O3/CO ratios instead of adding a proper explanation.

The ozone enhancement shown in Figure 8 for the close proximity fire is substan- tial and ozone values of almost 100ppb are detected in the plume. This means that there has been significant photochemical processing of potentially several hours dur- ing plume transport. If the plume would be really fresh, ozone would actually be titrated. Again VOC/CO could be very helpful here and should be looked into. Also a compar- ison to a nighttime plume measurement would be very useful. Again, I doubt that this plume is very fresh.

*The ozone spike during BBP4 was likely a result of cross contamination within the ozone analyzer from other UV-absorbing species. High concentrations, the time of evening and a low f44 value from the AMS indicate very fresh smoke and this is supported by observations of the burned area. O3/CO ratios have been removed from the manuscript and all discussion and data has been replaced with acetonitrile, toluene and their ratio (Figure 7).*

Outcomes of SAFIRED

The paper is rather long in its current form and in particular this chapter is more of a literature review, of what could potentially be done with the specific measurement. I actually think this is not appropriate for an overview paper and would be more appropri- ately discussed in the detailed follow-up papers. I suggest deleting this whole section and just briefly mentioning the potential major outcomes in the "Looking Forward" section.

*The Outcomes of SAFIRED section has been shortened and does no longer include "background" text which, as the referee points out, is not appropriate for this section. This section now gives a brief overview of the campaign papers and the overall results from the study.*

The picture quality of all Figures needs to be improved.

*The resolution and file format has been improved and font sizes have been increased.*

[revised manuscript text omitted]

Centered, Position:Horizontal: Left, Relative to: Column, Vertical:  0", Relative to: Paragraph, Horizontal:  0.13", Wrap Around

| Page 9: [2] Formatted | Marc Mallet | 5/15/17 12:52:00 AM |
|---|---|---|

Centered, Position:Horizontal: Left, Relative to: Column, Vertical:  0", Relative to: Paragraph, Horizontal:  0.13", Wrap Around

| Page 9: [3] Formatted | Marc Mallet | 5/15/17 12:52:00 AM |
|---|---|---|

Centered, Position:Horizontal: Left, Relative to: Column, Vertical:  0", Relative to: Paragraph, Horizontal:  0.13", Wrap Around

| Page 9: [4] Formatted | Marc Mallet | 5/15/17 12:52:00 AM |
|---|---|---|

Centered, Position:Horizontal: Left, Relative to: Column, Vertical:  0", Relative to: Paragraph, Horizontal:  0.13", Wrap Around

| Page 9: [5] Formatted | Marc Mallet | 5/15/17 12:52:00 AM |
|---|---|---|

Centered, Position:Horizontal: Left, Relative to: Column, Vertical:  0", Relative to: Paragraph, Horizontal:  0.13", Wrap Around

| Page 14: [6] Deleted | Marc Mallet | 5/1/17 10:12:00 PM |
|---|---|---|

**Radon-222 (radon) is a naturally occurring radioactive noble gas that arises from the alpha-particle decay of radium-226, which is ubiquitous in most soil and rock types. With a half-life of 3.82 days, radon has thus proven to be an excellent indicator of recent (within 2-3 weeks) terrestrial influences on air masses for observations at coastal or island sites (Chambers et al., 2014). Radon is unreactive and poorly soluble, and its only atmospheric sink is radioactive decay. Furthermore, it has a half-life comparable to the lifetimes of short-lived atmospheric pollutants (e.g., NOx, SO$_2$) and atmospheric residence time of water and**

aerosols. Radon's unique combination of physical characteristics make it an ideal tracer for (i) regional air mass transport studies, in which it is often used in conjunction with air mass back trajectories for fetch analyses (Williams et al., 2009;Chambers et al., 2014); (ii) investigations of vertical mixing processes within the daytime convective boundary layer (Williams et al., 2011) or nocturnal boundary layer (Chambers et al., 2015); (iii) identifying periods of minimal terrestrial influence on a measured air mass ("baseline" studies; (Chambers et al., 2016)); (iv) performing regional flux or inventory analyses for trace atmospheric species with similar source distributions (Biraud et al., 2000) and (v) evaluating the performance of transport and mixing schemes in climate and chemical-transport models (Locatelli et al., 2015).

| Page 24: [7] Deleted | Marc Mallet | 5/2/17 1:32:00 AM |
|---|---|---|

Afternoon radon concentrations provide further information regarding the regional air mass fetch and the degree of contact with the land surface (red line, Figure 3a). Over the campaign period, air masses with the least terrestrial fetch (low radon indicates strongest oceanic signature) were observed on June 4-6 and 20-22, whereas June 8-9, 17-18 and 29-30 represented periods of particularly extensive continental fetch.

| Page 26: [8] Deleted | Marc Mallet | 5/2/17 1:31:00 AM |
|---|---|---|

[Figure]

Figure **3** Hourly ATARS radon observations for June 2014: (a) observed hourly data, and afternoon-to-afternoon interpolated values (indicative of changes in the regional air mass fetch); and (b) difference in radon concentration between the hourly observations and interpolated afternoon values (indicative of diurnal variability).

A pronounced diurnal variability can clearly be seen in the ΔRn signal (Figure 3b). Mean hourly diurnal composites of radon concentrations, wind speed, wind direction and dew point temperature at the ATARS site during the period of the SAFIRED campaign are shown in Figure 4. Following the technique described in Chambers et al. (2016), these composites have been computed separately for three diurnal mixing categories based on the mean ΔRn over the 12 hour period 2000-0800 h:

Strong mixing: $\qquad$ $\Delta Rn_{12} < 5400$ mBq m$^{-3}$

Moderate mixing: $\qquad$ $5400 \leq \Delta Rn_{12} < 6700$ mBq m$^{-3}$

Weak mixing: $\qquad$ $\Delta Rn_{12} \geq 6700$ mBq m$^{-3}$

The air masses predominantly originated from the southeast as indicated in Figure 1 and Figure 4c. Starting from approximately 10:00 am each morning, however, sea breeze circulations slowly turn the measured wind direction around from southeast to northeast, before reverting back to the dominant wind direction again at around midnight. Wind speeds reached a maximum just before midday and were at their lowest just before midnight (Figure 4b). The "strong mixing" category was associated with generally higher wind speeds, which cause increased mechanical turbulence leading to deeper nocturnal mixing layers (i.e., hinder the development of a shallow nocturnal inversion layer).

| Page 31: [9] Deleted | Marc Mallet | 5/14/17 10:38:00 PM |

Gas and Aerosol measurements

The campaign average, standard deviation, median and Q25/Q75 values for the major gaseous and aerosol species are shown in Table 1. The median values for each species are likely to be representative of background concentrations in this region. The average concentrations for most species were higher than the median concentrations, due to the periods of close or intense fires. The extent of the influence of these close fires are demonstrated by the maximum concentrations.

Table 1 The campaign average, standard deviation, maximum, median, Q25 and Q75 values for key measured gas and aerosol species. All parts-per notation refer to mole fractions unless otherwise indicated.

| Species (unit) | Average | Standard deviation | Maximum | Median | Q25 | Q75 |
|---|---|---|---|---|---|---|
| CO (ppb) | 229 | 494 | 18900 | 130 | 87 | 214 |
| $CO_2$ (ppm) | 404.68 | 11.539 | 513.578 | 402.454 | 394.728 | 411.299 |
| $O_3$ (ppbv) | 24.616 | 9.903 | 99.784 | 22.771 | 17.896 | 29.778 |
| $CH_4$ (ppb) | 1839.88 | 68.06 | 3766.81 | 1820.11 | 1802.26 | 1852.97 |
| $N_2O$ (ppb) | 326.329 | 0.449 | 334.871 | 326.276 | 326.121 | 326.444 |
| GEM (ng m$^{-3}$) | 0.992 | 0.081 | 1.734 | 0.986 | 0.952 | 1.020 |
| Acetonitrile (ppb) | 0.351 | 0.629 | 9.775 | 0.197 | 0.129 | 0.337 |

| | | | | | | |
|---|---|---|---|---|---|---|
| Organics (ug m$^{-3}$) | 11.081 | 22.385 | 347.657 | 4.160 | 2.335 | 13.279 |
| SO$_4^{2-}$ (ug m$^{-3}$) | 0.514 | 0.318 | 2.254 | 0.411 | 0.294 | 0.679 |
| NH$_4^+$ (ug m$^{-3}$) | 0.351 | 0.676 | 18.17 | 0.180 | 0.096 | 0.415 |
| NO$_3^-$ (ug m$^{-3}$) | 0.187 | 0.456 | 10.925 | 0.042 | 0.004 | 0.189 |
| Cl$^-$ (ug m$^{-3}$) | 0.166 | 1.271 | 53.270 | 0.029 | 0.016 | 0.076 |
| PNC (cm$^{-3}$) | 8182 | 19031 | 40300 | 2032 | 2032 | 8335 |
| Mode diameter (nm) | 104 | 31 | - | 102 | 85 | 122 |
| Geom. SD | 1.71 | 0.13 | - | 1.70 | 1.65 | 1.75 |

In order to demonstrate the influence of close fires and the changing inversion layer, the time series of major greenhouse gases (CO, $CO_2$, $CH_4$ and $N_2O$), gaseous elemental mercury, acetonitrile and ozone throughout the campaign are shown in Figure 5. Sub-micron non-refractory aerosol organic, sulfate, ammonium and nitrate mass concentrations, organic mass fraction, $PM_1$ OC and EC mass concentrations and particle size distributions for the sampling period are shown in Figure 6. Periods of missing data correspond to times when instruments were not operating. Most of these time series display a clear diurnal trend as a result of the varying inversion layer height. Other enhancements in concentrations can be clearly seen and correspond to periods of frequent close fires (Figure 2). Over the entire sampling period from the 29th of May 2014 until the 28th of June 2014, four biomass burning related periods (BBP) and a "coastal" period (CP) have been distinguished. The dates for these periods are shown in Table 2. These periods are also displayed in Figure 5 and Figure 6.

Table 2 The start and end dates for the four identified Biomass Burning Periods (BBP1, BBP2, BBP3 and BBP4) and the Coastal Period (CP).

| Period | Start date (mm/dd/yy hh:mm) | End date (mm/dd/yy hh:mm) |
|---|---|---|
| BBP1 | 05/30/14 00:00 | 05/31/14 23:59 |
| BBP2 | 06/06/14 00:00 | 06/12/14 23:59 |
| BBP3 | 06/14/14 00:00 | 06/17/14 23:59 |
| CP | 06/19/14 12:00 | 06/22/14 23:59 |
| BBP3 | 06/23/14 00:00 | 06/28/14 23:59 |

[Figure]

Figure **5** The time series of the major measured gaseous species during the SAFIRED campaign: (a) carbon monoxide, (b) carbon dioxide, (c) methane, (d) nitrogen dioxide, (e) gaseous elemental mercury, (f) acetonitrile and (g) ozone and $\Delta O_3/\Delta CO$. The biomass burning and coastal periods are indicated by the red dotted lines. All parts-per notation refer to mole fractions unless otherwise indicated.

[Figure]

Figure **6** The times series of the major aerosol properties during the SAFIRED campaign: (a) the non-refractory PM$_1$ organic mass concentration (left) and organic mass fraction (right), b) the inorganic non-refractory PM$_1$ mass concentrations, (c) the 12-hour filter OC and EC PM$_1$ mass concentrations (left) and the ratio of OC to OC+EC (right), (d) the particle size distributions and particle size mode (left) and the total particle number concentration (right) and (e) the wind direction at ATARS.

Over the campaign organics dominated the non-refectory sub-micron aerosol mass contributing, on average 90% (median; 86%) of the total mass. Sulphate, nitrates, ammonium and chloride species contributed the rest of this mass, with the largest contributions from sulphate and ammonium. Sulphate contributions were very significant during the coastal period, contributing up to 32% of the total mass. Although chlorides contributed the least to the total mass, on average, during clear biomass burning events where sharp increases in CO and organics were observed, chlorides made up the largest component of inorganic aerosol. The maximum chloride concentration during the campaign reached 53 $\mu g\ m^{-3}$. High soil and vegetation chloride contents have been observed in savannah and coastal environments (Lobert et al., 1999;Andreae et al., 1996). The strong elevations of chloride signals observed, particularly during burning events BBP1, BBP2 and BBP4 are likely a result of emission of these chloride ions. Outside of the "burning events" where very sharp increases in concentrations were observed, chloride concentrations were very low. This either suggests that these chloride species are short-lived, or only present in fires very close to the coast and therefore the ATARS site.

| Page 31: [10] Formatted | Marc Mallet | 11/8/16 11:22:00 PM |
|---|---|---|

Font:Not Bold, Font color: Text 1

| Page 31: [10] Formatted | Marc Mallet | 11/8/16 11:22:00 PM |
|---|---|---|

Font:Not Bold, Font color: Text 1

| Page 31: [10] Formatted | Marc Mallet | 11/8/16 11:22:00 PM |
|---|---|---|

Font:Not Bold, Font color: Text 1

| Page 31: [10] Formatted | Marc Mallet | 11/8/16 11:22:00 PM |
|---|---|---|

Font:Not Bold, Font color: Text 1

| Page 31: [10] Formatted | Marc Mallet | 11/8/16 11:22:00 PM |
|---|---|---|

Font:Not Bold, Font color: Text 1

| Page 31: [11] Formatted | Marc Mallet | 11/8/16 11:22:00 PM |
|---|---|---|

Font:10 pt, Font color: Text 1

| Page 31: [11] Formatted | Marc Mallet | 11/8/16 11:22:00 PM |
|---|---|---|

Font:10 pt, Font color: Text 1

| Page 31: [11] Formatted | Marc Mallet | 11/8/16 11:22:00 PM |
|---|---|---|

Font:10 pt, Font color: Text 1

| Page 31: [11] Formatted | Marc Mallet | 11/8/16 11:22:00 PM |
|---|---|---|

Font:10 pt, Font color: Text 1

| Page 31: [11] Formatted | Marc Mallet | 11/8/16 11:22:00 PM |
|---|---|---|

Font:10 pt, Font color: Text 1

| Page 31: [11] Formatted | Marc Mallet | 11/8/16 11:22:00 PM |
|---|---|---|

Font:10 pt, Font color: Text 1

| Page 31: [11] Formatted | Marc Mallet | 11/8/16 11:22:00 PM |
|---|---|---|

Font:10 pt, Font color: Text 1

| Page 31: [11] Formatted | Marc Mallet | 11/8/16 11:22:00 PM |
|---|---|---|

Font:10 pt, Font color: Text 1

| Page 31: [11] Formatted | Marc Mallet | 11/8/16 11:22:00 PM |
|---|---|---|

Font:10 pt, Font color: Text 1

| Page 31: [11] Formatted | Marc Mallet | 11/8/16 11:22:00 PM |
|---|---|---|

Font:10 pt, Font color: Text 1

| Page 31: [11] Formatted | Marc Mallet | 11/8/16 11:22:00 PM |
|---|---|---|

Font:10 pt, Font color: Text 1

| Page 31: [11] Formatted | Marc Mallet | 11/8/16 11:22:00 PM |
|---|---|---|

Font:10 pt, Font color: Text 1

| Page 31: [11] Formatted | Marc Mallet | 11/8/16 11:22:00 PM |
|---|---|---|

Font:10 pt, Font color: Text 1

| Page 31: [11] Formatted | Marc Mallet | 11/8/16 11:22:00 PM |
|---|---|---|

Font:10 pt, Font color: Text 1

| Page 31: [11] Formatted | Marc Mallet | 11/8/16 11:22:00 PM |

Font:10 pt, Font color: Text 1

| Page 31: [11] Formatted | Marc Mallet | 11/8/16 11:22:00 PM |

Font:10 pt, Font color: Text 1

| Page 31: [11] Formatted | Marc Mallet | 11/8/16 11:22:00 PM |

Font:10 pt, Font color: Text 1

| Page 31: [11] Formatted | Marc Mallet | 11/8/16 11:22:00 PM |

Font:10 pt, Font color: Text 1

| Page 31: [11] Formatted | Marc Mallet | 11/8/16 11:22:00 PM |

Font:10 pt, Font color: Text 1

| Page 31: [11] Formatted | Marc Mallet | 11/8/16 11:22:00 PM |

Font:10 pt, Font color: Text 1

| Page 31: [11] Formatted | Marc Mallet | 11/8/16 11:22:00 PM |

Font:10 pt, Font color: Text 1

| Page 31: [11] Formatted | Marc Mallet | 11/8/16 11:22:00 PM |

Font:10 pt, Font color: Text 1

| Page 31: [11] Formatted | Marc Mallet | 11/8/16 11:22:00 PM |

Font:10 pt, Font color: Text 1

| Page 31: [11] Formatted | Marc Mallet | 11/8/16 11:22:00 PM |

Font:10 pt, Font color: Text 1

| Page 31: [11] Formatted | Marc Mallet | 11/8/16 11:22:00 PM |

Font:10 pt, Font color: Text 1

| Page 31: [11] Formatted | Marc Mallet | 11/8/16 11:22:00 PM |

Font:10 pt, Font color: Text 1

| Page 31: [11] Formatted | Marc Mallet | 11/8/16 11:22:00 PM |
|---|---|---|

Font:10 pt, Font color: Text 1

| Page 31: [11] Formatted | Marc Mallet | 11/8/16 11:22:00 PM |
|---|---|---|

Font:10 pt, Font color: Text 1

| Page 31: [11] Formatted | Marc Mallet | 11/8/16 11:22:00 PM |
|---|---|---|

Font:10 pt, Font color: Text 1

| Page 31: [11] Formatted | Marc Mallet | 11/8/16 11:22:00 PM |
|---|---|---|

Font:10 pt, Font color: Text 1

| Page 31: [11] Formatted | Marc Mallet | 11/8/16 11:22:00 PM |
|---|---|---|

Font:10 pt, Font color: Text 1

| Page 31: [11] Formatted | Marc Mallet | 11/8/16 11:22:00 PM |
|---|---|---|

Font:10 pt, Font color: Text 1

| Page 31: [11] Formatted | Marc Mallet | 11/8/16 11:22:00 PM |
|---|---|---|

Font:10 pt, Font color: Text 1

| Page 31: [11] Formatted | Marc Mallet | 11/8/16 11:22:00 PM |
|---|---|---|

Font:10 pt, Font color: Text 1

| Page 31: [11] Formatted | Marc Mallet | 11/8/16 11:22:00 PM |
|---|---|---|

Font:10 pt, Font color: Text 1

| Page 31: [11] Formatted | Marc Mallet | 11/8/16 11:22:00 PM |
|---|---|---|

Font:10 pt, Font color: Text 1

| Page 31: [11] Formatted | Marc Mallet | 11/8/16 11:22:00 PM |
|---|---|---|

Font:10 pt, Font color: Text 1

| Page 31: [11] Formatted | Marc Mallet | 11/8/16 11:22:00 PM |
|---|---|---|

Font:10 pt, Font color: Text 1

| Page 31: [11] Formatted | Marc Mallet | 11/8/16 11:22:00 PM |
|---|---|---|

Font:10 pt, Font color: Text 1

| Page 31: [11] Formatted | Marc Mallet | 11/8/16 11:22:00 PM |
|---|---|---|

Font:10 pt, Font color: Text 1

| Page 31: [11] Formatted | Marc Mallet | 11/8/16 11:22:00 PM |
|---|---|---|

Font:10 pt, Font color: Text 1

| Page 31: [11] Formatted | Marc Mallet | 11/8/16 11:22:00 PM |
|---|---|---|

Font:10 pt, Font color: Text 1

| Page 31: [11] Formatted | Marc Mallet | 11/8/16 11:22:00 PM |
|---|---|---|

Font:10 pt, Font color: Text 1

| Page 31: [11] Formatted | Marc Mallet | 11/8/16 11:22:00 PM |
|---|---|---|

Font:10 pt, Font color: Text 1

| Page 31: [11] Formatted | Marc Mallet | 11/8/16 11:22:00 PM |
|---|---|---|

Font:10 pt, Font color: Text 1

| Page 31: [11] Formatted | Marc Mallet | 11/8/16 11:22:00 PM |
|---|---|---|

Font:10 pt, Font color: Text 1

| Page 31: [11] Formatted | Marc Mallet | 11/8/16 11:22:00 PM |
|---|---|---|

Font:10 pt, Font color: Text 1

| Page 31: [11] Formatted | Marc Mallet | 11/8/16 11:22:00 PM |
|---|---|---|

Font:10 pt, Font color: Text 1

| Page 31: [11] Formatted | Marc Mallet | 11/8/16 11:22:00 PM |
|---|---|---|

Font:10 pt, Font color: Text 1

| Page 31: [11] Formatted | Marc Mallet | 11/8/16 11:22:00 PM |
|---|---|---|

Font:10 pt, Font color: Text 1

| Page 31: [11] Formatted | Marc Mallet | 11/8/16 11:22:00 PM |
|---|---|---|

Font:10 pt, Font color: Text 1

| Page 31: [11] Formatted | Marc Mallet | 11/8/16 11:22:00 PM |

Font:10 pt, Font color: Text 1

| Page 31: [11] Formatted | Marc Mallet | 11/8/16 11:22:00 PM |

Font:10 pt, Font color: Text 1

| Page 31: [11] Formatted | Marc Mallet | 11/8/16 11:22:00 PM |

Font:10 pt, Font color: Text 1

| Page 31: [11] Formatted | Marc Mallet | 11/8/16 11:22:00 PM |

Font:10 pt, Font color: Text 1

| Page 31: [11] Formatted | Marc Mallet | 11/8/16 11:22:00 PM |

Font:10 pt, Font color: Text 1

| Page 31: [11] Formatted | Marc Mallet | 11/8/16 11:22:00 PM |

Font:10 pt, Font color: Text 1

| Page 31: [11] Formatted | Marc Mallet | 11/8/16 11:22:00 PM |

Font:10 pt, Font color: Text 1

| Page 31: [11] Formatted | Marc Mallet | 11/8/16 11:22:00 PM |

Font:10 pt, Font color: Text 1

| Page 31: [11] Formatted | Marc Mallet | 11/8/16 11:22:00 PM |

Font:10 pt, Font color: Text 1

| Page 31: [11] Formatted | Marc Mallet | 11/8/16 11:22:00 PM |

Font:10 pt, Font color: Text 1

| Page 31: [11] Formatted | Marc Mallet | 11/8/16 11:22:00 PM |

Font:10 pt, Font color: Text 1

| Page 31: [11] Formatted | Marc Mallet | 11/8/16 11:22:00 PM |

Font:10 pt, Font color: Text 1

| Page 31: [11] Formatted | Marc Mallet | 11/8/16 11:22:00 PM |
|---|---|---|

Font:10 pt, Font color: Text 1

| Page 31: [11] Formatted | Marc Mallet | 11/8/16 11:22:00 PM |
|---|---|---|

Font:10 pt, Font color: Text 1

| Page 31: [11] Formatted | Marc Mallet | 11/8/16 11:22:00 PM |
|---|---|---|

Font:10 pt, Font color: Text 1

| Page 31: [11] Formatted | Marc Mallet | 11/8/16 11:22:00 PM |
|---|---|---|

Font:10 pt, Font color: Text 1

| Page 31: [11] Formatted | Marc Mallet | 11/8/16 11:22:00 PM |
|---|---|---|

Font:10 pt, Font color: Text 1

| Page 31: [11] Formatted | Marc Mallet | 11/8/16 11:22:00 PM |
|---|---|---|

Font:10 pt, Font color: Text 1

| Page 31: [11] Formatted | Marc Mallet | 11/8/16 11:22:00 PM |
|---|---|---|

Font:10 pt, Font color: Text 1

| Page 31: [11] Formatted | Marc Mallet | 11/8/16 11:22:00 PM |
|---|---|---|

Font:10 pt, Font color: Text 1

| Page 31: [11] Formatted | Marc Mallet | 11/8/16 11:22:00 PM |
|---|---|---|

Font:10 pt, Font color: Text 1

| Page 31: [11] Formatted | Marc Mallet | 11/8/16 11:22:00 PM |
|---|---|---|

Font:10 pt, Font color: Text 1

| Page 31: [11] Formatted | Marc Mallet | 11/8/16 11:22:00 PM |
|---|---|---|

Font:10 pt, Font color: Text 1

| Page 31: [11] Formatted | Marc Mallet | 11/8/16 11:22:00 PM |
|---|---|---|

Font:10 pt, Font color: Text 1

| Page 31: [11] Formatted | Marc Mallet | 11/8/16 11:22:00 PM |
|---|---|---|

Font:10 pt, Font color: Text 1

| Page 31: [11] Formatted | Marc Mallet | 11/8/16 11:22:00 PM |
|---|---|---|

Font:10 pt, Font color: Text 1

| Page 31: [11] Formatted | Marc Mallet | 11/8/16 11:22:00 PM |
|---|---|---|

Font:10 pt, Font color: Text 1

| Page 31: [11] Formatted | Marc Mallet | 11/8/16 11:22:00 PM |
|---|---|---|

Font:10 pt, Font color: Text 1

| Page 31: [12] Formatted | Marc Mallet | 11/8/16 11:22:00 PM |
|---|---|---|

Font color: Text 1

| Page 31: [12] Formatted | Marc Mallet | 11/8/16 11:22:00 PM |
|---|---|---|

Font color: Text 1

| Page 31: [12] Formatted | Marc Mallet | 11/8/16 11:22:00 PM |
|---|---|---|

Font color: Text 1

| Page 31: [12] Formatted | Marc Mallet | 11/8/16 11:22:00 PM |
|---|---|---|

Font color: Text 1

| Page 31: [12] Formatted | Marc Mallet | 11/8/16 11:22:00 PM |
|---|---|---|

Font color: Text 1

| Page 31: [13] Formatted | Marc Mallet | 11/8/16 11:22:00 PM |
|---|---|---|

Font color: Text 1

| Page 31: [13] Formatted | Marc Mallet | 11/8/16 11:22:00 PM |
|---|---|---|

Font color: Text 1

| Page 31: [13] Formatted | Marc Mallet | 11/8/16 11:22:00 PM |
|---|---|---|

Font color: Text 1

| Page 31: [13] Formatted | Marc Mallet | 11/8/16 11:22:00 PM |
|---|---|---|

Font color: Text 1

| Page 31: [14] Moved to page 23 (Move #1) | Marc Mallet | 5/2/17 1:32:00 AM |
|---|---|---|

[Figure]

Figure **5** The time series of the major measured gaseous species during the SAFIRED campaign: (a) carbon monoxide, (b) carbon dioxide, (c) methane, (d) nitrogen dioxide, (e) gaseous elemental mercury, (f) acetonitrile and (g) ozone and $\Delta O_3/\Delta CO$. The biomass burning and coastal periods are indicated by the red dotted lines. All parts-per notation refer to mole fractions unless otherwise indicated.

| Page 31: [15] Formatted | Marc Mallet | 11/8/16 11:22:00 PM |
|---|---|---|

Font color: Text 1

| Page 31: [15] Formatted | Marc Mallet | 11/8/16 11:22:00 PM |
|---|---|---|

Font color: Text 1

| Page 31: [16] Moved to page 24 (Move #2) | Marc Mallet | 5/2/17 1:32:00 AM |
|---|---|---|

[Figure]

Figure 6 The times series of the major aerosol properties during the SAFIRED campaign: (a) the non-refractory PM$_1$ organic mass concentration (left) and organic mass fraction (right), b) the inorganic non-refractory PM$_1$ mass concentrations, (c) the 12-hour filter OC and EC PM$_1$ mass concentrations (left) and the ratio of OC to OC+EC (right), (d) the particle size distributions and particle size mode (left) and the total particle number concentration (right) and (e) the wind direction at ATARS.

| Page 31: [17] Formatted | Marc Mallet | 11/8/16 11:22:00 PM |
|---|---|---|

Font color: Text 1

| Page 31: [17] Formatted | Marc Mallet | 11/8/16 11:22:00 PM |
|---|---|---|

Font color: Text 1

| Page 31: [17] Formatted | Marc Mallet | 11/8/16 11:22:00 PM |
| --- | --- | --- |

Font color: Text 1

| Page 31: [17] Formatted | Marc Mallet | 11/8/16 11:22:00 PM |
| --- | --- | --- |

Font color: Text 1

| Page 31: [17] Formatted | Marc Mallet | 11/8/16 11:22:00 PM |
| --- | --- | --- |

Font color: Text 1

| Page 31: [17] Formatted | Marc Mallet | 11/8/16 11:22:00 PM |
| --- | --- | --- |

Font color: Text 1

| Page 31: [17] Formatted | Marc Mallet | 11/8/16 11:22:00 PM |
| --- | --- | --- |

Font color: Text 1

| Page 31: [17] Formatted | Marc Mallet | 11/8/16 11:22:00 PM |
| --- | --- | --- |

Font color: Text 1

| Page 31: [18] Formatted | Marc Mallet | 11/8/16 11:22:00 PM |
| --- | --- | --- |

Font color: Text 1

| Page 31: [18] Formatted | Marc Mallet | 11/8/16 11:22:00 PM |
| --- | --- | --- |

Font color: Text 1

| Page 32: [19] Deleted | Marc Mallet | 5/14/17 10:42:00 PM |
| --- | --- | --- |

BBP1, BBP2 and BBP4 correspond to the periods when fires were burning within 10 km of ATARS. Large enhancements of biomass burning related emissions were observed during these three periods. There were distinct enhancements of all measured gaseous and aerosol species during these periods. Differences between the maximum and background concentrations were very prominent for CO (note the logarithmic scale in Figure 5a), $CH_4$, $N_2O$, acetonitrile (an established marker for biomass burning) and organic, nitrate and chloride non-refractory sub-micron aerosol species. Similar enhancements of $CH_4$ were also observed outside of these BB periods, which suggests another source of methane in this region. Only slight enhancements of GEM concentrations above background were observed during BBP2 and BBP4. Similar to much of the rest of the campaign sampling period, the non-refractory submicron aerosol was dominated by organics, with contributions typically varying between 70% and 95% of the mass. Relative to background concentrations, there were also large enhancements of nitrate and chloride species during these periods. While there were also enhancements of sulfate and ammonium species during these periods, similar enhancements were observed outside of these periods, again indicating a non-fire source of these species. The ratio of $O_3$ to CO concentrations above background (taken as 10 ppbv and 66 ppbv, respectively) gives an indication of the photochemical age of a smoke plume. $\Delta O_3/\Delta CO$ were lowest during BBP2 and BBP4 (and not measured during BBP1) relative to the rest of the campaign, indicating that the biomass burning signals during these periods had not undergone extensive photochemical aging and are therefore characteristic of fresh smoke.

Elevated signals during BBP1 were likely a result of a series of close fires within 5 km ENE of ATARS. The VIIRS and MODIS sensors on the SUMO NPP, Terra and Aqua satellites observes smaller fires at approximately 2 pm on the 30th of May. Winds were northeasterly during these two events. It is therefore likely that these signals were continuation or evolution of those fires. Burned vegetation was also visually observed the next morning at these locations. The large burst event later on the evening of the 31st of May is unlikely to be associated with these fires as the wind direction during this event was from the SSW and SSE. Large clusters of fires were observed at approximately 100 km and 150 km SE of the station by the Terra and Aqua satellites. The signals observed during this event could be a result of the plumes from this fire, although the possibility of a fire ignited after the satellite flyovers, or a combination of these cannot be eliminated.

Large signal enhancements on the 8th of June during BBP2 is likely a result of a cluster of fires approximately 100 km south east of the station. The MODIS sensors on the Terra and Aqua satellites observed the small cluster of fires along the back-trajectory at 11:14 am and 1:56 pm. The source of BB emissions for the large event on the 9th of June during this period is unclear. Several fires approximately 5 km from the station along the back-trajectory were detected by the MODIS sensor on Aqua and the VIIRS sensor on SUOMI NPP at approximately 2:30 pm on the 9th of June. There were also numerous fires detected between 100 km and 200 km southeast along this trajectory. The signals associated with this event could therefore be a result of the closer fires that started to blaze later in the evening, the distant fires or a combination of both.

Only one fire within 20km of ATARS was observed during BBP3 on the 17[th] of June. Numerous fires were observed further than 20km from the station and is possible that the signals during this period were more aged. While photochemical aging and coagulation typically lead to larger particles, particle size distributions were smaller during this period and the ratio of OC to OC+EC was 70%, 10% lower than the ratio during the rest of the campaign. Whether these observations were a result of burn conditions or aging processes

| Page 32: [20] Deleted | Marc Mallet | 5/14/17 10:42:00 PM |

(i.e. evaporation of organic compounds from the aerosol phase) is unclear, although the highest $\Delta O_3/\Delta CO$ values during the campaign were observed during BBP3, which indicates photochemical aging was more extensive during this period.

One close fire was also observed during CP, however wind directions during this period were typically north-easterly and concentrations were therefore much lower. 5-day HYSPLIT

trajectories also show that air mass during the CP originated along the east coast of Australia before travelling towards the sampling station with very little terrestrial influence.

Close proximity fires

With numerous fires occurring across the region and the limitations of once-per-day satellite fly-overs and stationary measurements, it can be difficult to identify the exact source of these elevated signals. For a portion of BBP4, however, fires were burning within several kilometers of ATARS and several plumes were easily observed from the station. The signals from these plumes are shown in Figure 8. The observed enhancements between 12:30 pm and 3:00 pm on the 25th June during BBP4 were a result of grass fires burning approximately 1 km south-east from the station. During this event, the wind direction (Figure 8k) was highly variable, changing between 140° and 80° True Bearing (TB) multiple times. As a result, the sampling changed from measuring the air mass with and without the plume from this fire, which led to sharp increases and decreases in biomass burning-related signals (Figure 8a through 8j). Visually, the fire area and extent of the plume was larger at 4:00pm than earlier, however the wind direction changed to north-easterly which directed the plume away from the station. From 4:00 pm until 10:00 pm, the wind direction was stable at approximately 50° TB. At 10:00 pm, the wind direction rapidly changed to directly south and the largest enhancements for the whole campaign were observed until approximately 2:00 am on the 26th of June. It is very likely that these signals were a result of a continuation and evolution of these fires as the night progressed. Portions of a ~0.25 $km^2$ grassland field within 500 m directly south of ATARS were observed to be burned upon arrival at the station on the morning of the 26th of June and we speculate that the burning of this field contributed to the large enhancements in measured biomass burning emissions. The emissions during this portion of BBP4

are likely to be the most representative of fresh biomass burning smoke during the SAFIRED campaign.

Page 32: [21] Moved to page 28 (Move #3)    Marc Mallet                    5/2/17 1:36:00 AM

[Figure]

**Figure 8 The major gas and aerosol concentrations measured during two biomass burning events within 1 km of ATARS during BBP4. (a) through (g) and (h) through (k) are as per Figures 5 and 6, respectively. All parts-per notation or mole fractions unless otherwise indicated.**

Page 35: [22] Deleted                        Marc Mallet                    5/14/17 10:49:00 PM

Australian fires are responsible for 6% of global $CO_2$ biomass burning emissions, most of which is due to savannah fires (Shi et al., 2015). Carbon sequestering during regrowth periods is considered to balance carbon emissions in tropical Australia (Haverd et al., 2013). Greenhouse gases emitted from savannah fires that are not sequestered, such as methane ($CH_4$) and nitrous oxide ($N_2O$), have been shown to contribute 2-4% of the annual accountable greenhouse gas emissions from Australia (Meyer et al., 2012). Seasonal emission factors for the major greenhouse gases are important for national greenhouse gas inventories and in understanding the impact of savannah fires. Furthermore, emission factors of $CO_2$ and CO can be used to infer mechanisms behind the emissions of other species, such as the connection between particulate matter and burning conditions.

**Page 35: [23] Deleted**        **Marc Mallet**        **5/14/17 10:50:00 PM**

The gaseous and aerosol data for the sample period were investigated to identify BB events and determine the emission factors of $CO_2$, CO, $CH_4$. $N_2O$, as well as Aitken and Accumulation mode aerosols and submicron particle species (organics, sulfates, nitrates, ammonium and chlorides) for several individual BB events. These emission factors were mostly found to be dependent on the combustion conditions (using the modified combustion efficiency as a proxy) of the fires. These results will be the first set of emission factors for aerosol particles from savannah fires in Australia. Furthermore, the variability in emission factors for different fires calls for a separation of single-value emission factors that are usually reported for savannah fires into grass and shrub components. A full discussion of these results are presented in Desservettaz et al. (2016, submitted).

Non-methane organic compounds (NMOCs)

Biomass burning is the second largest source of NMOCs globally with a recent global estimate of at least 400 Tg year$^{-1}$, second only to biogenic sources (Akagi et al., 2011). Biomass burning produces a complex mix of NMOCs, which may be saturated or unsaturated, aliphatic or aromatic, and contain substitutions of oxygen, sulfur, nitrogen, halogens and other atoms. NMOC emission rates are strongly tied to the efficiency of combustion, with smouldering fires emitting NMOC at higher rates than flaming fires (Andreae and Merlet, 2001). Biomass burning derived NMOCs fuel the production of tropospheric ozone in diluted, aged biomass burning plumes, with higher ozone enhancements observed when biomass burning plumes interact with NOx-rich urban plumes (Jaffe and Wigder, 2012;Wigder et al., 2013;Akagi et al., 2013). Oxidation of NMOCs results in lower volatility products that partition to the aerosol phase and contribute significantly to secondary organic aerosol (Hallquist et al., 2009). Biomass burning produces significant amounts of semi-volatile NMOC which can be difficult to quantify and identify with current measurement techniques. However recent studies have shown that including semi volatile NMOC chemistry in models improves the agreement between the modeled and observed organic aerosol (Alvarado et al., 2015; Konovalov et al., 2015) and ozone (Alvarado et al., 2015). High quality NMOC emission factors are crucial for models to assess the impact of biomass burning plumes on air quality and climate.

**PAHs**

Polycyclic aromatic hydrocarbons (PAHs) are a group of chemicals that are formed and emitted during combustion processes. Globally, major sources include residential/commercial biomass burning, open-field biomass burning and vehicular emissions (Shen et al., 2013). In Oceania in 2007, 31% of PAH emissions were estimated to be attributed to deforestation and wildfires (Shen et al., 2013). With control strategies targeting and reducing vehicular emission of PAHs over the last few decades, the relative contribution of other emission sources, such as savannah fires, has increased (Friedman et al., 2013;Kallenborn et al., 2012;Wang et al., 2016). Although most of these emissions are in the gas-phase (Jenkins et al., 1996;Atkins et al., 2010), the particle-phase PAHs, such as benzo[a]pyrene (BaP), may have high genotoxicity (IARC., 2015). However, field-based studies on emissions of PAHs from open-field biomass burning, including savannah fires remain limited in Australia (Freeman and Cattell, 1990).

Emission factors of PAHs from biomass burning related to savannah fires in northern Australia will be estimated form the data collected during this campaign. This estimation will be based on the (background subtracted) concentrations of PAHs and $CO_2$ (and CO) during the events where biomass burning contributes most to these concentrations measured at the sampling site. The concentrations of 13 major PAHs (gaseous plus particle-associated phase) varied from ~ 1 to over 15 ng m$^{-3}$ within different BB events. In the gas phase, 3- and 4-ring compounds typically contributed ~ 90% to the sum concentrations whereas the particle-associated PAHs were dominated by 5- and 6-ring compounds (> 80%). Measured PAH concentrations were significantly higher (paired $t$-test, $P < <0.05$) during BB events E, F and G. For these events, concentrations of BaP exceeded the monitoring investigation level for atmospheric BaP in Australia (National-Environment-Protection-Council-Service-Corporation, 2011), i.e. 0.30 ng m$^{-3}$, by 66% (BB event E) and 200% (BB events F and G). A full discussion of these results can be found in (Wang et al., 2016, under review).

Mercury

The atmosphere is the dominant transport pathway for mercury globally, with emissions to the atmosphere from both natural and anthropogenic origins (Driscoll et al., 2013). Whilst our understanding of the natural cycling of mercury has improved markedly over the past decades (Pirrone et al., 2010), large uncertainties still exist; specifically, global emission estimates to the atmosphere from biomass burning currently range between 300 and 600 Mg year$^{-1}$ (Driscoll et al., 2013). In the atmosphere, mercury exists as one of three operationally-defined species: gaseous elemental mercury (GEM), gaseous oxidised mercury (GOM) and particulate-bound mercury (PBM), each with differing abundances, solubility and depositional characteristics and with in-air conversion between all three species possible (Lin and Pehkonen, 1999). Mercury can be scavenged from the atmosphere through both wet and dry depositional processes, and the monsoonal climate of northern Australia results in varying significance of each of these processes through the year (Packham et al., 2009). Upon deposition, mercury may be stored in plant tissue via stomatal or cuticular uptake (Rea et al., 2002) or sequestered within soils (Gustin et al., 2008). Release from both of these pools is achieved from burning events that may volatilise or thermally desorb mercury from biomass and soil, respectively (Melendez-Perez et al., 2014). Subsequently this mercury pool is redistributed through the atmospheric pathway to ecosystems that may methylate mercury, thereby enhancing its bioavailability to the local food chain.

SAFIRED represents the first measurements of atmospheric mercury undertaken in the tropical region of the Australian continent. The mean observed GEM concentration over the study period was $0.99 \pm 0.09$ ng m$^{-3}$, similar to the average over that month (0.96 ng m$^{-3}$) for 5 other Southern Hemisphere sites and slightly lower than the average (1.15 ng m$^{-3}$) for 5 tropical sites (Sprovieri et al., 2016). Mean GOM and PBM concentrations were $11 \pm 5$ pg m$^{-3}$ and $6 \pm 3$ pg m$^{-3}$ respectively, representing $0.6 - 3.4\%$ of total observed atmospheric mercury.

Atmospheric mercury measurements were available only during the final four identified burn events. During these events, spikes in GEM concentrations were observed, though there were no significant increases in GOM or PBM. Emission ratios calculated during the campaign were two orders of magnitude higher than those reported by Andreae and Merlet (2001), though those were from scrub, rather than grass, BB events (Desservettaz et al., 2016). Future outcomes from the SAFIRED campaign will focus on the use of micrometeorological techniques and the passive tracer radon to quantify delivery of atmospheric mercury to tropical savannah ecosystems. ATARS

also now serves as an additional site measuring continuous GEM as part of the Global Mercury Observation System (GMOS), one of only two tropical observing sites in the Eastern Hemisphere and the third such site located in Australia.

**Aging of aerosols**
* * *
Atmospheric chemistry and radiative forcing will depend on how gaseous and aerosol emissions from fires age as they move and interact with each other and existing species in the atmosphere. Biomass burning aerosols can be involved in condensation and coagulation (Radhi et al., 2012), undergo water uptake (Mochida and Kawamura, 2004) form cloud droplets (Novakov and Corrigan, 1996), and be exposed to photochemical aging processes, including those involving the gaseous components of fire emissions (Keywood et al., 2011;Keywood et al., 2015). With a reported lifetime of $3.8 \pm 0.8$ days (Edwards et al., 2006), biomass burning aerosols are able to travel intercontinental distances (Rosen et al., 2000) and are therefore present in the atmosphere long enough for substantial changes due to aging. Furthermore, tropical convection is likely to affect the aging of BB emissions in the region around ATARS, due to the immediate proximity to the warm waters in the Timor Sea (Allen et al., 2008). This introduces further uncertainty to the effect of BB emissions on radiation flux.

Primary organic aerosol directly emitted from biomass burning can interact with NMOCs to change composition and mass, resulting in secondary organic aerosol (Hallquist et al., 2009). Photochemical oxidation of NMOCs occurs during the daytime by either hydroxyl radicals or ozone. Ozone is also typically produced in the aging processes of tropical biomass burning plumes when NMOCs can oxidise to produce peroxy radicals that react with NO. Photochemical reactions also may lead to an overall increase in total aerosol mass through the condensation of NMOCs onto existing particles (Reid et al., 1998;Yokelson et al., 2009;Akagi et al., 2012;DeCarlo et al., 2008). Some studies have shown the opposite, i.e., photo-oxidation can also lead to the evaporation of some primary organic constituents, resulting in an overall mass reduction (Hennigan et al., 2011;Akagi et al., 2012). With thousands of organic compounds in the atmosphere, each with different volatilities and potential reaction mechanisms, our understanding of secondary organic aerosol production is limited (Goldstein and Galbally, 2007;Keywood et al., 2011). Furthermore, secondary organic aerosol can also form through aqueous phase reactions where water-soluble organics dissolve into water on existing particles (Lim et al., 2010).

Further analysis into the aerosol chemical composition will elucidate the aging of early dry season biomass burning emissions. Fractional analysis (e.g., f44 and f60, the fraction of m/z 44 and m/z 60 to all organic masses, indicated oxygenation and BB sources, respectively) and factor analysis using positive matrix factorisation (PMF) of cToF-AMS data has been investigated over the entire sampling period. Outside of the periods of significant influence from BB events, three PMF-resolved organic aerosol factors were identified. A BB organic aerosol factor was found to comprise 24% of the submicron non-refractory organic mass, with an oxygenated organic aerosol factor and a biogenic isoprene-related secondary organic aerosol factor comprising 47% and 29%, respectively. These results indicate the significant influence of fresh and aged BB on aerosol composition in the early dry season. The emission of precursors from fires is likely responsible for some of the SOA formation. A full discussion of these results can be found in Milic et al. (2016). Future analysis will investigate the gas and particle-phase composition for individual BB events.

Water uptake of aerosols

The water uptake by aerosols is determined by their size and composition, as well as the atmospheric humidity (McFiggans et al., 2006). The hygroscopic properties of all of the different components of an aerosol particle contribute to its total hygroscopicity (Chen et al., 1973;Stokes and Robinson, 1966). The presence of different water-soluble and water-insoluble organics and inorganics will therefore strongly influence water uptake. Furthermore, chamber studies that have investigated emissions from biomass fuels, both separately and in combination, have shown that the hygroscopic response can vary significantly depending on fuel type (Carrico et al., 2010). Understanding the water uptake of atmospheric aerosols is further complicated when considering other aging processes as described previously. Nonetheless, it is important to characterise the water uptake, as this will, in turn, influence other atmospheric chemistry processes, radiation scattering and absorption as well as cloud processing.

Biomass burning aerosols can act as cloud condensation nuclei if they are large enough for water to easily condense onto their surface, or if the particles have a large affinity for water due to their composition (Novakov and Corrigan, 1996). Ultimately, this means that BB emissions can lead to a higher number of cloud droplets. This is important in reflecting solar radiation and cooling the earth's surface. Cloud albedo is more susceptible to changes when cloud condensation nuclei concentrations are relatively low (Twomey, 1991), such as in marine environments like the Timor Sea off the coast of northern Australia.

The water uptake of aerosols has been further investigated to identify the possible influence of early dry season BB in this region on cloud formation. The concentrations of cloud condensation nuclei at a constant supersaturation of 0.5% were typically of the order of 2000 cm$^{-3}$ and reached well over 10000 cm$^{-3}$ during intense BB events. Variations in the ratio of aerosol particles activating cloud droplets showed a distinct diurnal trend, with an activation ratio of 40% ± 20% during the night and 60% ± 20% during the day. The particle size distribution and the hygroscopicity of the particles were found to significantly influence this activation ratio. A full discussion of these results can be found in Mallet et al. (2016, submitted). Future analysis will elucidate the contribution of different biomass burning aerosol components on the hygroscopicity.

Trace metal deposition

The deposition and dissolution of aerosols containing trace metals into the ocean may provide important micronutrients required for marine primary production. Conversely, the deposition of soluble iron can trigger toxic algal blooms, such as Trichodesmium, in nutrient-poor tropical and subtropical waters (LaRoche and Breitbarth, 2005). Trichodesmium blooms require large quantities of soluble iron, of which aerosols are a source (Boyd and Ellwood, 2010;Rubin et al., 2011). To date, most studies have assumed that mineral dust aerosols represent the primary source of soluble iron in the atmosphere (Baker and Croot, 2010); however fire emissions and oil combustion are other likely sources (Ito, 2011;Schroth et al., 2009;Sedwick et al., 2007). A few studies have shown that iron contained in biomass burning emissions is significantly more soluble than mineral dust (Guieu et al., 2005;Luo et al., 2008;Schroth et al., 2009) but, to date, no data exists for Australian fires.

The aim of the trace metal aerosol component of SAFIRED is to quantify, for the first time, the fractional solubility of aerosol iron, and other trace metals, derived from Australian dry season BB. The fractional iron solubility is an important variable determining iron availability for biological uptake. On a global scale, the large variability in the observed fractional iron solubility results, in part, from a mixture of different aerosol sources. Estimates of fractional iron solubility from fire combustion  (1 - 60 %) are thought to be greater than those originating from mineral dust (1 - 2%) (Chuang et al., 2005;Guieu et al., 2005;Sedwick et al., 2007), and may vary in relationship to biomass and fire characteristics as well as that of the underlying terrain (Paris et al., 2010;Ito, 2011). Iron associated with BB may provide information with respect to BB inputs of iron to the ocean (Giglio et al., 2013;e.g. Meyer et al., 2008). The ATARS provides an ideal location to further investigate BB derived fractional iron solubility at the source. The results from this study can be found in Winton et al. (2016) and show that soluble iron concentrations from BB sources are significantly higher than those observed in Southern Ocean baseline air masses from the Cape Grim Baseline Air Pollution Station, Tasmania, Australia (Winton et al., 2015). Aerosol iron at SAFIRED was a mixture of fresh BB, mineral dust, sea spray and industrial pollution sources. The fractional iron solubility (2 - 12%) was relatively high throughout the campaign and the variability was related to the mixing and enhancement of mineral dust iron solubility with BB species.

---

## Author Response (AR2)

**Editor's comment:**

Thank you again for your patience. The original reviewers requested to see the revised manuscript, and thus, I allowed them to re-review it. I've finally heard back from both reviewers. They agree the manuscript is much improved and should be accepted; however, one reviewer requested you address the very minor comments I've copied below. Once you address these final comments, I'll immediately accept as is.

Thanks again, Jason Surratt

Remaining comments from one of the reviewers:

"The authors clearly took the reviewers comments into account and I think that the manuscript has been improved significantly, for example the addition of the science questions in the Introduction section was very helpful. I just have a few small additional comments.

1.) The addition of Table 1 was important, but I still would like to see a little more details on the instruments such as the sensitivity and limits of detection.

2.) The diurnal variations of the trace gases that are in the supplement should be in the main text combined with the meteorological parameters in Figure 5.

3.) In Figure 3, I meant to have CO on a linear AND a log scale.

4.) How confident are you in the ozone data in Figure 5? If there are cross contaminations in the ozone data should they be included in this Figure or taken out here as well. If the data are used the interference has to be discussed in detail and described, which data are still useful.

5.) The outcomes of SAFIRED section is now much improved and very useful.

**Author's response:**

The author's thank the editor and reviewers for the suggestions and comments. Table 1 has been expanded to include more instrumental details such as detection limits and uncertainties where available. The diurnal trends of trace gases from the supplementary material has been combined in the main body within Figure 5. The CO time series in FIgure 3 now shows both a linear and a log scale. Periods of data when interferences were suspected in ozone measurements have been excluded from Figure 6. This has been briefly explained in the experimental section. This was done by investigating periods of strong biomass burning signals (using e.g. acetonitrile as a marker) and removing periods where concentrations were high and well correlated with ozone (they should not be co-emitted).

**Title: Biomass burning emissions in north Australia during the early dry season: an overview of the 2014 SAFIRED campaign**

**Authors:**

Marc D. Mallet[1], Maximilien J. Desservettaz[2], Branka Miljevic[1*], Andelija Milic[1], Zoran D. Ristovski[1], Joel Alroe[1], Luke T. Cravigan[1], E. Rohan Jayaratne[1], Clare Paton-Walsh[2], David W.T. Griffith[2], Stephen R. Wilson[2], Graham Kettlewell[2], Marcel V. van der Schoot[3], Paul Selleck[3], Fabienne Reisen[3], Sarah J. Lawson[3], Jason Ward[3], James Harnwell[3], Min Cheng[3], Rob W. Gillett[3], Suzie B. Molloy[3], Dean Howard[4], Peter F. Nelson[4], Anthony L. Morrison[4], Grant C. Edwards[4], Alastair G. Williams[5], Scott D. Chambers[5], Sylvester Werczynski[5], Leah R. Williams[6], V. Holly L. Winton[7,n], Brad Atkinson[8], Xianyu Wang[9], Melita D. Keywood[3*]

**Affiliations:**

[1]Department of Chemistry, Physics and Mechanical Engineering, Queensland University of Technology, Queensland, Brisbane, 4000, Australia

[2]Centre for Atmospheric Chemistry, University of Wollongong, Wollongong, New South Wales, 2522, Australia

[3]CSIRO Oceans and Atmosphere, Aspendale, Victoria, 3195, Australia

[4]Department of Environmental Sciences, Macquarie University, Sydney, New South Wales, 2109, Australia

[5]Australian Nuclei Science and Technology Organisation, Sydney, New South Wales, 2232, Australia

[6]Aerodyne Research, Inc., Billerica, Massachusetts, 01821, USA

[7]Physics and Astronomy, Curtin University, Perth, Western Australia, 6102, Australia

[8]Bureau of Meteorology, Darwin, Northern Territory, 0810, Australia

[9]National Research Centre for Environmental Toxicology, Brisbane, Queensland, 4108, Australia

[n]Now at the British Antarctic Survey, Cambridge, CB3 0ET, United Kingdom

**\*Corresponding Authors:**

Dr Melita Keywood

Contact Phone: +613 9239 4596

Contact Email: melita.keywood@csiro.au

Dr Branka Miljevic

Contact Phone: +61 7 3138 3827

Contact Email: b.miljevic@qut.edu.au

**Keywords:**

Biomass burning | savannah fires | greenhouse gases | aerosols | mercury

**Abstract**

The SAFIRED (Savannah Fires in the Early Dry Season) campaign took place from 29th of May, 2014 until the 30th June, 2014 at the Australian Tropical Atmospheric Research Station (ATARS) in the Northern Territory, Australia. The purpose of this campaign was to investigate emissions from fires in the early dry season in northern Australia. Measurements were made of biomass burning aerosols, volatile organic compounds, polycyclic aromatic carbons, greenhouse gases, radon, speciated atmospheric mercury, and trace metals. Aspects of the biomass burning aerosol emissions investigated included; emission factors of various species, physical and chemical aerosol properties, aerosol aging, micronutrient supply to the ocean, nucleation, and aerosol water uptake. Over the course of the month-long campaign, biomass burning signals were prevalent and emissions from several large single burning events were observed at ATARS.

Biomass burning emissions dominated the gas and aerosol concentrations in this region. Dry season fires are extremely frequent and widespread across the northern region of Australia, which suggests that the measured aerosol and gaseous emissions at ATARS are likely representative of signals across the entire region of north Australia. Air mass forward trajectories show that these biomass burning emissions are carried north west over the Timor Sea and could influence the atmosphere over Indonesia and the tropical atmosphere over the Indian Ocean. Here, we present characteristics of the biomass burning observed at the sampling site and provide an overview of the more specific outcomes of the SAFIRED campaign.

Comment [MM3]: R1:

Abstract; This section is too long and needs to be tightened up considerably.

Comment [MM4]: R1:
Line 45. How does one measure the "mercury cycle"? It is possible to measure the chemical species that make up the mercury cycle.

**1. Introduction**

Tropical north Australia is dominated by savannah ecosystems. This region consists of dense native and exotic grasslands and scattered trees and shrubs. Conditions are hot, humid and wet in the summer months of December through March with hot, dry conditions for the rest of the year giving rise to frequent fires between June and November each year. Human settlements are relatively scarce in northern Australia, outside of the territory capital, Darwin (population of 146 000). To the north of the continent are the tropical waters of the Timor Sea, as well as the highly populated Indonesian archipelago. South of the savannah grasslands are the Tanami, Simpson and Great Sandy Deserts, spanning hundreds of thousands of square kilometers. Emissions from fires in the savannah regions of northern Australia are therefore the most significant regional source of greenhouse and other trace gases, as well as atmospheric aerosol. Globally, savannah and grassland fires are the largest source of carbon emissions from biomass burning (van der Werf et al., 2010;Shi et al., 2015) and play a significant role in the earth's radiative budget. It is therefore important to quantify, characterise and fully understand the emissions from savannah fires in northern Australia, taking into account the complexity, variability and diversity of the species emitted.

In Australia approximately 550 000 km$^2$ of tropical and arid savannahs burn each year (Meyer et al., 2012;Russell-Smith et al., 2007), representing 7% of the continent's land area. In the tropical north of Australia, the fires during the early dry season in May/June consist of naturally occurring and accidental fires, as well as prescribed burns under strategic fire management practice to reduce the frequency and intensity of more extensive fires in the late dry season in October and November (Andersen et al., 2005).
* * *
**Comment [MM8]:** R3:
While some of the relevant literature has been cited, it would be worthwhile to discuss (likely in the Introduction) similar measurement campaigns performed in other parts of the world that have examined emissions from biomass burning and how those earlier lab and field efforts (e.g., BBOP, SCREAM, FLAME1-5, etc) have helped inform critical gaps, research questions, instrumentation, analysis techniques etc. for the SAFIRED campaign.

**Comment [MM9]:** R3:
Being an overview article, I think the manuscript could benefit from a schematic and/or cartoon in the introduction that sketches the region of interest (Northern Aus- tralia) and caricatures the emissions, processes and impacts being studied in detail in this campaign. Furthermore, a bulleted list in the beginning of the manuscript that lists the research/science questions for SAFIRED would provide context for the various measurements and analysis performed.

**Comment [MM10]:** R1:
Lines 80 and 81. Savannah and grassland fires are not the largest source of carbon to the atmosphere, as is clear when comparing the numbers from the quoted references with the global anthropogenic source of $CO_2$ for example. Do the authors mean the largest source of black carbon?

[revised manuscript text omitted]

PTR-MS which measures these species was not operational during BBP1 and CP. The

Comment [MM32]: R2:
The ozone enhancement shown in Figure 8 for the close proximity fire is substan- tial and ozone values of almost 100ppb are detected in the plume. This means that there has been significant photochemical processing of potentially several hours dur- ing plume transport. If the plume would be really fresh, ozone would actually be titrated. Again VOC/CO could be very helpful here and should be looked into. Also a compar- ison to a nighttime plume measurement would be very useful. Again, I doubt that this plume is very fresh.

Comment [MM33]: R2:
O3/CO ratios: The O3/CO is used in Figures 5 and 8 and is described at giving an indication of photochemical age. Unfortunately O3/CO are much more complicated than that and depend on many different factors such as VOC/NOx ratios such that the ratio really cannot be used as "photochemical age". I think for this paper here, it is best to remove the O3/CO ratios instead of adding a proper explanation.

Line 623. The authors seem to be ignoring the high O3 plume during BBP4 which indicates faster O3 production in this plume. This would seem to be one of the more interesting observations of this study.

Moved (insertion) [3]

diurnal trends for the toluene and acetonitrile concentrations and the toluene/acetonitrile ratio is shown in Figure 7 for BBP2, BBP3 and BBP4. The toluene/acetonitrile ratio was highest during the night, indicating more photochemically aged smoke throughout the day. Interestingly, while the toluene and acetonitrile concentrations were consistently higher during BBP2 and BBP4 than BBP3, the toluene/acetonitrile ratio was of the same magnitude and followed the same trend. It is therefore plausible that, while there were not large enhancements in concentrations during BBP3 and there were few fires detected close-by during the daytime satellite flyovers, there were small-scale burns during the night that were close enough for the emissions to reach sampling site. This observation highlights the limitation of using satellite hotspot detection in fully understanding the aging processes of biomass burning emissions.

[Figure]

(a)    (b)    (c)

**Figure 7 Mean hourly diurnal (a) acetonitrile concentration, (b) toluene concentration, (c) toluene/acetonitrile ratio, separated into different biomass burning periods (BBP).**

Particle size distributions were unimodal for the majority of the sampling period with a mode of approximately 100 nm on average (see Figure 8). The SMPS was not operational during BBP1. Although the shape of the BBP4 size distribution was similar to the campaign average, concentrations were much higher and a result of close fires. BBP2 had a slightly larger size distribution centered on 110 nm. The size distribution during BBP3 was slightly smaller than the campaign average and BBP2 and BBP4, with a mode centered on ~95 nm. Furthermore, the diurnal trends of the BBA mode diameter during BBP2, BBP3 and BBP4 and CP all showed a clear maximum during the night (see Supplementary Figure S2d). The diurnal trends of the toluene/acetonitrile ratios (Figure 7c) as well as the ratio of oxygenated organic aerosol to total organics (see Supplementary Figure S2c) suggest that the larger night time particle sizes are more associated with fresh biomass burning. The contrast between these size distributions could be a result of atmospheric aging and dilution in which organic mass condenses onto or evaporates from the particle. Variations in fuel load or burning conditions could also contribute to this difference. The size and concentration of particles during the Coastal Period (CP) were much smaller than the rest of the campaign. There were two periods during CP where a bimodal size distribution was observed; one from approximately 3 pm until midnight on the 19th of June and the other between 2 pm and 6 pm on the 20th of June. The size distributions for both of these periods had a mode at approximately 20 nm and another at approximately 85 nm. Submicron sulfates made up to 32% of the total submicron non-refractory mass concentrations, as reported by the cToF-AMS from the period of midday on the 19th of June until midnight on the 22nd of June, whereas the average sulfate contribution for the rest of the campaign was approximately 8%. The low radon values, small particle concentrations, bimodal size distributions and significant contributions of sulfate during this period also suggest very little biomass burning signal and a more marine-like aerosol. No particle nucleation events were observed over the entire sampling period (See Supplementary Figure S3). This is likely due to the elevated particle concentrations acting as a condensation sink.

[Figure]

**Comment [MM44]:** R3:
9. Figure 7: Are those raw SMPS data or lognormal fits to the SMPS data? The distributions look uncannily smooth.

[revised manuscript text omitted]

Caption, Keep with next

| Page 10: [2] Formatted | Marc Mallet | 3/27/17 1:35:00 AM |

Left:  1", Right:  1", Top:  1.25", Bottom:  1.25", Width:  11.69", Height:  8.26", Header distance from edge:  0.49", Footer distance from edge:  0.49"

| Page 10: [3] Formatted Table | Marc Mallet | 9/15/17 7:54:00 AM |

Formatted Table

| Page 10: [4] Formatted | Marc Mallet | 9/14/17 9:29:00 PM |

Subscript

| Page 10: [4] Formatted | Marc Mallet | 9/14/17 9:29:00 PM |

Subscript

| Page 10: [4] Formatted | Marc Mallet | 9/14/17 9:29:00 PM |

Subscript

| Page 10: [4] Formatted | Marc Mallet | 9/14/17 9:29:00 PM |

Subscript

| Page 10: [4] Formatted | Marc Mallet | 9/14/17 9:29:00 PM |

Subscript

| Page 10: [4] Formatted | Marc Mallet | 9/14/17 9:29:00 PM |

Subscript

| Page 10: [4] Formatted | Marc Mallet | 9/14/17 9:29:00 PM |

Subscript

| Page 10: [4] Formatted | Marc Mallet | 9/14/17 9:29:00 PM |

Subscript

| Page 10: [4] Formatted | Marc Mallet | 9/14/17 9:29:00 PM |

Subscript

| Page 10: [5] Formatted | Marc Mallet | 9/14/17 9:09:00 PM |
|---|---|---|

Subscript

| Page 10: [5] Formatted | Marc Mallet | 9/14/17 9:09:00 PM |
|---|---|---|

Subscript

| Page 10: [5] Formatted | Marc Mallet | 9/14/17 9:09:00 PM |
|---|---|---|

Subscript

| Page 10: [5] Formatted | Marc Mallet | 9/14/17 9:09:00 PM |
|---|---|---|

Subscript

| Page 10: [6] Formatted | Marc Mallet | 9/14/17 4:34:00 PM |
|---|---|---|

Font:8 pt

| Page 10: [6] Formatted | Marc Mallet | 9/14/17 4:34:00 PM |
|---|---|---|

Font:8 pt

| Page 10: [7] Formatted | Marc Mallet | 5/15/17 12:51:00 AM |
|---|---|---|

Font:8 pt

| Page 10: [7] Formatted | Marc Mallet | 5/15/17 12:51:00 AM |
|---|---|---|

Font:8 pt

| Page 10: [8] Formatted | Marc Mallet | 5/15/17 12:51:00 AM |
|---|---|---|

Font:8 pt

| Page 10: [9] Formatted | Marc Mallet | 9/11/17 9:52:00 AM |
|---|---|---|

Superscript

| Page 10: [10] Formatted | Marc Mallet | 5/15/17 12:51:00 AM |
|---|---|---|

Font:8 pt

| Page 10: [11] Formatted | Marc Mallet | 9/11/17 9:13:00 AM |
|---|---|---|

Superscript

| Page 10: [11] Formatted | Marc Mallet | 9/11/17 9:13:00 AM |
|---|---|---|

Superscript

| Page 10: [11] Formatted | Marc Mallet | 9/11/17 9:13:00 AM |
|---|---|---|

Superscript

| Page 10: [12] Formatted | Marc Mallet | 9/15/17 7:56:00 AM |
|---|---|---|

Font:Bold, Superscript

| Page 10: [13] Formatted | Marc Mallet | 6/15/17 9:00:00 AM |
|---|---|---|

Font:Not Bold

| Page 10: [14] Formatted | Marc Mallet | 6/15/17 9:00:00 AM |
|---|---|---|

Font:8 pt

| Page 10: [15] Formatted | Marc Mallet | 5/15/17 12:52:00 AM |
|---|---|---|

Centered, Position:Horizontal: Left, Relative to: Column, Vertical:  0", Relative to: Paragraph,

Horizontal:  0.13", Wrap Around

| Page 10: [16] Formatted | Marc Mallet | 5/15/17 12:51:00 AM |
|---|---|---|

Font:8 pt

| Page 10: [17] Formatted | Marc Mallet | 5/15/17 12:52:00 AM |
|---|---|---|

Centered, Position:Horizontal: Left, Relative to: Column, Vertical:  0", Relative to: Paragraph,

Horizontal:  0.13", Wrap Around

| Page 10: [18] Formatted | Marc Mallet | 5/15/17 12:51:00 AM |
|---|---|---|

Font:8 pt

| Page 10: [19] Formatted | Marc Mallet | 5/15/17 12:52:00 AM |
|---|---|---|

Centered, Position:Horizontal: Left, Relative to: Column, Vertical:  0", Relative to: Paragraph,

Horizontal:  0.13", Wrap Around

| Page 10: [20] Formatted | Marc Mallet | 5/15/17 12:51:00 AM |
|---|---|---|

Font:8 pt

| Page 10: [21] Formatted | Marc Mallet | 9/11/17 9:16:00 AM |
|---|---|---|

Subscript

| Page 10: [22] Formatted | Marc Mallet | 5/15/17 12:52:00 AM |
|---|---|---|

Centered, Position:Horizontal: Left, Relative to: Column, Vertical:  0", Relative to: Paragraph,

Horizontal:  0.13", Wrap Around

| Page 10: [23] Formatted | Marc Mallet | 5/15/17 12:51:00 AM |
|---|---|---|

Font:8 pt

| Page 10: [24] Formatted | Marc Mallet | 9/14/17 9:00:00 PM |
|---|---|---|

Superscript

| Page 10: [25] Formatted | Marc Mallet | 9/11/17 9:16:00 AM |
|---|---|---|

Subscript

| Page 10: [26] Formatted | Marc Mallet | 5/15/17 12:52:00 AM |
|---|---|---|

Centered, Position:Horizontal: Left, Relative to: Column, Vertical:  0", Relative to: Paragraph,

Horizontal:  0.13", Wrap Around

| Page 10: [27] Formatted | Marc Mallet | 5/15/17 12:51:00 AM |
|---|---|---|

Font:8 pt

| Page 10: [28] Formatted | Marc Mallet | 9/15/17 7:53:00 AM |
|---|---|---|

Superscript

| Page 10: [29] Formatted | Marc Mallet | 9/11/17 9:16:00 AM |
|---|---|---|

Subscript

| Page 10: [30] Formatted | Marc Mallet | 5/15/17 12:51:00 AM |
|---|---|---|

Font:8 pt

| Page 10: [31] Formatted | Marc Mallet | 5/15/17 12:52:00 AM |
|---|---|---|

Centered, Position:Horizontal: Left, Relative to: Column, Vertical: 0", Relative to: Paragraph,

Horizontal: 0.13", Wrap Around

| Page 10: [32] Formatted | Marc Mallet | 9/11/17 9:14:00 AM |
|---|---|---|

Superscript

| Page 10: [32] Formatted | Marc Mallet | 9/11/17 9:14:00 AM |
|---|---|---|

Superscript

| Page 10: [32] Formatted | Marc Mallet | 9/11/17 9:14:00 AM |
|---|---|---|

Superscript

| Page 10: [32] Formatted | Marc Mallet | 9/11/17 9:14:00 AM |
|---|---|---|

Superscript

| Page 10: [32] Formatted | Marc Mallet | 9/11/17 9:14:00 AM |
|---|---|---|

Superscript

| Page 10: [32] Formatted | Marc Mallet | 9/11/17 9:14:00 AM |
|---|---|---|

Superscript

| Page 10: [32] Formatted | Marc Mallet | 9/11/17 9:14:00 AM |
|---|---|---|

Superscript

| Page 10: [33] Formatted | Marc Mallet | 5/15/17 12:52:00 AM |
|---|---|---|

Centered, Position:Horizontal: Left, Relative to: Column, Vertical: 0", Relative to: Paragraph,

Horizontal: 0.13", Wrap Around

| Page 10: [34] Formatted | Marc Mallet | 5/15/17 12:51:00 AM |
|---|---|---|

Font:8 pt

| Page 10: [35] Formatted | Marc Mallet | 9/11/17 9:46:00 AM |
| --- | --- | --- |

Line spacing: double

| Page 10: [36] Formatted | Marc Mallet | 9/14/17 9:07:00 PM |
| --- | --- | --- |

Keep with next

| Page 16: [37] Deleted | Marc Mallet | 5/1/17 10:12:00 PM |
| --- | --- | --- |

**Radon-222 (radon) is a naturally occurring radioactive noble gas that arises from the alpha-particle decay of radium-226, which is ubiquitous in most soil and rock types. With a half-life of 3.82 days, radon has thus proven to be an excellent indicator of recent (within 2-3 weeks) terrestrial influences on air masses for observations at coastal or island sites (Chambers et al., 2014). Radon is unreactive and poorly soluble, and its only atmospheric sink is radioactive decay. Furthermore, it has a half-life comparable to the lifetimes of short-lived atmospheric pollutants (e.g., NOx, SO$_2$) and atmospheric residence time of water and aerosols[MM1]. Radon's unique combination of physical characteristics make it an ideal tracer for (i) regional air mass transport studies, in which it is often used in conjunction with air mass back trajectories for fetch [MM2]analyses (Williams et al., 2009;Chambers et al., 2014); (ii) investigations of vertical mixing processes within the daytime convective boundary layer (Williams et al., 2011) or nocturnal boundary layer (Chambers et al., 2015); (iii) identifying periods of minimal terrestrial influence on a measured air mass ("baseline" studies; (Chambers et al., 2016)); (iv) performing regional flux or inventory analyses for trace atmospheric species with similar source distributions (Biraud et al., 2000) and (v) evaluating the performance of transport and mixing schemes in climate and chemical-transport models (Locatelli et al., 2015).**

**Afternoon radon concentrations provide further information regarding the regional air mass fetch and the degree of contact with the land surface (red line, Figure 3a). Over the campaign period, air masses with the least terrestrial fetch (low radon indicates strongest oceanic signature) were observed on June 4-6 and 20-22, whereas June 8-9, 17-18 and 29-30 represented periods of particularly extensive continental fetch.**

[Figure]

Figure **3** Hourly ATARS radon observations for June 2014: (a) observed hourly data, and afternoon-to-afternoon interpolated values (indicative of changes in the regional air mass fetch); and (b) difference in radon concentration between the hourly observations and interpolated afternoon values (indicative of diurnal variability).

A pronounced diurnal variability can clearly be seen in the ΔRn signal (Figure 3b). Mean hourly diurnal composites of radon concentrations, wind speed, wind direction and dew point temperature at the ATARS site during the period of the SAFIRED campaign are shown in Figure 4. Following the technique described in Chambers et al. (2016), these composites have been computed separately for three diurnal mixing categories based on the mean ΔRn over the 12 hour period 2000-0800 h:

Strong mixing:                $\Delta Rn_{12} < 5400$ mBq m$^{-3}$

Moderate mixing:     $5400 \leq \Delta Rn_{12} < 6700$ mBq m$^{-3}$

Weak mixing: $\Delta Rn_{12} \geq 6700$ mBq m$^{-3}$

The air masses predominantly originated from the southeast as indicated in Figure 1 and Figure 4c. Starting from approximately 10:00 am each morning, however, sea breeze circulations slowly turn the measured wind direction around from southeast to northeast, before reverting back to the dominant wind direction again at around midnight. Wind speeds reached a maximum just before midday and were at their lowest just before midnight (Figure 4b). The "strong mixing" category was associated with generally higher wind speeds, which cause increased mechanical turbulence leading to deeper nocturnal mixing layers (i.e., hinder the development of a shallow nocturnal inversion layer).[MM3]

| Page 32: [40] Deleted | Marc Mallet | 5/14/17 10:38:00 PM |

Gas and Aerosol measurements

The campaign average, standard deviation, median and Q25/Q75 values for the major gaseous and aerosol species are shown in Table 1. The median values for each species are likely to be representative of background concentrations in this region[MM4]. The average concentrations for most species were higher than the median concentrations, due to the periods of close or intense fires. The extent of the influence of these close fires are demonstrated by the maximum concentrations.

Table 1 The campaign average, standard deviation, maximum, median, Q25 and Q75 values for key measured gas and aerosol species. All parts-per notation refer to mole fractions unless otherwise indicated.

| Species (unit) | Average | Standard deviation | Maximum | Median | Q25 | Q75 |
|---|---|---|---|---|---|---|
| CO (ppb) | 229 | 494 | 18900 | 130 | 87 | 214 |
| CO$_2$ (ppm) | 404.68 | 11.539 | 513.578 | 402.454 | 394.728 | 411.299 |
| O$_3$ (ppbv) | 24.616 | 9.903 | 99.784 | 22.771 | 17.896 | 29.778 |

| | | | | | | |
|---|---|---|---|---|---|---|
| CH$_4$ (ppb) | 1839.88 | 68.06 | 3766.81 | 1820.11 | 1802.26 | 1852.97 |
| N$_2$O (ppb) | 326.329 | 0.449 | 334.871 | 326.276 | 326.121 | 326.444 |
| GEM (ng m$^{-3}$) | 0.992 | 0.081 | 1.734 | 0.986 | 0.952 | 1.020 |
| Acetonitrile (ppb) | 0.351 | 0.629 | 9.775 | 0.197 | 0.129 | 0.337 |
| Organics (ug m$^{-3}$) | 11.081 | 22.385 | 347.657 | 4.160 | 2.335 | 13.279 |
| SO$_4^{2-}$ (ug m$^{-3}$) | 0.514 | 0.318 | 2.254 | 0.411 | 0.294 | 0.679 |
| NH$_4^+$ (ug m$^{-3}$) | 0.351 | 0.676 | 18.17 | 0.180 | 0.096 | 0.415 |
| NO$_3^-$ (ug m$^{-3}$) | 0.187 | 0.456 | 10.925 | 0.042 | 0.004 | 0.189 |
| Cl$^-$ (ug m$^{-3}$) | 0.166 | 1.271 | 53.270 | 0.029 | 0.016 | 0.076 |
| PNC (cm$^{-3}$) | 8182 | 19031 | 40300 | 2032 | 2032 | 8335 |
| Mode diameter (nm) | 104 | 31 | - | 102 | 85 | 122 |
| Geom. SD | 1.71 | 0.13 | - | 1.70 | 1.65 | 1.75 |

[MM5][MM6]

In order to demonstrate the influence of close fires and the changing inversion layer, the time series of major greenhouse gases (CO, CO$_2$, CH$_4$ and N$_2$O), gaseous elemental mercury, acetonitrile and ozone throughout the campaign are shown in Figure 5. Sub-micron non-refractory aerosol organic, sulfate, ammonium and nitrate mass concentrations, organic mass fraction, PM$_1$ OC and EC mass concentrations and particle size distributions for the sampling period are shown in Figure 6. Periods of missing data correspond to times when instruments were not operating. Most of these time series display a clear diurnal trend as a result of the varying inversion layer height. Other enhancements in concentrations can be clearly seen and correspond to periods of frequent close fires (Figure 2). [MM7]Over the entire sampling period from the 29th of May 2014 until the 28th of June 2014, four biomass burning related periods (BBP) and a "coastal" period (CP) have been distinguished. The dates for these periods are shown in Table 2. These periods are also displayed in Figure 5 and Figure 6. [MM8]

Table 2 The start and end dates for the four identified Biomass Burning Periods (BBP1, BBP2, BBP3 and BBP4) and the Coastal Period (CP).

| Period | Start date (mm/dd/yy hh:mm) | End date (mm/dd/yy hh:mm) |
|--------|------------------------------|----------------------------|
| BBP1 | 05/30/14 00:00 | 05/31/14 23:59 |
| BBP2 | 06/06/14 00:00 | 06/12/14 23:59 |
| BBP3 | 06/14/14 00:00 | 06/17/14 23:59 |
| CP | 06/19/14 12:00 | 06/22/14 23:59 |
| BBP3 | 06/23/14 00:00 | 06/28/14 23:59 |

[Figure]

[MM9]

Figure 5 The time series of the major measured gaseous species during the SAFIRED campaign: (a) carbon monoxide, (b) carbon dioxide, (c) methane, (d) nitrogen dioxide[MM10], (e) gaseous elemental mercury, (f) acetonitrile and (g) ozone and ΔO₃/ΔCO. The biomass burning and coastal periods are indicated by the red dotted lines. All parts-per notation refer to mole fractions unless otherwise indicated.

[Figure]

[MM11]

Figure 6 The times series of the major aerosol properties during the SAFIRED campaign: (a) the non-refractory PM$_1$ organic mass concentration (left) and organic mass fraction (right), b) the inorganic non-refractory PM$_1$ mass concentrations, (c) the 12-hour filter OC and EC PM$_1$ mass concentrations (left) and the ratio of OC to OC+EC (right), (d) the particle size distributions and particle size mode (left) and the total particle number concentration (right) and (e) the wind direction at ATARS.

Over the campaign organics dominated the non-refectory sub-micron aerosol mass contributing, on average 90% (median; 86%) of the total mass. Sulphate, nitrates, ammonium and chloride species contributed the rest of this mass, with the largest contributions from sulphate and ammonium. Sulphate contributions were very significant during the coastal period, contributing up to 32% of the total mass. Although chlorides contributed the least to the total mass, on average, during clear biomass burning events where sharp increases in CO and organics were observed, chlorides made up the largest component of inorganic aerosol[MM12]. The maximum chloride concentration during the campaign reached 53 μg m$^{-3}$. High soil and vegetation chloride contents have been observed in savannah and coastal environments (Lobert et al., 1999;Andreae et al., 1996). The strong elevations of chloride signals observed, particularly during burning events BBP1, BBP2 and BBP4 are likely a result of emission of these chloride ions. Outside of the "burning events" where very sharp increases in concentrations were observed, chloride concentrations were very low. This either suggests that these chloride species are short-lived, or only present in fires very close to the coast and therefore the ATARS site. [MM13]

| Page 32: [41] Formatted | Marc Mallet | 11/8/16 11:22:00 PM |
|---|---|---|

Font:Not Bold, Font color: Text 1

| Page 32: [41] Formatted | Marc Mallet | 11/8/16 11:22:00 PM |
|---|---|---|

Font:Not Bold, Font color: Text 1

| Page 32: [41] Formatted | Marc Mallet | 11/8/16 11:22:00 PM |
|---|---|---|

Font:Not Bold, Font color: Text 1

| Page 32: [41] Formatted | Marc Mallet | 11/8/16 11:22:00 PM |
|---|---|---|

Font:Not Bold, Font color: Text 1

| Page 32: [41] Formatted | Marc Mallet | 11/8/16 11:22:00 PM |
|---|---|---|

Font:Not Bold, Font color: Text 1

| Page 32: [42] Formatted | Marc Mallet | 11/8/16 11:22:00 PM |
|---|---|---|

Font:10 pt, Font color: Text 1

| Page 32: [42] Formatted | Marc Mallet | 11/8/16 11:22:00 PM |
|---|---|---|

Font:10 pt, Font color: Text 1

| Page 32: [42] Formatted | Marc Mallet | 11/8/16 11:22:00 PM |
|---|---|---|

Font:10 pt, Font color: Text 1

| Page 32: [42] Formatted | Marc Mallet | 11/8/16 11:22:00 PM |
|---|---|---|

Font:10 pt, Font color: Text 1

| Page 32: [42] Formatted | Marc Mallet | 11/8/16 11:22:00 PM |
|---|---|---|

Font:10 pt, Font color: Text 1

| Page 32: [42] Formatted | Marc Mallet | 11/8/16 11:22:00 PM |
|---|---|---|

Font:10 pt, Font color: Text 1

| Page 32: [42] Formatted | Marc Mallet | 11/8/16 11:22:00 PM |
|---|---|---|

Font:10 pt, Font color: Text 1

| Page 32: [42] Formatted | Marc Mallet | 11/8/16 11:22:00 PM |
|---|---|---|

Font:10 pt, Font color: Text 1

| Page 32: [42] Formatted | Marc Mallet | 11/8/16 11:22:00 PM |
|---|---|---|

Font:10 pt, Font color: Text 1

| Page 32: [42] Formatted | Marc Mallet | 11/8/16 11:22:00 PM |
|---|---|---|

Font:10 pt, Font color: Text 1

| Page 32: [42] Formatted | Marc Mallet | 11/8/16 11:22:00 PM |
|---|---|---|

Font:10 pt, Font color: Text 1

| Page 32: [42] Formatted | Marc Mallet | 11/8/16 11:22:00 PM |
|---|---|---|

Font:10 pt, Font color: Text 1

| Page 32: [42] Formatted | Marc Mallet | 11/8/16 11:22:00 PM |
|---|---|---|

Font:10 pt, Font color: Text 1

| Page 32: [42] Formatted | Marc Mallet | 11/8/16 11:22:00 PM |
|---|---|---|

Font:10 pt, Font color: Text 1

| Page 32: [42] Formatted | Marc Mallet | 11/8/16 11:22:00 PM |
|---|---|---|

Font:10 pt, Font color: Text 1

| Page 32: [42] Formatted | Marc Mallet | 11/8/16 11:22:00 PM |
|---|---|---|

Font:10 pt, Font color: Text 1

| Page 32: [42] Formatted | Marc Mallet | 11/8/16 11:22:00 PM |
|---|---|---|

Font:10 pt, Font color: Text 1

| Page 32: [42] Formatted | Marc Mallet | 11/8/16 11:22:00 PM |
|---|---|---|

Font:10 pt, Font color: Text 1

| Page 32: [42] Formatted | Marc Mallet | 11/8/16 11:22:00 PM |
|---|---|---|

Font:10 pt, Font color: Text 1

| Page 32: [42] Formatted | Marc Mallet | 11/8/16 11:22:00 PM |
|---|---|---|

Font:10 pt, Font color: Text 1

| Page 32: [42] Formatted | Marc Mallet | 11/8/16 11:22:00 PM |
|---|---|---|

Font:10 pt, Font color: Text 1

| Page 32: [42] Formatted | Marc Mallet | 11/8/16 11:22:00 PM |
|---|---|---|

Font:10 pt, Font color: Text 1

| Page 32: [42] Formatted | Marc Mallet | 11/8/16 11:22:00 PM |
|---|---|---|

Font:10 pt, Font color: Text 1

| Page 32: [42] Formatted | Marc Mallet | 11/8/16 11:22:00 PM |
|---|---|---|

Font:10 pt, Font color: Text 1

| Page 32: [42] Formatted | Marc Mallet | 11/8/16 11:22:00 PM |
|---|---|---|

Font:10 pt, Font color: Text 1

| Page 32: [42] Formatted | Marc Mallet | 11/8/16 11:22:00 PM |
|---|---|---|

Font:10 pt, Font color: Text 1

| Page 32: [42] Formatted | Marc Mallet | 11/8/16 11:22:00 PM |
|---|---|---|

Font:10 pt, Font color: Text 1

| Page 32: [42] Formatted | Marc Mallet | 11/8/16 11:22:00 PM |
|---|---|---|

Font:10 pt, Font color: Text 1

| Page 32: [42] Formatted | Marc Mallet | 11/8/16 11:22:00 PM |
|---|---|---|

Font:10 pt, Font color: Text 1

| Page 32: [42] Formatted | Marc Mallet | 11/8/16 11:22:00 PM |
|---|---|---|

Font:10 pt, Font color: Text 1

| Page 32: [42] Formatted | Marc Mallet | 11/8/16 11:22:00 PM |
|---|---|---|

Font:10 pt, Font color: Text 1

| Page 32: [42] Formatted | Marc Mallet | 11/8/16 11:22:00 PM |
|---|---|---|

Font:10 pt, Font color: Text 1

| Page 32: [42] Formatted | Marc Mallet | 11/8/16 11:22:00 PM |
|---|---|---|

Font:10 pt, Font color: Text 1

| Page 32: [42] Formatted | Marc Mallet | 11/8/16 11:22:00 PM |
|---|---|---|

Font:10 pt, Font color: Text 1

| Page 32: [42] Formatted | Marc Mallet | 11/8/16 11:22:00 PM |
|---|---|---|

Font:10 pt, Font color: Text 1

| Page 32: [42] Formatted | Marc Mallet | 11/8/16 11:22:00 PM |
|---|---|---|

Font:10 pt, Font color: Text 1

| Page 32: [42] Formatted | Marc Mallet | 11/8/16 11:22:00 PM |
|---|---|---|

Font:10 pt, Font color: Text 1

| Page 32: [42] Formatted | Marc Mallet | 11/8/16 11:22:00 PM |

Font:10 pt, Font color: Text 1

| Page 32: [42] Formatted | Marc Mallet | 11/8/16 11:22:00 PM |

Font:10 pt, Font color: Text 1

| Page 32: [42] Formatted | Marc Mallet | 11/8/16 11:22:00 PM |

Font:10 pt, Font color: Text 1

| Page 32: [42] Formatted | Marc Mallet | 11/8/16 11:22:00 PM |

Font:10 pt, Font color: Text 1

| Page 32: [42] Formatted | Marc Mallet | 11/8/16 11:22:00 PM |

Font:10 pt, Font color: Text 1

| Page 32: [42] Formatted | Marc Mallet | 11/8/16 11:22:00 PM |

Font:10 pt, Font color: Text 1

| Page 32: [42] Formatted | Marc Mallet | 11/8/16 11:22:00 PM |

Font:10 pt, Font color: Text 1

| Page 32: [42] Formatted | Marc Mallet | 11/8/16 11:22:00 PM |

Font:10 pt, Font color: Text 1

| Page 32: [42] Formatted | Marc Mallet | 11/8/16 11:22:00 PM |

Font:10 pt, Font color: Text 1

| Page 32: [42] Formatted | Marc Mallet | 11/8/16 11:22:00 PM |

Font:10 pt, Font color: Text 1

| Page 32: [42] Formatted | Marc Mallet | 11/8/16 11:22:00 PM |

Font:10 pt, Font color: Text 1

| Page 32: [42] Formatted | Marc Mallet | 11/8/16 11:22:00 PM |

Font:10 pt, Font color: Text 1

| Page 32: [42] Formatted | Marc Mallet | 11/8/16 11:22:00 PM |
|---|---|---|

Font:10 pt, Font color: Text 1

| Page 32: [42] Formatted | Marc Mallet | 11/8/16 11:22:00 PM |
|---|---|---|

Font:10 pt, Font color: Text 1

| Page 32: [42] Formatted | Marc Mallet | 11/8/16 11:22:00 PM |
|---|---|---|

Font:10 pt, Font color: Text 1

| Page 32: [42] Formatted | Marc Mallet | 11/8/16 11:22:00 PM |
|---|---|---|

Font:10 pt, Font color: Text 1

| Page 32: [42] Formatted | Marc Mallet | 11/8/16 11:22:00 PM |
|---|---|---|

Font:10 pt, Font color: Text 1

| Page 32: [42] Formatted | Marc Mallet | 11/8/16 11:22:00 PM |
|---|---|---|

Font:10 pt, Font color: Text 1

| Page 32: [42] Formatted | Marc Mallet | 11/8/16 11:22:00 PM |
|---|---|---|

Font:10 pt, Font color: Text 1

| Page 32: [42] Formatted | Marc Mallet | 11/8/16 11:22:00 PM |
|---|---|---|

Font:10 pt, Font color: Text 1

| Page 32: [42] Formatted | Marc Mallet | 11/8/16 11:22:00 PM |
|---|---|---|

Font:10 pt, Font color: Text 1

| Page 32: [42] Formatted | Marc Mallet | 11/8/16 11:22:00 PM |
|---|---|---|

Font:10 pt, Font color: Text 1

| Page 32: [42] Formatted | Marc Mallet | 11/8/16 11:22:00 PM |
|---|---|---|

Font:10 pt, Font color: Text 1

| Page 32: [42] Formatted | Marc Mallet | 11/8/16 11:22:00 PM |
|---|---|---|

Font:10 pt, Font color: Text 1

| Page 32: [42] Formatted | Marc Mallet | 11/8/16 11:22:00 PM |
|---|---|---|

Font:10 pt, Font color: Text 1

| Page 32: [42] Formatted | Marc Mallet | 11/8/16 11:22:00 PM |
|---|---|---|

Font:10 pt, Font color: Text 1

| Page 32: [42] Formatted | Marc Mallet | 11/8/16 11:22:00 PM |
|---|---|---|

Font:10 pt, Font color: Text 1

| Page 32: [42] Formatted | Marc Mallet | 11/8/16 11:22:00 PM |
|---|---|---|

Font:10 pt, Font color: Text 1

| Page 32: [42] Formatted | Marc Mallet | 11/8/16 11:22:00 PM |
|---|---|---|

Font:10 pt, Font color: Text 1

| Page 32: [42] Formatted | Marc Mallet | 11/8/16 11:22:00 PM |
|---|---|---|

Font:10 pt, Font color: Text 1

| Page 32: [42] Formatted | Marc Mallet | 11/8/16 11:22:00 PM |
|---|---|---|

Font:10 pt, Font color: Text 1

| Page 32: [42] Formatted | Marc Mallet | 11/8/16 11:22:00 PM |
|---|---|---|

Font:10 pt, Font color: Text 1

| Page 32: [42] Formatted | Marc Mallet | 11/8/16 11:22:00 PM |
|---|---|---|

Font:10 pt, Font color: Text 1

| Page 32: [42] Formatted | Marc Mallet | 11/8/16 11:22:00 PM |
|---|---|---|

Font:10 pt, Font color: Text 1

| Page 32: [42] Formatted | Marc Mallet | 11/8/16 11:22:00 PM |
|---|---|---|

Font:10 pt, Font color: Text 1

| Page 32: [42] Formatted | Marc Mallet | 11/8/16 11:22:00 PM |
|---|---|---|

Font:10 pt, Font color: Text 1

| Page 32: [42] Formatted | Marc Mallet | 11/8/16 11:22:00 PM |

Font:10 pt, Font color: Text 1

| Page 32: [42] Formatted | Marc Mallet | 11/8/16 11:22:00 PM |

Font:10 pt, Font color: Text 1

| Page 32: [42] Formatted | Marc Mallet | 11/8/16 11:22:00 PM |

Font:10 pt, Font color: Text 1

| Page 32: [42] Formatted | Marc Mallet | 11/8/16 11:22:00 PM |

Font:10 pt, Font color: Text 1

| Page 32: [42] Formatted | Marc Mallet | 11/8/16 11:22:00 PM |

Font:10 pt, Font color: Text 1

| Page 32: [42] Formatted | Marc Mallet | 11/8/16 11:22:00 PM |

Font:10 pt, Font color: Text 1

| Page 32: [42] Formatted | Marc Mallet | 11/8/16 11:22:00 PM |

Font:10 pt, Font color: Text 1

| Page 32: [42] Formatted | Marc Mallet | 11/8/16 11:22:00 PM |

Font:10 pt, Font color: Text 1

| Page 32: [42] Formatted | Marc Mallet | 11/8/16 11:22:00 PM |

Font:10 pt, Font color: Text 1

| Page 32: [42] Formatted | Marc Mallet | 11/8/16 11:22:00 PM |

Font:10 pt, Font color: Text 1

| Page 32: [42] Formatted | Marc Mallet | 11/8/16 11:22:00 PM |

Font:10 pt, Font color: Text 1

| Page 32: [42] Formatted | Marc Mallet | 11/8/16 11:22:00 PM |

Font:10 pt, Font color: Text 1

| Page 32: [42] Formatted | Marc Mallet | 11/8/16 11:22:00 PM |
|---|---|---|

Font:10 pt, Font color: Text 1

| Page 32: [42] Formatted | Marc Mallet | 11/8/16 11:22:00 PM |
|---|---|---|

Font:10 pt, Font color: Text 1

| Page 32: [42] Formatted | Marc Mallet | 11/8/16 11:22:00 PM |
|---|---|---|

Font:10 pt, Font color: Text 1

| Page 32: [42] Formatted | Marc Mallet | 11/8/16 11:22:00 PM |
|---|---|---|

Font:10 pt, Font color: Text 1

| Page 32: [43] Formatted | Marc Mallet | 11/8/16 11:22:00 PM |
|---|---|---|

Font color: Text 1

| Page 32: [43] Formatted | Marc Mallet | 11/8/16 11:22:00 PM |
|---|---|---|

Font color: Text 1

| Page 32: [43] Formatted | Marc Mallet | 11/8/16 11:22:00 PM |
|---|---|---|

Font color: Text 1

| Page 32: [43] Formatted | Marc Mallet | 11/8/16 11:22:00 PM |
|---|---|---|

Font color: Text 1

| Page 32: [43] Formatted | Marc Mallet | 11/8/16 11:22:00 PM |
|---|---|---|

Font color: Text 1

| Page 32: [44] Formatted | Marc Mallet | 11/8/16 11:22:00 PM |
|---|---|---|

Font color: Text 1

| Page 32: [44] Formatted | Marc Mallet | 11/8/16 11:22:00 PM |
|---|---|---|

Font color: Text 1

| Page 32: [44] Formatted | Marc Mallet | 11/8/16 11:22:00 PM |
|---|---|---|

Font color: Text 1

| Page 32: [44] Formatted | Marc Mallet | 11/8/16 11:22:00 PM |
|---|---|---|

Font color: Text 1

| Page 32: [44] Formatted | Marc Mallet | 11/8/16 11:22:00 PM |
|---|---|---|

Font color: Text 1

| Page 32: [44] Formatted | Marc Mallet | 11/8/16 11:22:00 PM |
|---|---|---|

Font color: Text 1

| Page 32: [45] Moved to page 25 (Move #1) | Marc Mallet | 5/2/17 1:32:00 AM |
|---|---|---|

[Figure]

[MM14]

Figure 5 The time series of the major measured gaseous species during the SAFIRED campaign: (a) carbon monoxide, (b) carbon dioxide, (c) methane, (d) nitrogen dioxide[MM15], (e) gaseous elemental mercury, (f) acetonitrile and (g) ozone and $\Delta O_3 / \Delta CO$. The biomass burning and coastal periods are indicated by the red dotted lines. All parts-per notation refer to mole fractions unless otherwise indicated.

| Page 32: [46] Formatted | Marc Mallet | 11/8/16 11:22:00 PM |
|---|---|---|

Font color: Text 1

| Page 32: [46] Formatted | Marc Mallet | 11/8/16 11:22:00 PM |
|---|---|---|

Font color: Text 1

| Page 32: [46] Formatted | Marc Mallet | 11/8/16 11:22:00 PM |
|---|---|---|

Font color: Text 1

| Page 32: [46] Formatted | Marc Mallet | 11/8/16 11:22:00 PM |
|---|---|---|

Font color: Text 1

| Page 32: [46] Formatted | Marc Mallet | 11/8/16 11:22:00 PM |
|---|---|---|

Font color: Text 1

| Page 32: [47] Moved to page 26 (Move #2) | Marc Mallet | 5/2/17 1:32:00 AM |
|---|---|---|

[Figure]

[MM16]

Figure 6 The times series of the major aerosol properties during the SAFIRED campaign: (a) the non-refractory PM$_1$ organic mass concentration (left) and organic mass fraction (right), b) the inorganic non-refractory PM$_1$ mass concentrations, (c) the 12-hour filter OC and EC PM$_1$ mass concentrations (left) and the ratio of OC to OC+EC (right), (d) the particle size distributions and particle size mode (left) and the total particle number concentration (right) and (e) the wind direction at ATARS.

| Page 32: [48] Formatted | Marc Mallet | 11/8/16 11:22:00 PM |
|---|---|---|

Font color: Text 1

| Page 32: [48] Formatted | Marc Mallet | 11/8/16 11:22:00 PM |
|---|---|---|

Font color: Text 1

| Page 32: [48] Formatted | Marc Mallet | 11/8/16 11:22:00 PM |
|---|---|---|

Font color: Text 1

| Page 32: [48] Formatted | Marc Mallet | 11/8/16 11:22:00 PM |
|---|---|---|

Font color: Text 1

| Page 32: [48] Formatted | Marc Mallet | 11/8/16 11:22:00 PM |
|---|---|---|

Font color: Text 1

| Page 32: [48] Formatted | Marc Mallet | 11/8/16 11:22:00 PM |
|---|---|---|

Font color: Text 1

| Page 32: [48] Formatted | Marc Mallet | 11/8/16 11:22:00 PM |
|---|---|---|

Font color: Text 1

| Page 32: [48] Formatted | Marc Mallet | 11/8/16 11:22:00 PM |
|---|---|---|

Font color: Text 1

| Page 34: [49] Deleted | Marc Mallet | 5/14/17 10:42:00 PM |
|---|---|---|

BBP1, BBP2 and BBP4 correspond to the periods when fires were burning within 10 km of ATARS. Large enhancements of biomass burning related emissions were observed during these three periods. There were distinct enhancements of all measured gaseous and aerosol species during these periods. Differences between the maximum and background concentrations were very prominent for CO (note the logarithmic scale in Figure 5a), $CH_4$, $N_2O$, acetonitrile (an established marker for biomass burning) and organic, nitrate and chloride non-refractory sub-micron aerosol species. Similar enhancements of $CH_4$ were also observed outside of these BB periods, which suggests another source of methane in this region. Only slight enhancements of GEM concentrations above background were observed during BBP2 and BBP4. Similar to much of the rest of the campaign sampling period, the non-refractory submicron aerosol was dominated by organics, with contributions typically varying between 70% and 95% of the mass. Relative to background concentrations, there were also large enhancements of nitrate and chloride species during these periods. While there were also enhancements of sulfate and ammonium species during these periods, similar enhancements were observed outside of these periods, again indicating a non-fire source of these species. The ratio of $O_3$ to CO concentrations above background (taken as 10 ppbv and 66 ppbv, respectively) gives an indication of the photochemical age of a smoke plume. $\Delta O_3/\Delta CO$ were lowest during BBP2 and BBP4 (and not measured during BBP1) relative to the rest of the campaign, indicating that the biomass burning signals during these periods had not undergone extensive photochemical aging and are therefore characteristic of fresh smoke. [MM17]

Elevated signals during BBP1 were likely a result of a series of close fires within 5 km ENE of ATARS. The VIIRS and MODIS sensors on the SUMO NPP, Terra and Aqua satellites observes smaller fires at approximately 2 pm on the 30th of May. Winds were northeasterly during these two events. It is therefore likely that these signals were continuation or evolution of those fires. Burned vegetation was also visually observed the next morning at these locations. The large burst event later on the evening of the 31st of May is unlikely to be associated with these fires as the wind direction during this event was from the SSW and SSE. Large clusters of fires were observed at approximately 100 km and 150 km SE of the station by the Terra and Aqua satellites. The signals observed during this event could be a result of the plumes from this fire, although the possibility of a fire ignited after the satellite flyovers, or a combination of these cannot be eliminated.

Large signal enhancements on the 8th of June during BBP2 is likely a result of a cluster of fires approximately 100 km south east of the station. The MODIS sensors on the Terra and Aqua satellites observed the small cluster of fires along the back-trajectory at 11:14 am and 1:56 pm.

The source of BB emissions for the large event on the 9th of June during this period is unclear. Several fires approximately 5 km from the station along the back-trajectory were detected by the MODIS sensor on Aqua and the VIIRS sensor on SUOMI NPP at approximately 2:30 pm on the 9th of June. There were also numerous fires detected between 100 km and 200 km southeast along this trajectory. The signals associated with this event could therefore be a result of the closer fires that started to blaze later in the evening, the distant fires or a combination of both.

Only one fire within 20km of ATARS was observed during BBP3 on the 17[th] of June. Numerous fires were observed further than 20km from the station and is possible that the signals during this period were more aged. While photochemical aging and coagulation typically lead to larger particles, particle size distributions were smaller during this period and the ratio of OC to OC+EC was 70%, 10% lower than the ratio during the rest of the campaign. Whether these observations were a result of burn conditions or aging processes [MM18][MM19]

| Page 34: [50] Deleted | Marc Mallet | 5/14/17 10:42:00 PM |
|---|---|---|

(i.e. evaporation of organic compounds from the aerosol phase) is unclear, although the highest $\Delta O_3/\Delta CO$ values during the campaign were observed during BBP3, which indicates photochemical aging was more extensive during this period.

One close fire was also observed during CP, however wind directions during this period were typically north-easterly and concentrations were therefore much lower. 5-day HYSPLIT trajectories also show that air mass during the CP originated along the east coast of Australia before travelling towards the sampling station with very little terrestrial influence.

Close proximity fires[MM20]

With numerous fires occurring across the region and the limitations of once-per-day satellite fly-overs and stationary measurements, it can be difficult to identify the exact source of these elevated signals. For a portion of BBP4, however, fires were burning within several kilometers of ATARS and several plumes were easily observed from the station. The signals from these plumes are shown in Figure 8. The observed enhancements between 12:30 pm and 3:00 pm on the 25th June during BBP4 were a result of grass fires burning approximately 1 km south-east from the station. During this event, the wind direction (Figure 8k) was highly variable, changing between 140° and 80° True Bearing (TB) multiple times. As a result, the sampling changed from measuring the air mass with and without the plume from this fire, which led to sharp increases and decreases in biomass burning-related signals (Figure 8a through 8j). Visually, the fire area and extent of the plume was larger at 4:00pm than earlier, however the wind direction changed to north-easterly which directed the plume away from the station. From 4:00 pm until 10:00 pm, the wind direction was stable at approximately 50° TB. At 10:00 pm, the wind direction rapidly changed to directly south and the largest enhancements for the whole campaign were observed until approximately 2:00 am on the 26th of June. It is very likely that these signals were a result of a continuation and evolution of these fires as the night progressed. Portions of a ~0.25 $km^2$ grassland field within 500 m directly south of ATARS were observed to be burned upon arrival at the station on the morning of the 26th of June and we speculate that the burning of this field contributed to the large enhancements in measured biomass burning emissions. The emissions during this portion of BBP4 are likely to be the most representative of fresh biomass burning smoke during the SAFIRED campaign. [MM21]

[Figure]

[MM22][MM23]

**Figure 8 The major gas and aerosol concentrations measured during two biomass burning events within 1 km of ATARS during BBP4. (a) through (g) and (h) through (k) are as per Figures 5 and 6, respectively. All parts-per notation or mole fractions unless otherwise indicated.**
* * *
| Page 34: [52] Commented | Marc Mallet | 3/20/17 4:36:00 PM |
|---|---|---|

R3:

11. The 'Outcomes' section could benefit from the following: (i) discussion of the re- sults in the context of earlier work and how the findings here are similar or different, (ii) how the SAFIRED measurements were insightful (iii) what questions still remain unanswered, and (iv) directions for future work.
* * *
| Page 34: [53] Commented | Marc Mallet | 3/20/17 4:37:00 PM |
|---|---|---|

R3:

12. Similar to comment 5, a Table listing the companion publications and its central finding would be helpful for the interested reader to track the measurement-specific paper.
* * *
| Page 37: [54] Deleted | Marc Mallet | 5/14/17 10:49:00 PM |
|---|---|---|

Australian fires are responsible for 6% of global $CO_2$ biomass burning emissions, most of which is due to savannah fires (Shi et al., 2015). Carbon sequestering during regrowth periods is considered to balance carbon emissions in tropical Australia (Haverd et al., 2013). Greenhouse gases emitted from savannah fires that are not sequestered, such as methane ($CH_4$) and nitrous oxide ($N_2O$), have been shown to contribute 2-4% of the annual accountable greenhouse gas emissions from Australia (Meyer et al., 2012). Seasonal emission factors for the major greenhouse gases are important for national greenhouse gas inventories and in understanding the impact of savannah fires. Furthermore, emission factors of $CO_2$ and CO can be used to infer mechanisms behind the emissions of other species, such as the connection between particulate matter and burning conditions.

| Page 37: [55] Formatted | Marc Mallet | 11/8/16 11:22:00 PM |
|---|---|---|

Font color: Text 1

| Page 37: [56] Formatted | Marc Mallet | 11/8/16 11:22:00 PM |
|---|---|---|

Font color: Text 1

| Page 37: [57] Formatted | Marc Mallet | 11/8/16 11:22:00 PM |
|---|---|---|

Font color: Text 1

| Page 37: [58] Formatted | Marc Mallet | 11/8/16 11:22:00 PM |
|---|---|---|

Font color: Text 1

| Page 37: [59] Formatted | Marc Mallet | 11/8/16 11:22:00 PM |
|---|---|---|

Font color: Text 1

| Page 37: [60] Formatted | Marc Mallet | 11/8/16 11:22:00 PM |
|---|---|---|

Font color: Text 1

| Page 37: [61] Formatted | Marc Mallet | 11/8/16 11:22:00 PM |
|---|---|---|

Font color: Text 1

The gaseous and aerosol data for the sample period were investigated to identify BB events and determine the emission factors of $CO_2$, CO, $CH_4$. $N_2O$, as well as Aitken and Accumulation mode aerosols and submicron particle species (organics, sulfates, nitrates, ammonium and chlorides) for several individual BB events. These emission factors were mostly found to be dependent on the combustion conditions (using the modified combustion efficiency as a proxy) of the fires. These results will be the first set of emission factors for aerosol particles from savannah fires in Australia. Furthermore, the variability in emission factors for different fires calls for a separation of single-value emission factors that are usually reported for savannah fires into grass and shrub components. A full discussion of these results are presented in Desservettaz et al. (2016, submitted).

Non-methane organic compounds (NMOCs)[MM24]

Biomass burning is the second largest source of NMOCs globally with a recent global estimate of at least 400 Tg year$^{-1}$, second only to biogenic sources (Akagi et al., 2011). Biomass burning produces a complex mix of NMOCs, which may be saturated or unsaturated, aliphatic or aromatic, and contain substitutions of oxygen, sulfur, nitrogen, halogens and other atoms. NMOC emission rates are strongly tied to the efficiency of combustion, with smouldering fires emitting NMOC at higher rates than flaming fires (Andreae and Merlet, 2001). Biomass burning derived NMOCs fuel the production of tropospheric ozone in diluted, aged biomass burning plumes, with higher ozone enhancements observed when biomass burning plumes interact with NOx-rich urban plumes (Jaffe and Wigder, 2012;Wigder et al., 2013;Akagi et al., 2013). Oxidation of NMOCs results in lower volatility products that partition to the aerosol phase and contribute significantly to secondary organic aerosol (Hallquist et al., 2009). Biomass burning produces significant amounts of semivolatile NMOC which can be difficult to quantify and identify with current measurement techniques. However recent studies have shown that including semi volatile NMOC chemistry in models improves the agreement between the modeled and observed organic aerosol (Alvarado et al., 2015; Konovalov et al., 2015) and ozone (Alvarado et al., 2015). High quality NMOC emission factors are crucial for models to assess the impact of biomass burning plumes on air quality and climate.

PAHs

Polycyclic aromatic hydrocarbons (PAHs) are a group of chemicals that are formed and emitted during combustion processes. Globally, major sources include residential/commercial biomass burning, open-field biomass burning and vehicular emissions (Shen et al., 2013). In Oceania in 2007, 31% of PAH emissions were estimated to be attributed to deforestation and wildfires (Shen et al., 2013). With control strategies targeting and reducing vehicular emission of PAHs over the last few decades, the relative contribution of other emission sources, such as savannah fires, has increased (Friedman et al., 2013;Kallenborn et al., 2012;Wang et al., 2016). Although most of these emissions are in the gas-phase (Jenkins et al., 1996;Atkins et al., 2010), the particle-phase PAHs, such as benzo[a]pyrene (BaP), may have high genotoxicity (IARC., 2015). However, field-based studies on emissions of PAHs from open-field biomass burning, including savannah fires remain limited in Australia (Freeman and Cattell, 1990).

Emission factors of PAHs from biomass burning related to savannah fires in northern Australia will be estimated form the data collected during this campaign. This estimation will be based on the (background subtracted) concentrations of PAHs and $CO_2$ (and CO) during the events where biomass burning contributes most to these concentrations measured at the sampling site. The concentrations of 13 major PAHs (gaseous plus particle-associated phase) varied from ~ 1 to over ng m$^{-3}$ within different BB events. In the gas phase, 3- and 4-ring compounds typically contributed ~ 90% to the sum concentrations whereas the particle-associated PAHs were dominated by 5- and 6-ring compounds (> 80%). Measured PAH concentrations were significantly higher (paired $t$-test, $P < <0.05$) during BB events E, F and G. For these events, concentrations of

BaP exceeded the monitoring investigation level for atmospheric BaP in Australia (National-

Environment-Protection-Council-Service-Corporation, 2011), i.e. 0.30 ng m$^{-3}$, by 66% (BB event

E) and 200% (BB events F and G). A full discussion of these results can be found in (Wang et al.,

2016, under review).

Mercury

The atmosphere is the dominant transport pathway for mercury globally, with emissions to the atmosphere from both natural and anthropogenic origins (Driscoll et al., 2013). Whilst our understanding of the natural cycling of mercury has improved markedly over the past decades (Pirrone et al., 2010), large uncertainties still exist; specifically, global emission estimates to the atmosphere from biomass burning currently range between 300 and 600 Mg year$^{-1}$ (Driscoll et al.,

2013). In the atmosphere, mercury exists as one of three operationally-defined species: gaseous elemental mercury (GEM), gaseous oxidised mercury (GOM) and particulate-bound mercury (PBM), each with differing abundances, solubility and depositional characteristics and with in-air conversion between all three species possible (Lin and Pehkonen, 1999). Mercury can be scavenged from the atmosphere through both wet and dry depositional processes, and the monsoonal climate of northern Australia results in varying significance of each of these processes through the year (Packham et al., 2009). Upon deposition, mercury may be stored in plant tissue via stomatal or cuticular uptake (Rea et al., 2002) or sequestered within soils (Gustin et al., 2008).

Release from both of these pools is achieved from burning events that may volatilise or thermally desorb mercury from biomass and soil, respectively (Melendez-Perez et al., 2014). Subsequently this mercury pool is redistributed through the atmospheric pathway to ecosystems that may methylate mercury, thereby enhancing its bioavailability to the local food chain.

SAFIRED represents the first measurements of atmospheric mercury undertaken in the tropical region of the Australian continent. The mean observed GEM concentration over the study period was $0.99 \pm 0.09$ ng m$^{-3}$, similar to the average over that month (0.96 ng m$^{-3}$) for 5 other Southern Hemisphere sites and slightly lower than the average (1.15 ng m$^{-3}$) for 5 tropical sites (Sprovieri et al., 2016). Mean GOM and PBM concentrations were $11 \pm 5$ pg m$^{-3}$ and $6 \pm 3$ pg m$^{-3}$ respectively, representing 0.6 – 3.4% of total observed atmospheric mercury.

Atmospheric mercury measurements were available only during the final four identified burn events. During these events, spikes in GEM concentrations were observed, though there were no significant increases in GOM or PBM. Emission ratios calculated during the campaign were two orders of magnitude higher than those reported by Andreae and Merlet (2001), though those were from scrub, rather than grass, BB events (Desservettaz et al., 2016). Future outcomes from the SAFIRED campaign will focus on the use of micrometeorological techniques and the passive tracer radon to quantify delivery of atmospheric mercury to tropical savannah ecosystems. ATARS also now serves as an additional site measuring continuous GEM as part of the Global Mercury Observation System (GMOS), one of only two tropical observing sites in the Eastern Hemisphere and the third such site located in Australia.

**Aging of aerosols**

| Page 37: [63] Formatted | Marc Mallet | 11/8/16 11:22:00 PM |
|---|---|---|

Font color: Text 1

| Page 37: [64] Formatted | Marc Mallet | 11/8/16 11:22:00 PM |
|---|---|---|

Font color: Text 1

| Page 37: [65] Formatted | Marc Mallet | 11/8/16 11:22:00 PM |
|---|---|---|

Font color: Text 1

| Page 37: [66] Formatted | Marc Mallet | 11/8/16 11:22:00 PM |
|---|---|---|

Font color: Text 1

| Page 37: [67] Formatted | Marc Mallet | 11/8/16 11:22:00 PM |
|---|---|---|

Font color: Text 1

| Page 37: [68] Formatted | Marc Mallet | 11/8/16 11:22:00 PM |
|---|---|---|

Font color: Text 1

| Page 37: [69] Formatted | Marc Mallet | 11/8/16 11:22:00 PM |
|---|---|---|

Font color: Text 1

| Page 37: [70] Formatted | Marc Mallet | 11/8/16 11:22:00 PM |
|---|---|---|

Font color: Text 1

| Page 37: [71] Formatted | Marc Mallet | 11/8/16 11:22:00 PM |
|---|---|---|

Font color: Text 1

| Page 37: [72] Formatted | Marc Mallet | 11/8/16 11:22:00 PM |
|---|---|---|

Font color: Text 1

| Page 37: [73] Formatted | Marc Mallet | 11/8/16 11:22:00 PM |
|---|---|---|

Font color: Text 1

| Page 37: [74] Formatted | Marc Mallet | 11/8/16 11:22:00 PM |
|---|---|---|

Font color: Text 1

| Page 37: [75] Formatted | Marc Mallet | 11/8/16 11:22:00 PM |
|---|---|---|

Font color: Text 1

| Page 37: [76] Formatted | Marc Mallet | 11/8/16 11:22:00 PM |
|---|---|---|

Font color: Text 1

| Page 37: [77] Formatted | Marc Mallet | 11/8/16 11:22:00 PM |
|---|---|---|

Font color: Text 1

| Page 37: [78] Formatted | Marc Mallet | 11/8/16 11:22:00 PM |
|---|---|---|

Font color: Text 1

| Page 37: [79] Formatted | Marc Mallet | 11/8/16 11:22:00 PM |
|---|---|---|

Font color: Text 1

| Page 37: [80] Formatted | Marc Mallet | 11/8/16 11:22:00 PM |
|---|---|---|

Font color: Text 1

| Page 37: [81] Formatted | Marc Mallet | 11/8/16 11:22:00 PM |
|---|---|---|

Font color: Text 1

| Page 37: [82] Formatted | Marc Mallet | 11/8/16 11:22:00 PM |
|---|---|---|

Font color: Text 1

| Page 37: [83] Formatted | Marc Mallet | 11/8/16 11:22:00 PM |
|---|---|---|

Font color: Text 1

| Page 37: [84] Formatted | Marc Mallet | 11/8/16 11:22:00 PM |
|---|---|---|

Font color: Text 1

| Page 37: [85] Formatted | Marc Mallet | 11/8/16 11:22:00 PM |
|---|---|---|

Font color: Text 1

| Page 37: [86] Formatted | Marc Mallet | 11/8/16 11:22:00 PM |

Font color: Text 1

| Page 37: [87] Formatted | Marc Mallet | 11/8/16 11:22:00 PM |

Font color: Text 1

| Page 37: [88] Formatted | Marc Mallet | 11/8/16 11:22:00 PM |

Font color: Text 1

| Page 37: [89] Formatted | Marc Mallet | 11/8/16 11:22:00 PM |

Font color: Text 1

| Page 37: [90] Formatted | Marc Mallet | 11/8/16 11:22:00 PM |

Font color: Text 1

| Page 37: [91] Formatted | Marc Mallet | 11/8/16 11:22:00 PM |

Font color: Text 1

| Page 37: [92] Formatted | Marc Mallet | 11/8/16 11:22:00 PM |

Font color: Text 1

| Page 37: [93] Formatted | Marc Mallet | 11/8/16 11:22:00 PM |

Font color: Text 1

| Page 37: [94] Formatted | Marc Mallet | 11/8/16 11:22:00 PM |

Font color: Text 1

| Page 37: [95] Formatted | Marc Mallet | 11/8/16 11:22:00 PM |

Font color: Text 1

| Page 37: [96] Formatted | Marc Mallet | 11/8/16 11:22:00 PM |

Font color: Text 1

| Page 37: [97] Formatted | Marc Mallet | 11/8/16 11:22:00 PM |

Font color: Text 1

| **Page 37: [98] Formatted** | **Marc Mallet** | **11/8/16 11:22:00 PM** |
|---|---|---|

Font color: Text 1

| **Page 37: [99] Formatted** | **Marc Mallet** | **11/8/16 11:22:00 PM** |
|---|---|---|

Font color: Text 1

| **Page 37: [100] Formatted** | **Marc Mallet** | **11/8/16 11:22:00 PM** |
|---|---|---|

Font color: Text 1

| **Page 37: [101] Formatted** | **Marc Mallet** | **11/8/16 11:22:00 PM** |
|---|---|---|

Font color: Text 1

| **Page 37: [102] Formatted** | **Marc Mallet** | **11/8/16 11:22:00 PM** |
|---|---|---|

Font color: Text 1

| **Page 39: [103] Deleted** | **Marc Mallet** | **5/14/17 10:54:00 PM** |
|---|---|---|

Atmospheric chemistry and radiative forcing will depend on how gaseous and aerosol emissions from fires age as they move and interact with each other and existing species in the atmosphere. Biomass burning aerosols can be involved in condensation and coagulation (Radhi et al., 2012), undergo water uptake (Mochida and Kawamura, 2004) form cloud droplets (Novakov and Corrigan, 1996), and be exposed to photochemical aging processes, including those involving the gaseous components of fire emissions (Keywood et al., 2011;Keywood et al., 2015). With a reported lifetime of $3.8 \pm 0.8$ days (Edwards et al., 2006), biomass burning aerosols are able to travel intercontinental distances (Rosen et al., 2000) and are therefore present in the atmosphere long enough for substantial changes due to aging. Furthermore, tropical convection is likely to affect the aging of BB emissions in the region around ATARS, due to the immediate proximity to the warm waters in the Timor Sea (Allen et al., 2008). This introduces further uncertainty to the effect of BB emissions on radiation flux.

Primary organic aerosol directly emitted from biomass burning can interact with NMOCs to change composition and mass, resulting in secondary organic aerosol (Hallquist et al., 2009)[MM25]. Photochemical oxidation of NMOCs occurs during the daytime by either hydroxyl radicals or ozone. Ozone is also typically produced in the aging processes of tropical biomass burning plumes when NMOCs can oxidise to produce peroxy radicals that react with NO. Photochemical reactions also may lead to an overall increase in total aerosol mass through the condensation of NMOCs onto existing particles (Reid et al., 1998;Yokelson et al., 2009;Akagi et al., 2012;DeCarlo et al., 2008). Some studies have shown the opposite, i.e., photo-oxidation can also lead to the evaporation of some primary organic constituents, resulting in an overall mass reduction (Hennigan et al., 2011;Akagi et al., 2012). With thousands of organic compounds in the atmosphere, each with different volatilities and potential reaction mechanisms, our understanding of secondary organic aerosol production is limited (Goldstein and Galbally, 2007;Keywood et al., 2011). Furthermore, secondary organic aerosol can also form through aqueous phase reactions where water-soluble organics dissolve into water on existing particles (Lim et al., 2010).

Further analysis into the aerosol chemical composition will elucidate the aging of early dry season biomass burning emissions. Fractional analysis (e.g., f44 and f60, the fraction of m/z 44 and m/z 60 to all organic masses, indicated oxygenation and BB sources, respectively) and factor analysis using positive matrix factorisation (PMF) of cToF-AMS data has been investigated over the entire sampling period. Outside of the periods of significant influence from BB events, three PMF-resolved organic aerosol factors were identified. A BB organic aerosol factor was found to comprise 24% of the submicron non-refractory organic mass, with an oxygenated organic aerosol factor and a biogenic isoprene-related secondary organic aerosol factor comprising 47% and 29%, respectively. These results indicate the significant influence of fresh and aged BB on aerosol composition in the early dry season. The emission of precursors from fires is likely responsible for some of the SOA formation. A full discussion of these results can be found in Milic et al. (2016). Future analysis will investigate the gas and particle-phase composition for individual BB events.

Water uptake of aerosols

The water uptake by aerosols is determined by their size and composition, as well as the atmospheric humidity (McFiggans et al., 2006). The hygroscopic properties of all of the different components of an aerosol particle contribute to its total hygroscopicity (Chen et al., 1973;Stokes and Robinson, 1966). The presence of different water-soluble and water-insoluble organics and inorganics will therefore strongly influence water uptake. Furthermore, chamber studies that have investigated emissions from biomass fuels, both separately and in combination, have shown that the hygroscopic response can vary significantly depending on fuel type (Carrico et al., 2010). Understanding the water uptake of atmospheric aerosols is further complicated when considering other aging processes as described previously. Nonetheless, it is important to characterise the water uptake, as this will, in turn, influence other atmospheric chemistry processes, radiation scattering and absorption as well as cloud processing.

Biomass burning aerosols can act as cloud condensation nuclei if they are large enough for water to easily condense onto their surface, or if the particles have a large affinity for water due to their composition (Novakov and Corrigan, 1996). Ultimately, this means that BB emissions can lead to a higher number of cloud droplets. This is important in reflecting solar radiation and cooling the earth's surface. Cloud albedo is more susceptible to changes when cloud condensation nuclei concentrations are relatively low (Twomey, 1991), such as in marine environments like the Timor Sea off the coast of northern Australia.

The water uptake of aerosols has been further investigated to identify the possible influence of early dry season BB in this region on cloud formation. The concentrations of cloud condensation nuclei at a constant supersaturation of 0.5% were typically of the order of 2000 $cm^{-3}$ and reached well over 10000 $cm^{-3}$ during intense BB events. Variations in the ratio of aerosol particles activating cloud droplets showed a distinct diurnal trend, with an activation ratio of 40% ± 20% during the night and 60% ± 20% during the day. The particle size distribution and the hygroscopicity of the particles were found to significantly influence this activation ratio. A full discussion of these results can be found in Mallet et al. (2016, submitted). Future analysis will elucidate the contribution of different biomass burning aerosol components on the hygroscopicity.

Trace metal deposition

The deposition and dissolution of aerosols containing trace metals into the ocean may provide important micronutrients required for marine primary production. Conversely, the deposition of soluble iron can trigger toxic algal blooms, such as Trichodesmium, in nutrient-poor tropical and subtropical waters (LaRoche and Breitbarth, 2005). Trichodesmium blooms require large quantities of soluble iron, of which aerosols are a source (Boyd and Ellwood, 2010;Rubin et al., 2011). To date, most studies have assumed that mineral dust aerosols represent the primary source of soluble iron in the atmosphere (Baker and Croot, 2010); however fire emissions and oil combustion are other likely sources (Ito, 2011;Schroth et al., 2009;Sedwick et al., 2007). A few studies have shown that iron contained in biomass burning emissions is significantly more soluble than mineral dust (Guieu et al., 2005;Luo et al., 2008;Schroth et al., 2009) but, to date, no data exists for Australian fires.

The aim of the trace metal aerosol component of SAFIRED is to quantify, for the first time, the fractional solubility of aerosol iron, and other trace metals, derived from Australian dry season BB. The fractional iron solubility is an important variable determining iron availability for biological uptake. On a global scale, the large variability in the observed fractional iron solubility results, in part, from a mixture of different aerosol sources. Estimates of fractional iron solubility from fire combustion  (1 - 60 %) are thought to be greater than those originating from mineral dust (1 - 2%) (Chuang et al., 2005;Guieu et al., 2005;Sedwick et al., 2007), and may vary in relationship to biomass and fire characteristics as well as that of the underlying terrain (Paris et al., 2010;Ito, 2011). Iron associated with BB may provide information with respect to BB inputs of iron to the ocean (Giglio et al., 2013;e.g. Meyer et al., 2008). The ATARS provides an ideal location to further investigate BB derived fractional iron solubility at the source. The results from this study can be found in Winton et al. (2016) and show that soluble iron concentrations from BB sources are significantly higher than those observed in Southern Ocean baseline air masses from the Cape Grim Baseline Air Pollution Station, Tasmania, Australia (Winton et al., 2015). Aerosol iron at SAFIRED was a mixture of fresh BB, mineral dust, sea spray and industrial pollution sources. The fractional iron solubility (2 - 12%) was relatively high throughout the campaign and the variability was related to the mixing and enhancement of mineral dust iron solubility with BB species.